# Reappraising the appropriate calculation of a common meteorological quantity: Potential Temperature

Manuel Baumgartner[1,2], Ralf Weigel[2], Allan H. Harvey[3], Felix Plöger[4,5], Ulrich Achatz[6], and Peter Spichtinger[2]

[1]Zentrum für Datenverarbeitung, Johannes Gutenberg University Mainz, Germany
[2]Institute for Atmospheric Physics, Johannes Gutenberg University Mainz, Germany
[3]Applied Chemicals and Materials Division, National Institute of Standards and Technology, Boulder, CO, USA
[4]Forschungszentrum Jülich GmbH, Institute of Energy and Climate Research (IEK-7), Jülich, Germany
[5]Institute for Atmospheric and Environmental Research, University of Wuppertal, Wuppertal, Germany
[6]Institut für Atmosphäre und Umwelt, Goethe-Universität Frankfurt, Frankfurt am Main, Germany

**Correspondence:** Manuel Baumgartner (manuel.baumgartner@uni-mainz.de)

**Abstract.** The potential temperature is a widely used quantity in atmospheric science since it is conserved for dry air's adiabatic changes of state. Its definition involves the specific heat capacity of dry air, which is traditionally assumed as constant. However, the literature provides different values of this allegedly constant parameter, which are reviewed and discussed in this study. Furthermore, we derive the potential temperature for a temperature-dependent parameterisation of the specific heat capacity of dry air, thus providing a new reference potential temperature with a more rigorous basis. This new reference shows different values and vertical gradients, in particular in the stratosphere and above, compared to the potential temperature that assumes constant heat capacity. The application of the new reference potential temperature is discussed for computations of the Brunt-Väisälä frequency, Ertel's potential vorticity, diabatic heating rates, and for the vertical sorting of observational data.

## 1 Introduction

According to the book *Thermodynamics of the Atmosphere* by Alfred Wegener (1911), the first published use of the expression *potential temperature* in meteorology is credited to Wladimir Köppen (1888)[1] and Wilhelm von Bezold (1888), both following the conclusions of Hermann von Helmholtz (1888) (see also Kutzbach, 2016). Even prior to the introduction of the entropy, Poisson (1833) and Thomson (1862) used the "adiabatic equation", the basis of what is understood today as "potential temperature"[2], to describe adiabatic processes, e.g., the coincident variation of temperature and pressure on the movement of air, which is "independent of the effects produced by the radiation or conduction of heat" (Thomson, 1862)[3]. Approximately 26 years later, von Helmholtz perceived that within the atmosphere the heat exchange between air masses of different temperatures, which are relatively moved, is insufficiently explained by heat transfer due only to radiation and convection. He argued

---

[1]Wegener mentioned a talk given by Köppen in a footnote on page 111. In the publication year (1911) of Wegener's book, Köppen's daughter Else got engaged to Alfred Wegener (Reinke-Kunze, 2013) and they married in the year 1913 (Hallam, 1975).

[2]Cf. Bauer (1908) where, for the first time, the potential temperature and the entropy are set in a relationship.

[3]These early applications of entropy in meteorology are also documented in Marquet (2019).

that wind phenomena (e.g., the trade winds), storm events, and the atmospheric circulation were more intense, of larger extent, and more persistent than observed if the air's heat exchange within the discontinuity region (the friction surface of the different air masses) was not mainly due to eddy-driven mixing. On his way to analytically describe the heat exchange of different air masses within the atmosphere, in May of 1880, von Helmholtz introduced the air's *immanent heat* while its absolute temperature changes with changing pressure (von Helmholtz, 1888). In essence, von Helmholtz concluded that the temperature gained by a volume of dry air due to its adiabatic descent from a certain initial pressure level ($p$) to ground pressure ($p_0$) corresponds to the air's immanent heat. In November of the same year, in agreement with von Helmholtz and probably inspired by a presentation that was given in June by Köppen (1888), this property was renamed and reintroduced as the air's *potential temperature* ($\theta$ in the following) by von Bezold (1888) with the following definition for strictly adiabatic changes of state:

$$\theta = T \left( \frac{p_0}{p} \right)^{\frac{\gamma - 1}{\gamma}}, \tag{1}$$

where $T$ and $p$ are the absolute temperature and pressure, respectively, of an air parcel at a certain initial (pressure-) altitude level. The quantities $\theta$ and $p_0$ are corresponding values of the same air parcel's absolute temperature and pressure if the air was exposed to conditions at ground level. The dimensionless coefficient $\gamma$, nowadays called the isentropic exponent, was specified as $1.41$ (von Bezold, 1888).

Moreover, in the same publication, von Bezold concluded that for moist air's adiabatic changes of state, its potential temperature remains unchanged as long as the change of state occurs within dry-adiabatic limits; and further, if there is condensation and precipitation, the potential temperature changes by a magnitude that is determined by the amount of water that falls out of the air parcel. From a modern perspective, it is clear that the air parcel is an isolated thermodynamic system, and adiabatic processes correspond to processes with conserved entropy (i.e., isentropic processes). The description of the immanent heat is then equivalent to the thermodynamic state function entropy, which corresponds to potential temperature of dry air in a one-to-one relationship.

In general, the potential temperature has the benefit of providing a practicable vertical coordinate (equivalent to the pressure level or the altitude above, e.g., sea level) to visualise and analyse the vertical distribution and variability of (measured) data related to any type of atmospheric parameter. Admittedly, the use of the potential temperature as a vertical coordinate is initially less intuitive than applying altitude or pressure coordinates. Indeed, the potential temperature bears a certain abstractness to describe an air parcel's state at a certain altitude level by its imaginary dry-adiabatic descent to ground conditions. However, one major advantage of using the potential temperature as a vertical coordinate is that the (measured) data are sortable with respect to the entropy state at which the atmospheric samples were taken. Thus, comparing repeated measurements of an atmospheric parameter on an isentropic surface or layer excludes any diabatic change of the probed air mass.

Apart from characterising the isentropes, the vertical profiles of the potential temperature ($\theta$ as a function of height $z$) are used as the reference for evaluating the atmosphere's actual vertical temperature gradient, which allows characterising its static stability. Notably, von Bezold (1888) already proposed the potential temperature as an atmospheric stability criterion. In its basic formulation, the potential temperature exclusively refers to the state of dry air, and thus the potential temperature characterises the atmosphere's static stability with respect to vertical displacements of a dry air parcel. In meteorology, the

static stability parameter is expressed in terms of the (squared) Brunt-Väisälä frequency $N$, often written in the form

$$N^2 = \frac{g}{\theta}\frac{\partial \theta}{\partial z},\tag{2}$$

where $g = 9.81\,\mathrm{m\,s^{-2}}$ is the gravitational acceleration. The potential temperature twice enters the formulation of the stability parameter, as the denominator ($\theta^{-1}$) and as the vertical gradient $\frac{\partial \theta}{\partial z}$. In the research field of dynamical meteorology, the potential vorticity (PV) is often used (Ertel, 1942; Hoskins et al., 1985; Schubert et al., 2004). The PV is proportional to the scalar product of the atmosphere's vorticity (the air's local spinning motion) and its stratification (the air's tendency to spread in layers of diminished exchange). More concretely, the PV is the scalar product of the absolute vorticity vector and the three-dimensional gradient of $\theta$, i.e., not only the potential temperature's vertical gradient but also its partial derivatives on the horizontal plane add to the resulting PV, although, particularly at stratospheric altitudes, the vertical gradient constitutes the dominant contribution. For the analytical description of a fluid's motion within a rotational system, as is the atmosphere, the PV provides a quantity that varies exclusively due to diabatic processes. Frequently, the PV is used to define the tropopause height (usually at 2 PV units, see, e.g., Gettelman et al., 2011) or the edge of a large-scale cyclone such as the polar winter vortex on specific $\theta$ levels (cf. Curtius et al., 2005).

While for a dry atmosphere (i.e., with little or no water vapour) the potential temperature is the correct conserved quantity (corresponding to entropy) for reversible processes, for an atmosphere containing water in two or more phases (vapour, liquid, and/or solid phases) energy transfers due to phase changes play a major role. Thus, the formulation of the potential temperature has to be extended since entropy is still the right quantity for reversible processes, including phase changes. Starting from the equation for the moist specific entropy, as derived from the first law of thermodynamics and the Gibbs equation, further extensions of the dry air potential temperature have been developed (Hauf and Höller, 1987; Emanuel, 1994; Marquet, 2011; Marquet and Geleyn, 2015) to account for phase changes and deviations from thermodynamic equilibrium, e.g., by irreversible processes. By assuming only reversible processes (i.e., conserved entropy), approximate formulas can be derived (e.g., Emanuel, 1994). However, in the case of large hydrometeors, liquid or solid particles are removed due to gravitational acceleration, leading to an irreversible process, hence the formulas based on the assumption of a reversible process are no longer applicable. Sometimes for this situation a so-called pseudo adiabatic potential temperature is defined, assuming instantaneous removal of hydrometeors from the air parcel; usually, meaningful approximations to this quantity are given, since generally it cannot be derived from first principles. Equivalent potential temperature including phase changes for vapour and liquid water is often used for the determination of convective instabilities. The general formulation can be easily adapted for an ice equivalent potential temperature, i.e., for reversible processes in pure ice clouds (see, e.g., Spichtinger, 2014). Although the latent heat of sublimation is larger than the latent heat of vaporisation, the absolute mass content of water vapour decreases exponentially with decreasing temperature, leading to only small corrections due to phase changes in pure ice clouds.

At altitudes above the clouds' top, within the upper troposphere and across the tropopause, the air is substantially dried out compared to tropospheric in-cloud conditions. Therefore, above clouds and further aloft, e.g., within the stratosphere, the conventional dry-air potential temperature may suffice to provide a meaningful vertical coordinate. Moreover, the potential temperature or the virtual potential temperature, which includes water vapour, are commonly used as prognostic variables in

numerical models for the formulations of the energy equation (e.g., Skamarock et al., 2005; Skamarock and Klemp, 2008; Zängl et al., 2015; Borchert et al., 2019). Thereby, very often both variants, the potential temperature as well as the equivalent potential temperature, are involved to account for dry air situations and cloud conditions.

In any case, the use of the potential temperature requires the following preconditions to be fulfilled:

1. $\theta$ should be based on a rigorous derivation to ensure its validity as a function of atmospheric altitude in order not to corrupt its character as a vertical coordinate that allows for appropriately comparing (measured) atmospheric parameters, and

2. $\theta$ should approximate to the greatest possible extent the true entropy state of a probed air mass and should preferably account for the implied dependencies on atmospheric variables, even under the assumption that air behaves as an ideal
gas,

with the aim that the potential temperature behaves as a rational physical variable. Thus, still abiding by the ideal gas assumption, a re-assessment of the fundamental atmospheric quantity $\theta$ is suggested, which is based on the state-of-knowledge of air's thermodynamic properties, and this re-assessed $\theta$ is comprehensively examined concerning its ability to hold also for atmospheric conditions above the troposphere.

In principle, the concept of the potential temperature is transferable to all systems of thermally stratified fluids such as a planetary gas atmosphere or an ocean, to investigate heat fluxes (advection or diffusion) or the static stability of the fluid. In astrophysics, the potential temperature is used almost identically as in atmospheric sciences to describe dynamic processes and thermodynamic properties (e.g., static stability or vorticity) in the atmosphere of planets other than the Earth. Here, the same value $p_0 = 1000\,\mathrm{hPa}$, as applied to the Earth's atmosphere, is frequently used as a reference pressure for the atmosphere of
other planets (Catling, 2015, Table 4), whereby the formulations of the specific heat capacity require adaptations to account for the individual gas composition of the respective planetary atmosphere. In order to simulate the weather in the atmosphere of other planets, the Weather Research and Forecasting model (WRF) was extended to "planetWRF" (Richardson et al., 2007) and the governing equations considered within the WRF model include a prognostic equation for the potential temperature (Skamarock et al., 2005; Skamarock and Klemp, 2008). However, the temperature dependency of the isobaric heat capacity $c_p$
is not generally negligible, especially when taking "deep atmospheres, such as on Venus" (Catling, 2015, p. 436) into account or the temperature lapse rates on other planets (Li et al., 2018). The atmosphere of Saturn's moon Titan, the only known moon with a substantial atmosphere, was comprehensively studied with frequent application of the potential temperature based on profile measurement of temperature and pressure in Titan's atmosphere by the Huygens probe (Müller-Wodarg et al., 2014).

Moreover, the potential temperature is a frequently used quantity in oceanography (e.g., McDougall et al., 2003; Feistel,
2008), while here the consideration of sea water's salinity and its impact on the specific heat capacity of sea water implies additional complexity. In particular, McDougall et al. (2003) suggests a re-assessment of the potential temperature as applied in oceanography to approximate the adiabatic lapse rate, thus this study bears certain parallels to the present investigation aiming at the reappraisal of the potential temperature for atmosphere-related purposes. These studies from other disciplines motivate the need for a re-assessment of the potential temperature for the atmospheric sciences. Thus, the approach provided

herein proposes a modified calculation of the widely used quantity of the potential temperature by additionally accounting for the current state of knowledge concerning air's properties.

The study is organised as follows: The derivation of the potential temperature for an ideal gas with constant specific heat capacity $c_p$ is recalled in Section 2. In Section 3 the assumption of a constant $c_p$ is discussed together with a synopsis of various $c_p$ values as provided in the literature. The temperature dependency of $c_p$ is examined in Section 4 and a parameterisation is given. Section 5 is devoted to the definition and computation of a new reference potential temperature $\theta_{\text{ref}}$ based on the temperature-dependent specific heat capacity, while Section 6 focuses on the influence of real-gas effects on the resulting potential temperature. Section 7 presents some implications of the use of $\theta_{\text{ref}}$ and concluding remarks are given in Section 8.

## 2 Derivation of the potential temperature for an ideal gas

The Gibbs equation (see, e.g., Kondepudi and Prigogine, 1998) is a general thermodynamic relation to describe the state of a system with $m$ components and reads as

$$T \, dS = dH - V \, dp - \sum_{k=1}^{m} \mu_k \, dM_k, \tag{3}$$

where $T$ denotes the absolute temperature in K, $S$ the entropy in $\text{J K}^{-1}$, $H$ the enthalpy in J, $V$ the volume in $\text{m}^3$, $\mu_k$ the chemical potential of component $k$ in $\text{J kg}^{-1}$, $M_k$ the mass of component $k$ in kg, and $p$ the static pressure in Pa. Assuming no phase conversion or chemical reaction within the system, the mass of each component does not change, hence $dM_k = 0$ for each component $k$.

In the following, dry air is assumed to be the single component in the system. Expressing the Gibbs equation in its specific form (i.e., division by the total mass $M_a$ of dry air; note, lowercase letters indicate specific variables, e.g., $h = H/M_a$, etc.) leads to

$$T \, ds = dh - \frac{V}{M_a} \, dp \quad \Leftrightarrow \quad ds = \frac{1}{T} \, dh - \frac{V}{M_a T} \, dp. \tag{4}$$

Furthermore, approximating dry air as an ideal gas leads to the following simplifications:

– The ideal gas law

$$pV = M_a R_a T \tag{5}$$

can be applied with the specific gas constant $R_a$ of dry air, which is

$$
\begin{aligned}
R_a &= \frac{R}{M_{\text{mol},a}} \\
&= \frac{8.31446261815324 \, \text{J mol}^{-1}\text{K}^{-1}}{0.0289586 \, \text{kg mol}^{-1} \pm 0.0000002 \, \text{kg mol}^{-1}} \\
&\in \left[ 287.11350 \, \text{J kg}^{-1}\text{K}^{-1}, \, 287.11748 \, \text{J kg}^{-1}\text{K}^{-1} \right],
\end{aligned}
\tag{6}
$$

with the molar gas constant $R$ in $\mathrm{J\,mol^{-1}K^{-1}}$ (Tiesinga et al., 2020; Newell et al., 2018) and $M_{\mathrm{mol},a}$ the molar mass of dry air (Lemmon et al., 2000), composed of nitrogen $N_2$, oxygen $O_2$, and argon $Ar$.

– The specific enthalpy is given by

$$\mathrm{d}h = c_p \,\mathrm{d}T \tag{7}$$

with $c_p$ the specific heat capacity of dry air.

Based on these assumptions, the change of the specific entropy (within the fluid *dry air*) is given by

$$\mathrm{d}s = \frac{c_p}{T}\,\mathrm{d}T - R_a \frac{\mathrm{d}p}{p}. \tag{8}$$

For isentropic changes of state, i.e., $\mathrm{d}s = 0$, equation (8) reduces to

$$\frac{c_p}{T}\,\mathrm{d}T = R_a \frac{\mathrm{d}p}{p}. \tag{9}$$

Note that the assumption of dry air being an ideal gas does not imply that in (9) the specific heat capacity $c_p$ is constant. While statistical mechanics excludes any pressure dependence in the ideal-gas heat capacity, the general derivation (cf. Appendix A) permits a temperature dependence of $c_p$. However, usually the temperature dependence is neglected in atmospheric physics and, instead, $c_p$ is assumed as constant (see, e.g., Ambaum, 2010, page 48/49, where vibrational modes of the air molecules are neglected). Immediately below and in Section 3, the treatment of $c_p$ as a temperature-independent constant is discussed. The introduction of the temperature dependence then follows in Section 4.

Treating $c_p$ as a constant, rearrangement of (9) leads to

$$\frac{\mathrm{d}T}{T} = \frac{R_a}{c_p}\frac{\mathrm{d}p}{p}. \tag{10}$$

Integration of (10) over the range from ground-level pressure and temperature $(p_0, T_0)$ to the pressure and temperature at a specific height $(p, T)$ yields

$$\ln\left(\frac{T}{T_0}\right) = \int_{T_0}^{T} \frac{\mathrm{d}T'}{T'} = \frac{R_a}{c_p} \int_{p_0}^{p} \frac{\mathrm{d}p'}{p'} = \frac{R_a}{c_p} \ln\left(\frac{p}{p_0}\right), \tag{11}$$

and, after another straightforward conversion, one arrives at

$$\ln\left(\frac{T_0}{T}\right) = \frac{R_a}{c_p} \ln\left(\frac{p_0}{p}\right). \tag{12}$$

With the definition $\theta_{c_p} = T_0$, equation (12) is transformed into the commonly used expression for determining the potential temperature

$$\theta_{c_p} = T \left(\frac{p_0}{p}\right)^{\frac{R_a}{c_p}}, \tag{13}$$

for which the ground-level pressure $p_0$ is arbitrary but usually set to $p_0 = 1000\,\text{hPa}$. This choice coincides with the definition of the World Meteorological Organisation (WMO, 1966) and the standard-state pressure (Tiesinga et al., 2020), but should not be confused with the standard atmosphere $101325\,\text{Pa}$ (Tiesinga et al., 2020). In the following, $\theta_{c_p}$ denotes the potential temperature based on a constant $c_p$ and, when a specific value of $c_p$ is applied, the subscript $c_p$ in the potential temperature's notation is replaced by the corresponding $c_p$ value.

## 3   Examining the assumption of constant $c_p$ for dry air

The general theory of thermodynamics, assuming dry air as an ideal gas, gives the expression

$$c_p = \left(1 + \frac{f}{2}\right) R_a \tag{14}$$

for the constant specific heat capacity, which is based on the results of statistical mechanics and the equipartition theorem (e.g., Huang, 1987). In (14), the parameter $f = f_{\text{trans}} + f_{\text{rot}} + f_{\text{vib}}$ is equal to the total number of degrees of freedom of the gas molecules of which dry air consists. The individual contributions to $f$ comprise the degrees of freedom of translation $f_{\text{trans}}$, rotation $f_{\text{rot}}$, and vibration $f_{\text{vib}}$. Assuming further that dry air exclusively consists of the linear molecules $N_2$ and $O_2$ (implying $f_{\text{trans}} = 3$ and $f_{\text{rot}} = 2$, while the contribution of $Ar$ remains disregarded) and additionally neglecting the vibrational degrees of freedom ($f_{\text{vib}} = 0$), the general relation (14) reduces to

$$c_p = \left(1 + \frac{3+2}{2}\right) R_a = \frac{7}{2} R_a. \tag{15}$$

Although the neglect of vibrational excitation, particularly at very low temperatures, seems plausible and appropriate, errors are already introduced by this assumption for the temperature range relevant in the atmosphere.

In atmospheric sciences, for the majority of computations that require the specific heat capacity of dry air, a constant value of $c_p$ may be appropriate. According to the WMO (1966), the recommended value for $c_p$ of dry air is $1005\,\text{J}\,\text{kg}^{-1}\text{K}^{-1}$ and, furthermore (*ibid.*), it is defined that $\gamma = \frac{c_p}{c_v} = \frac{7}{5} = 1.4$, cf. (1). This definition is consistent with the general thermodynamic theory together with all aforementioned additional assumptions and results in (15) as well.

Even assuming a universally valid constant $c_p$, a single consistently used value of $c_p$ was not found. Instead, the specified values of $c_p$ vary among different textbooks and other sources. In Table 1, some of the available values of constant specific heat capacity for dry air are compiled, indicating a variability of $c_p$ that ranges from $994\,\text{J}\,\text{kg}^{-1}\text{K}^{-1}$ to $1011\,\text{J}\,\text{kg}^{-1}\text{K}^{-1}$. However, the extremes in Table 1 are from old references of historical interest only; to reflect recently stated values the narrower range $1000\,\text{J}\,\text{kg}^{-1}\text{K}^{-1}$ to $1010\,\text{J}\,\text{kg}^{-1}\text{K}^{-1}$ is considered.

These different values of constant $c_p$ scatter within a small range (below $\pm 1.1\%$) around the WMO's recommendation $1005\,\text{J}\,\text{kg}^{-1}\text{K}^{-1}$, which may seem negligible if $c_p$ contributes only as a linear coefficient within an equation (e.g., in the expression of a correction factor, cf. Weigel et al., 2016). However, in the formulation of the potential temperature $\theta_{c_p}$, cf. (13), the specific heat capacity $c_p$ does not contribute linearly but rather as the denominator in the exponent. Thus, the variety of different $c_p$ values, although scattering within a small range, impact the resulting $\theta_{c_p}$ significantly. To illustrate this impact, a

| constant dry air's specific heat capacity $c_p$ in $\mathrm{J\,kg^{-1}K^{-1}}$ | literature source |
|---|---|
| 994 | Wegener and Wegener (1935, converted from units other than SI) |
| 1000 | Roedel and Wagner (2011, page 66) |
| 1003 | "minimum of range of actual values" (WMO, 1966) |
| | Tripoli and Cotton (1981, the appendix therein) |
| 1004 | Holton (2004, page 491) |
| | Wallace and Hobbs (2006, page 75) |
| | Schumann (2012) |
| | Wendisch and Brenguier (2013, page 24) |
| | Liou (2002, appendix F) |
| | Ambaum (2010, table "Useful Data") |
| 1004.8 | Pruppacher and Klett (2010, converted from units other than SI; p. 489) |
| 1004.86 | Curry and Webster (1998, page 62) |
| 1005 | recommended by WMO (1966) |
| | Bohren et al. (1998, page 384) |
| | Houghton (2002, page 275) |
| | Zdunkowski and Bott (2003, page 705) |
| | Brasseur and Solomon (2005, page 426) |
| | Seinfeld and Pandis (2006, page 1178) |
| | Cotton et al. (2011, table 2.1) |
| $1005.7 \pm 2.5$ | Bolton (1980) |
| | Emanuel (1994, appendix 2) |
| 1006 | Wendisch and Brenguier (2013, page 69) |
| | Stamnes et al. (2017, page 14) |
| 1010 | Chang et al. (2006) |
| | Tiwary and Williams (2019, beneath eq. 8.8; possibly a typo, as indicated by inconsistencies on reproducing their conclusions based on this value) |
| | Brusseau et al. (2019, page 59) |
| 1011 | "maximum of range of actual values" (WMO, 1966) |

**Table 1.** Synopsis of temperature-independent constant values given mainly in textbooks for the specific heat capacity $c_p$ of dry air from various sources (non-exhaustive). Note, the WMO (1966) indicates a minimum and maximum "range of actual values" together with their recommended value $c_p = 1005\,\mathrm{J\,kg^{-1}K^{-1}}$.

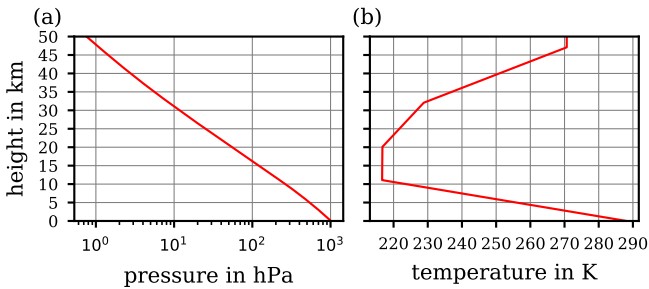

**Figure 1.** Vertical profiles of (a) atmospheric pressure and (b) temperature as functions of height, corresponding to the US Standard Atmosphere.

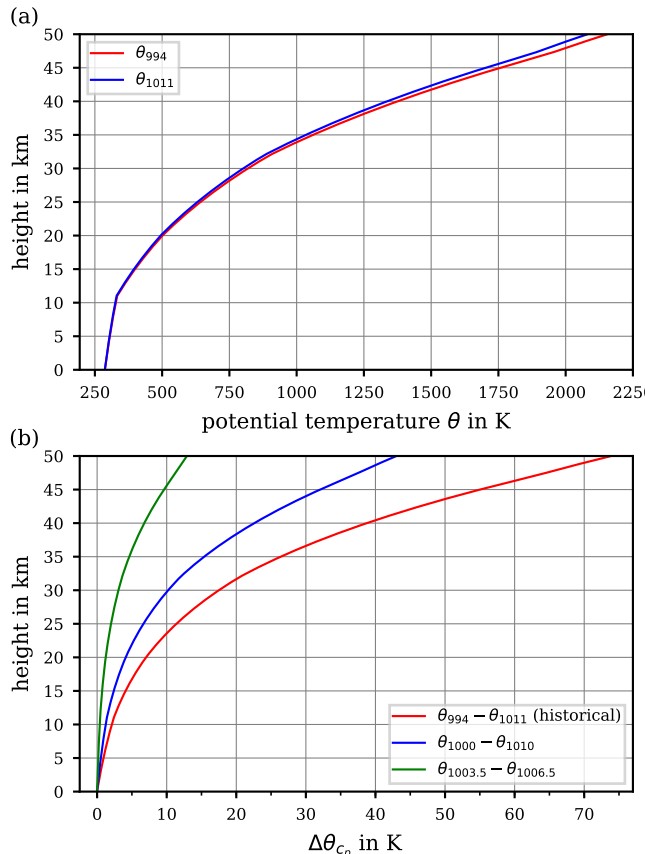

**Figure 2.** Computed vertical course of the potential temperature $\theta_{c_p}$ based on the two extremes of constant values for the specific heat capacity $c_p$ provided in the literature including the historical extreme values (panel (a); cf. Table 1), and (b) the absolute differences $\Delta\theta_{c_p} = \theta_{994} - \theta_{1011}$ and $\Delta\theta_{c_p} = \theta_{1000} - \theta_{1010}$ between the two resulting curves of $\theta_{c_p}$. The absolute difference $\Delta\theta_{c_p} = \theta_{1003.5} - \theta_{1006.5}$ is also shown (green curve), corresponding to a more realistic interval of $c_p$ values.

computation of $\theta_{c_p}$ by using (13) was based on the values of static pressure ($p$, cf. Figure 1a) and absolute temperature ($T$, cf. Figure 1b) corresponding to the US Standard Atmosphere (United States Committee on Extension to the Standard Atmosphere, 1976). From the list of the different $c_p$ in Table 1, the extreme values were selected in order to initially illustrate the sensitivity of the resulting $\theta_{c_p}$ to variations in $c_p$ in the range of $\sim 1\%$, as seen in the literature. In Figure 2a, the individual profiles of

$\theta_{c_p}$ are shown for the extremes of the historic $c_p$ values (Table 1), while Figure 2b illustrates the absolute differences $\Delta\theta_{c_p} = \theta_{994} - \theta_{1011}$ (red curve), $\Delta\theta_{c_p} = \theta_{1000} - \theta_{1010}$ (blue curve), and $\Delta\theta_{c_p} = \theta_{1003.5} - \theta_{1006.5}$ (green curve). The absolute error exhibited with the blue curve in Figure 2b is based on the extremes of most recently referred $c_p$ values in the literature (Table 1). At an altitude of $8.5\,\mathrm{km}$, the difference $\Delta\theta_{c_p}$ already exceeds $1\,\mathrm{K}$ (blue curve). The values of $\Delta\theta_{c_p}$ reach approximately $1.2\,\mathrm{K}$ at $10\,\mathrm{km}$ and rise further, above $4\,\mathrm{K}$, with increasing altitude up to $20\,\mathrm{km}$. At $50\,\mathrm{km}$, approximately where the stratopause

is located, which is the chosen upper height limit for this investigation, the computed $\Delta\theta_{c_p}$ reaches $43\,\mathrm{K}$. The green curve corresponds to the more realistic $c_p$ interval $1005\,\mathrm{J\,kg^{-1}K^{-1}} \pm 1.5\,\mathrm{J\,kg^{-1}K^{-1}}$ as recommended by the WMO; the difference reaches approximately $13\,\mathrm{K}$ at the stratopause.

Figure 2 illustrates the possible spread of $\theta_{c_p}$ based on a range of $c_p$ values from different literature references; hence, if one uses a different value for $c_p$ from the literature than that defined by WMO (1966), the difference $\theta_{1005} - \theta_{c_p}$ might be significant.

Since the $c_p$ values provided by some literature references are close to the value $c_p = 1005\,\mathrm{J\,kg^{-1}K^{-1}}$ recommended by the WMO (1966), the subsequent comparisons will be made to $\theta_{1005}$. The $\theta_{c_p}$ based on $c_p$ values other than $1005\,\mathrm{J\,kg^{-1}K^{-1}}$ are only used to illustrate respective deviations. Although the curves in Figure 2b depict extremes in the deviation of potential temperatures, as they are based on the extremes of $c_p$ values (cf. Table 1), they nevertheless illustrate the sensitive response of $\theta_{c_p}$ to even small variations in $c_p$, on the order of $1\%$. Further proof of this sensitivity from the mathematical perspective is

provided in Appendix B. The impact of this sensitivity becomes important at altitudes of $\sim 10\,\mathrm{km}$ and above, thus, where the use of the potential temperature becomes increasingly meaningful. Here, and in particular above the cloud tops, the small-scale and comparatively fast tropospheric dynamics (causing vertical transport and implying diabatic processes) become diminished, while further above, towards the stratosphere, an increasingly layered vertical structure of the atmosphere is taking over.

As indicated above, the reason for this sensitivity to small variations of air's specific heat capacity is that it affects the

exponent of the equation for $\theta_{c_p}$. The studies of Ooyama (1990, 2001) document an interesting attempt to formulate, e.g., the energy balance equations for the moist atmosphere, wherein entropy replaces the more common formulation using the potential temperature. This substitution avoids the use of the potential temperature, which "is merely an exponential transform of the entropy expressed in units of temperature" (Ooyama, 2001), thus, within this equation, air's specific heat capacity is implied exclusively as a linear coefficient. Consequently, a parameterisation for the temperature dependence of the specific

heat capacity ($c_p(T)$, cf. Section 4) may be easily adopted. However, the crucial drawback of the entropy-based equations is that to gain a numerical model for, e.g., weather forecast purposes, the parameterisations of most of the physical processes within the atmosphere would require a reformulation.

It should be noted that not only do literature values of air's specific heat capacity $c_p$ vary, but also the values of the gas constant $R_a$ vary slightly due to different historical approximations for the molar gas constant[4] $R$ and for the composition of

---

[4]The value of $R$ is now defined exactly, cf. Tiesinga et al. (2020); Newell et al. (2018) and is used in Equation (6).

dry air. The variation of values for $R_a$ is typically only on the order of $0.1\,\mathrm{J\,kg^{-1}K^{-1}}$, whereas the variability in $c_p$ is on the order of a few $\mathrm{J\,kg^{-1}K^{-1}}$ (cf. Table 1). Therefore, within the exponent of the expression (13) for $\theta_{c_p}$, the variability of $c_p$ has by far a stronger impact on the resulting $\theta_{c_p}$ value than the variability of $R_a$.

However, accepting for a moment the WMO's definition (15) of $c_p$ (WMO, 1966), the variability of air's $c_p$ should naturally be constrained to certain limits. With the specific gas constant $R_a = 287.05\,\mathrm{J\,kg^{-1}K^{-1}}$ (WMO, 1966), the WMO's definition leads to $c_p = 1004.675\,\mathrm{J\,kg^{-1}K^{-1}}$. In contrast, taking into account the uncertainty introduced in $R_a$ by the molar mass of dry air, cf. Equation (6), the resulting range for air's specific heat capacity is $1004.897\,\mathrm{J\,kg^{-1}K^{-1}} \leq c_p \leq 1004.912\,\mathrm{J\,kg^{-1}K^{-1}}$. It may be surmised that the rounded value $c_p = 1005\,\mathrm{J\,kg^{-1}K^{-1}}$ as recommended by the WMO (1966) had the main goal to simplify certain calculations, which at the time may have been mostly done by hand.

## 4 Accounting for the temperature dependence of air's specific heat capacity

Next, while retaining the ideal-gas assumption, we consider the dependence of air's $c_p$ on temperature, mainly over the atmospherically relevant range ($180\,\mathrm{K}$ to $300\,\mathrm{K}$). The temperature dependence of $c_p$ is, of course, not a new finding. Experimental approaches for determining the calorimetric properties of air and the temperature dependence of a fluid's specific heat capacity are described by Witkowski (1896), who investigated the change of the mean $c_p$ as a function of temperature intervals between room temperature (as a fixed reference) and various warmer and colder temperatures, for atmospheric pressures and slightly beyond. Despite the potentially high uncertainty of the experimental results from these times, Witkowski (1896) already indicated that with decreasing temperature the experimentally determined $c_p$ values initially decline, then pass a minimum, and subsequently increase again at lower temperatures ($T < 170\,\mathrm{K}$). The description of refined experiments and ascertainable data of air's $c_p(T)$ for temperatures below $293\,\mathrm{K}$ is summarised by Scheel and Heuse (1912), Jakob (1923), and Roebuck (1925, 1930), illustrating in comprehensive detail the experimental effort and providing the resulting data. The review by Awano (1936) compiled and compared the data of $c_p(T)$ of dry air ("*air containing neither carbon-dioxide nor steam*", Awano, 1936) and he attested—at that time—the previously mentioned studies to constitute "*the most reliable experiments*". During the decades following these experiments, further insights were gained and landmarks were reached which are summarised in the comprehensive survey by Lemmon et al. (2000) of the progress of modern formulations for the thermodynamic properties of air and about the experiments the previous formulations were based on.

Figure 3 illustrates the range of suggested constant values for the specific heat capacity as indicated in Table 1 (dashed curves) together with the measurements that were made to obtain air's behaviour as a function of temperature and pressure. Note, Figure 3 includes data at other atmospheric pressures, indicated by squares, diamonds, and triangles. In the same figure, calculated values of $c_p(T)$ of dry air are displayed resulting from the equation of state which was derived from experimental $p$, $V$, and $T$ data by Vasserman et al. (1966), who provided an extensive review of previous experimental and theoretical works and of the state of knowledge at that time. In addition, Figure 3 exhibits two different parameterisations, by Lemmon et al. (2000) and by Dixon (2007, see page 376 in his book, the accuracy is "within $0.1\,\%$ from $200\,\mathrm{K}$ to $450\,\mathrm{K}$"), which account for the temperature dependence of the specific heat capacity $c_p(T)$. The parameterisation by Lemmon et al. (2000), to be discussed

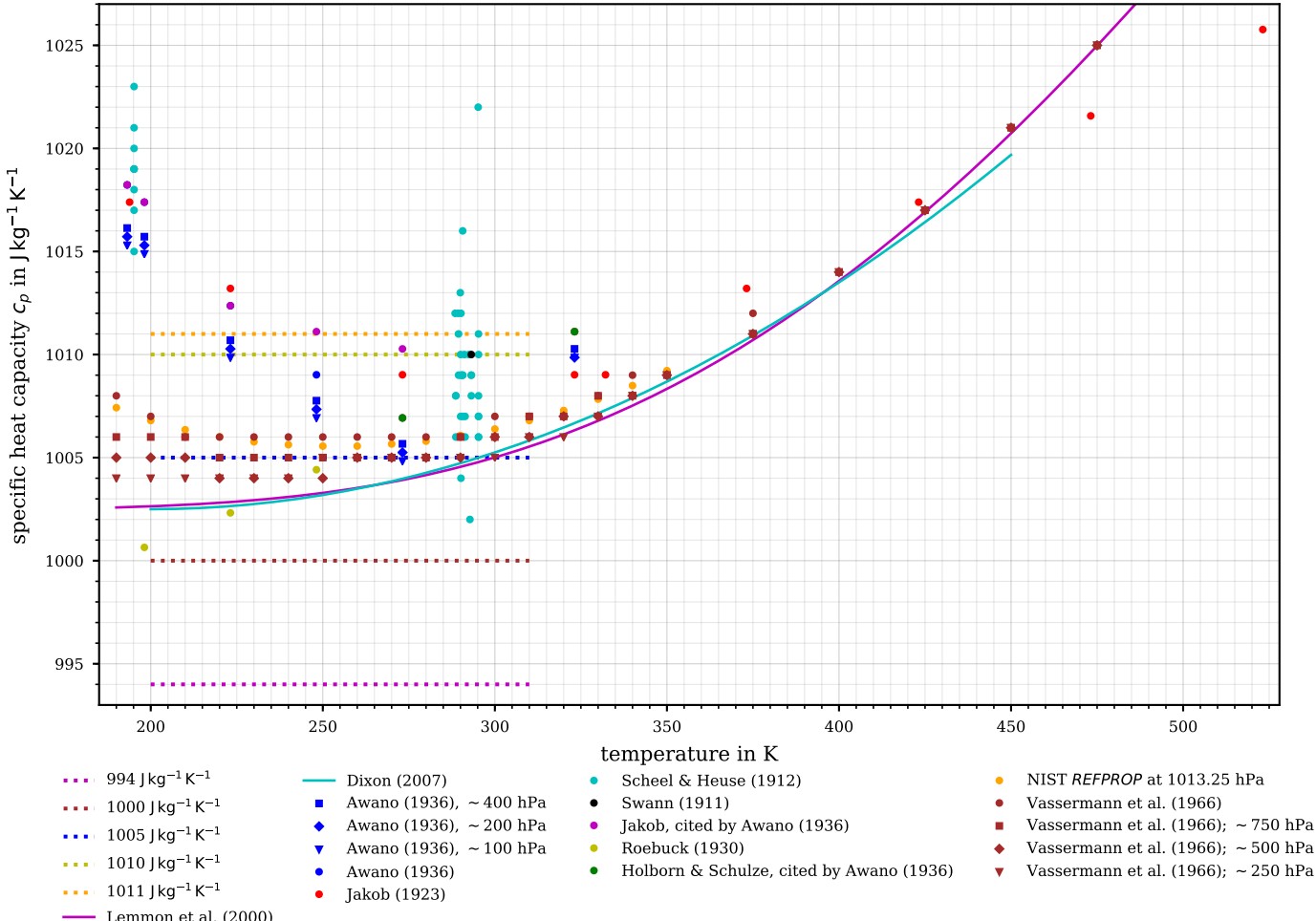

**Figure 3.** Variety of suggested values for the specific heat capacity of air. Ranges of constant values for $c_p$ (including the historical) together with the recommended value by the WMO (1966) are displayed as given in Table 1 (dashed lines). The parameterisations of air's $c_p^0(T)$, assuming dry air as an ideal gas, accounting for its temperature dependence by Lemmon et al. (2000, solid magenta curve) and by Dixon (2007, solid cyan curve) are displayed. Discrete measurement and literature data at about $1000 \, \text{hPa}$ (i.e., as often specified, at *about one atmosphere*) are indicated by dots. In addition, the studies by Awano (1936) and Vasserman et al. (1966) provide data at other atmospheric pressures, as indicated by squares, diamonds, and triangles.

in detail in Section 4.2, is valid for dry air assumed as an ideal gas whereas this distinction is not explicitly made in Dixon (2007). Moreover, Figure 3 contains discrete values of dry air's $c_p(T)$ extracted from the database *REFPROP* (Reference Fluid Thermodynamic and Transport Properties Database by NIST, the National Institute of Standards and Technology, Lemmon et al., 2018), which is based on parameterisations resulting from thermodynamic considerations discussed later.

The measurement data, as well as the parameterisations, clearly indicate a dependence of air's specific heat capacity on the temperature. At temperatures above $300\,\mathrm{K}$, the data points by Jakob (1923) are surprisingly well captured by the parameterisations, while below $270\,\mathrm{K}$ the course of the parameterised and measured $c_p(T)$ diverge significantly. Possible reasons for this include:

- the measurements of $c_p(T)$ have a precision likely no better than $1\,\%$ (in particular the historical measurements), and there could be systematic errors, especially at low temperatures;

- the measured data reflect the true thermodynamic behaviour of the real gas, rather than that of an ideal gas.

However, it is immediately obvious from Figure 3 that a good agreement among (i) the experimentally determined $c_p(T)$ data, (ii) a constant $c_p$ (e.g., $1005\,\mathrm{J\,kg^{-1}K^{-1}}$; WMO (1966)), and (iii) the parameterised $c_p(T)$ is found only for a temperature interval ranging from $270\,\mathrm{K}$ to $300\,\mathrm{K}$. For air temperatures below $270\,\mathrm{K}$, the constant value $c_p = 1005\,\mathrm{J\,kg^{-1}K^{-1}}$ is only comparable with the values from Vasserman et al. (1966), but fails to coincide with other parameterised or experimentally determined values of $c_p(T)$.

## 4.1 The temperature dependence of the ideal-gas specific heat capacity

As already indicated by the data depicted in Figure 3, the specific heat capacity $c_p$ depends on the gas temperature. With regard to measured values, the lack of constancy may be due to real-gas effects or to a dependence of the ideal-gas heat capacity on temperature. In this section, we focus on the latter effect, denoting the ideal-gas isobaric specific heat capacity by $c_p^0(T)$, where the superscript 0 indicates the underlying ideal-gas assumption. For an individual gas, there is always a contribution from the three translational degrees of freedom, $c_{p,\mathrm{trans}}^0 = \frac{5}{2}R_i$, where $R_i$ is the specific gas constant of the gas. If the molecule is assumed to be a rigid rotor, there is also a rotational contribution given by

$$c_{p,\mathrm{rot}}^0 = \begin{cases} R_i, & \text{for linear (e.g., diatomic) molecules,} \\ \frac{3}{2}R_i, & \text{for nonlinear molecules.} \end{cases} \tag{16}$$

As mentioned previously, at finite temperatures molecules also have contributions to $c_p^0(T)$ from intramolecular vibrations (and, at high temperatures, excited electronic states). To arrive at a temperature-dependent parameterisation for the ideal-gas specific heat capacity of dry air, the compounds' individual contributions, considering all degrees of freedom, need to be parameterised and then combined according to each compound's proportion in the mixture. For the following, dry air is considered a three-component mixture: the diatomic gases nitrogen ($N_2$) and oxygen ($O_2$) and the monatomic gas argon (Ar).

To determine the contribution of $N_2$ to $c_p^0(T)$, both Bücker et al. (2002) and Lemmon et al. (2000) use the ideal-gas heat capacity from the reference equation of state of Span et al. (2000) that compares well with the findings from other studies within an uncertainty $\Delta c_p^0$ of less than $0.02\%$.

For the contribution of $O_2$, Lemmon et al. (2000) use the formulation given by Schmidt and Wagner (1985). Alternatively, Bücker et al. (2002) provide a slightly different formulation from the International Union of Pure and Applied Chemistry (IUPAC, Wagner and de Reuck, 1987), after refitting it to more recently obtained data, thereby achieving an overall uncertainty $\Delta c_p^0$ of less than $\pm 0.015\%$ for $O_2$ (Bücker et al., 2002). However, the difference in the resulting specific heat capacity contribution by $O_2$ between the two approaches (Lemmon et al. (2000) or Bücker et al. (2002)) is comparatively small. The recent work of Furtenbacher et al. (2019) leads to values of $c_p^0$ for $O_2$ with even smaller uncertainties, but the differences from the values used here are negligible in our context.

For a monoatomic gas such as $Ar$, vibrational and rotational contributions to the heat capacity do not exist, and Bücker et al. (2002) consider that argon's excited electronic states are relevant only at temperatures above $3500\,K$. Hence, the contribution of $Ar$ to the specific heat capacity of air reduces to $c_p^0 = \frac{5}{2}R_{Ar}$.

The approach by Bücker et al. (2002) additionally considers the contribution of further constituents of air, such as water, carbon monoxide, carbon dioxide, and sulfur dioxide. These authors provide an analytical expression for specific heat capacity, accounting for this more complex but proportionally invariant air composition which is specified to deviate from the used reference by $\Delta c_p^0 \leq \pm 0.015\%$ in the temperature range of $200\,K \leq T \leq 3300\,K$. At atmospheric altitudes above the clouds' top, i.e., on average above $\sim 11\,km$, the air is assumed to have lost most of its water and is deemed as dry. Furthermore, for the following, trace gases that contribute to air's composition by molar fractions of less than that of $Ar$ are neglected.

## 4.2  NIST's parameterisation of $c_p^0(T)$

Besides a comprehensive survey of the available experimental data for the specific heat capacity of air, Lemmon et al. (2000) also provide state-of-the-art knowledge for other thermodynamic properties (isochoric heat capacity, speed of sound, vapour-liquid-equilibrium, etc.). Additionally, they give two approaches to derive air's thermodynamic properties, including the vapour-liquid equilibrium:

1. an empirical model-based *equation of state* for standard (dry) air considered as a pseudo-pure fluid, and

2. assembly of a *mixture model* from equations of state for each pure fluid.

Each approach allows calculating the thermodynamic properties, e.g., $c_p$, of gas mixtures such as dry air, and both are real-gas models with the ideal-gas behaviour as a boundary condition. The major difference between the models is that the first approach considers air as a pseudo-pure fluid while the second, more rigorous approach treats air as a mixture composed of $N_2$, $O_2$, and $Ar$, in molar fractions of 0.7812, 0.2096, and 0.0092, respectively, following Lemmon et al. (2000, their table 3). This fractional composition of dry air is assumed to be constant from ground level up to $80\,km$ height (United States Committee on Extension to the Standard Atmosphere, 1976) and its fractional composition would have to be shifted significantly to cause a serious deviation of the resulting potential temperature. The contribution to the composition by carbon dioxide ($CO_2$) and of

any other trace species was assumed to be negligible. The validity of both approaches is specified for various states of dry air, from its solidification point (59.75 K) up to temperatures of 1000 K, and for pressures up to 100 MPa and even much further beyond the pressure range that is relevant for atmospheric investigations. Both the pseudo-pure fluid model and the mixture model are implemented in NIST's *REFPROP* database (cf. https://www.nist.gov/srd/refprop) for various physical properties of fluids over a wide range of temperatures and pressures.

Both the pseudo-pure fluid model and the mixture model of Lemmon et al. (2000) use the same expression for the ideal-gas heat capacity, which is rigorously given as a sum of the pure-component contributions:

$$
\frac{C_p^0(T)}{R} = x_{\mathrm{N_2}} \left( \frac{C_p^0(T)}{R} \right)_{\mathrm{N_2}} + x_{\mathrm{Ar}} \left( \frac{C_p^0(T)}{R} \right)_{\mathrm{Ar}}
$$
$$
+ x_{\mathrm{O_2}} \left( \frac{C_p^0(T)}{R} \right)_{\mathrm{O_2}},
$$

(17)

where $x_i$ denotes the molar fraction of species $i$, and $C_p^0$ as well as the molar gas constant $R$ are given in units of $\mathrm{J\,mol^{-1}K^{-1}}$.

Like Bücker et al. (2002), Lemmon et al. (2000) use the expression of Span et al. (2000) for the contribution of $\mathrm{N_2}$ to the heat capacity and adopt $C_p^0 = \frac{5}{2}R$ for Ar. Together with the contribution by $\mathrm{O_2}$ according to the formulation by Schmidt and Wagner (1985), the expression provided by Lemmon et al. (2000, equation 18 therein) for the ideal-gas heat capacity of dry air is

$$
\frac{C_p^0(T)}{R} = N_1 + N_2 T + N_3 T^2 + N_4 T^3 + N_5 T^{-\frac{3}{2}}
$$
$$
+ N_6 \frac{\frac{N_9^2}{T^2} \exp\left(\frac{N_9}{T}\right)}{\left(\exp\left(\frac{N_9}{T}\right) - 1\right)^2} + N_7 \frac{\frac{N_{10}^2}{T^2} \exp\left(\frac{N_{10}}{T}\right)}{\left(\exp\left(\frac{N_{10}}{T}\right) - 1\right)^2}
$$
$$
+ \frac{2N_8}{3} \frac{\frac{N_{11}^2}{T^2} \exp\left(-\frac{N_{11}}{T}\right)}{\left(\frac{2}{3}\exp\left(-\frac{N_{11}}{T}\right) + 1\right)^2},
$$

(18)

with the scalar coefficients $N_i$ for dry air (*ibid.*),

$$
\begin{array}{ll}
N_1 = 3.490888032, & N_2 = 2.395525583 \cdot 10^{-6}, \\
N_3 = 7.172111248 \cdot 10^{-9}, & N_4 = -3.115413101 \cdot 10^{-13}, \\
N_5 = 0.223806688, & N_6 = 0.791309509, \\
N_7 = 0.212236768, & N_8 = 0.197938904, \\
N_9 = 3364.011, & N_{10} = 2242.45, \\
N_{11} = 11580.4, &
\end{array}
$$

(19)

which is specified as valid for temperatures from 60 K to 2000 K. Because the underlying calculations are based on rigorous statistical mechanics and accurate spectroscopic data, $\frac{C_p^0(T)}{R}$ should be accurate to within 0.01 % throughout this range, as discussed by Span et al. (2000).

The parameterisation (18) provides the isobaric specific heat capacity of dry air, considered as a mixture of ideal gases. This represents a more rigorous and accurate behaviour than assuming it to be a constant.

## 4.3 The parameterisation of $c_p^0(T)$ from an engineer's perspective

The parameterisation from Dixon (2007)

$$c_p(T) = 1002.5 + 275 \cdot 10^{-6} \cdot (T - 200)^2 \tag{20}$$

for $200\,\mathrm{K} \leq T \leq 450\,\mathrm{K}$ is not explicitly described to be based on particular assumptions or data sets. The author indicates his
suggested parameterisation to hold within $0.1\,\%$ for temperatures between $200\,\mathrm{K}$ and $450\,\mathrm{K}$. For elevated air temperatures, the deviation between the ideal-gas limit $c_p^0(T)$ (Lemmon et al., 2000) and Dixon's parameterisation substantially increases. This is most likely due to the chosen type of polynomial approximation (Dixon, 2007), which increasingly departs from the reference $c_p^0(T)$ for gas temperatures exceeding $450\,\mathrm{K}$.

Concerning the thermophysical properties of humid air, the study by Tsilingiris (2008) provides further insight. Its purpose was to evaluate the transport properties as a function of different levels of the relative humidity and as a function of temperature (from $273\,\mathrm{K}$ to $373\,\mathrm{K}$) for the gas mixture of air with water vapour at a constant pressure ($1013\,\mathrm{hPa}$). The atmospherically relevant pressure range below $1013\,\mathrm{hPa}$ and temperatures smaller than $273\,\mathrm{K}$ were not considered. Although this study focused on providing a comprehensive account of moisture within air, mainly for technical purposes and engineering calculations, the possible usefulness of these findings to atmospheric investigations is also apparent. However, the impact of water vapour on the resulting gas mixture's $c_p(T)$ is significantly larger (cf. Tsilingiris, 2008) than the uncertainty of dry air's $c_p(T)$ that is discussed in the present work. Furthermore, the consideration of water vapour as a component of air requires very individual and case-specific computations of $c_p(T)$ of moist air, as water vapour is among the most variable constituents of the atmosphere.

The effort required to produce an analytical formulation for gas properties which best reflects the true gas behaviour may indicate that for engineering purposes (pneumatic shock absorbers, engines' combustion efficiency, improvements of turbofan/-prop propulsion, aerodynamics, material sciences, etc.), especially where pressures exceed atmospheric, the assumption of ideal-gas behaviour introduces excessive uncertainty.

## 5 The $\theta_{c_p(T)}$ from the temperature-dependent specific heat capacity of air

Previously introduced approaches for computing the specific heat capacity of dry air call for a brief discussion on how to use the obtained $c_p(T)$ to derive the potential temperature. In the following, $\theta_{c_p(T)}$ denotes the derived potential temperature that accounts for the temperature dependence of dry air's specific heat capacity. Furthermore, it should be noted that simply substituting any $c_p(T)$ value into the conventionally used and defining equation (13) for $\theta_{c_p}$ (WMO, 1966) may appear tempting but definitely leads to results inconsistent with $\theta_{c_p(T)}$ that is based on the reference parameterisation of dry air's $c_p(T)$. Therefore, the thermodynamically consistent use of $c_p(T)$ in the derivation of $\theta$ is described in the following.

### 5.1 Derivation of $\theta_{c_p(T)}$ based on the temperature-dependent specific heat capacity of dry air

In the derivation of the potential temperature (cf. Section 2), we note that, until reaching the expression for isentropic changes of state (9), no assumption was made about the specific heat capacity. As soon as the temperature dependence of the specific

heat capacity comes into play, the re-assessment of (9) leads to

$$\frac{c_p(T)}{T}\,\mathrm{d}T = R_a\frac{\mathrm{d}p}{p}. \tag{21}$$

Integration of (21) from the basic state $(p_0, \theta_{c_p(T)})$ to any other state $(p, T)$ yields

$$R_a \ln\left(\frac{p}{p_0}\right) = R_a \int_{p_0}^{p} \frac{\mathrm{d}p'}{p'} = \int_{\theta_{c_p(T)}}^{T} \frac{c_p(T')}{T'}\,\mathrm{d}T', \tag{22}$$

where $\theta_{c_p(T)}$ is the desired potential temperature.

The rearrangement of (22) makes evident that the desired potential temperature is a zero of the function $F(x)$, given by

$$F(x) = \int_{x}^{T} \frac{c_p(T')}{T'}\,\mathrm{d}T' - R_a \ln\left(\frac{p}{p_0}\right). \tag{23}$$

To arrive at the desired potential temperature $\theta_{c_p(T)}$ for any given temperature and pressure, the equation $0 = F(x)$ must be
solved for the variable $x$, which is the desired $\theta_{c_p(T)}$. Equation (23) has at most only one real zero, since its integrand is strictly positive which means $F(x)$ is strictly monotonic.

In the following, the ideal-gas reference potential temperature $\theta_{\mathrm{ref}}$ is introduced, based on the formulation of the ideal-gas limit of dry air's specific heat capacity $c_p^0(T)$ in accordance with (18) as formulated by Lemmon et al. (2000). This reference potential temperature $\theta_{\mathrm{ref}}$ represents the zero of $F(x)$ in (23), wherein $c_p(T')$ is to be replaced by $c_p^0(T')$, i.e. for given $p, T$
the reference potential temperature $\theta_{\mathrm{ref}}$ solves the equation

$$0 = F(\theta_{\mathrm{ref}}) = \int_{\theta_{\mathrm{ref}}}^{T} \frac{c_p(T')}{T'}\,\mathrm{d}T' - R_a \ln\left(\frac{p}{p_0}\right). \tag{24}$$

The parameterisation of $c_p^0(T')$ is stated to give accurate values for temperatures from $60\,\mathrm{K}$ to $2000\,\mathrm{K}$ (cf. Section 4.2), thus values of $\theta_{\mathrm{ref}}$ should not exceed $2000\,\mathrm{K}$, since otherwise $c_p^0(T')$ within the integrand in (23) is evaluated outside of its range of validity. However, due to the division by $T'$, the value of the integrand $\frac{c_p^0(T')}{T'}$ may be expected to give nevertheless a good
approximation even if the accuracy of $c_p^0(T')$ is decreased, hence values $\theta_{\mathrm{ref}} > 2000\,\mathrm{K}$ should not be discarded.

It may be noted that further variants of a reference potential temperature are derivable by replacing $c_p(T')$ in (23) by any other expression of the specific heat capacity of air which may appear sufficiently accurate. The steps to compute or approximate the zero of the function (23), described in this study, are independent of the chosen heat capacity formulation.

Unfortunately, for a straightforward solution of the integral (23), the suggested parameterisation of $c_p$ is too complex and
405 an analytically insolvable nonlinear equation $0 = F(x)$ could result. Thus, an approximation of the equation's desired zero is required. Newton's method (cf., e.g., Deuflhard, 2011) provides a standard approach to numerically approximate the zero of a nonlinear equation. Proceeding from an initial guess $x_0$, Newton's method constructs a sequence $\{x_k\}_{k\in\mathbb{N}}$ defined by the

recursion

$$x_{k+1} = x_k - \frac{F(x_k)}{F'(x_k)} = x_k - \frac{F(x_k)}{-\frac{c_p(x_k)}{x_k}}$$

$$= \frac{x_k}{c_p(x_k)} \left[ c_p(x_k) + F(x_k) \right] \tag{25}$$

$$= \frac{x_k}{c_p(x_k)} \left[ c_p(x_k) - R_a \ln\left(\frac{p}{p_0}\right) + \int_{x_k}^{T} \frac{c_p(T')}{T'} \, \mathrm{d}T' \right].$$

The constructed sequence $\{x_k\}_{k\in\mathbb{N}}$ converges to the equation's desired zero. For the computations described here, the iteration is stopped as soon as the absolute difference $|x_{k+1} - x_k|$ of two consecutive iterations falls below $10^{-8}\,\mathrm{K}$.

   For the reference of air's specific heat capacity, $c_p^0(T)$, the integral (23) turns out not to be explicitly solvable. Therefore, with each iteration, the solution of the integral $\int_{x_k}^{T} \frac{c_p^0(T')}{T'} \, \mathrm{d}T'$ is approximated by subdividing the entire integration range, $[x_k, T]$, into intermediate intervals with respective size of at most $0.1\,\mathrm{K}$, and by applying Simpson's rule on each subinterval.

As a first guess $x_0$ for the Newton iteration, the conventional definition of $\theta_{c_p}$ based on a constant specific heat capacity (WMO, 1966) is inserted:

$$x_0 = T \left(\frac{p_0}{p}\right)^{\frac{R_a}{1005\,\mathrm{J\,kg^{-1}K^{-1}}}} = \theta_{1005}. \tag{26}$$

In the course of Newton's method, the sequence $\{x_k\}_{k\in\mathbb{N}}$ will converge to the unique zero for any initial guess $x_0$ due to the monotonicity of $F(x)$. However, the right choice of the initial guess $x_0$ substantially decreases the error of the first iteration $x_1$,

speeding up convergence to the desired zero of the function $F(x)$. Therefore, it seems wise to use the conventional definition of $\theta_{c_p}$ as the first guess for the Newton iteration (25).

   Solving the previously described root-finding problem by Newton's method over the comprehensive range of iteration steps (until the set requirement, i.e., $|x_{k+1} - x_k| < 10^{-8}\,\mathrm{K}$, is fulfilled) finally leads to the reference potential temperature $\theta_{\mathrm{ref}}$. This $\theta_{\mathrm{ref}}$ is based on the ideal-gas limit of dry air's specific heat capacity $c_p^0(T)$, which refers to the current thermodynamic state-

425 of-knowledge and, thus, we use $\theta_{\mathrm{ref}}$ as our reference for the potential temperature in the following. For evaluating the results, the air temperature and pressure from the US Standard Atmosphere are used once more to set up the vertical profiles of the potential temperature. Figure 4a exhibits the resulting reference profile, i.e., $\theta_{\mathrm{ref}}$ (red curve). Additionally, for comparison with the reference, further potential temperature profiles $\theta_{c_p}$ are shown based on the two (historical) extremes $c_p = 994\,\mathrm{J\,kg^{-1}K^{-1}}$ and $c_p = 1011\,\mathrm{J\,kg^{-1}K^{-1}}$ (dashed curves), and based on the range limits of more recent values $c_p = 1000\,\mathrm{J\,kg^{-1}K^{-1}}$ and

430 $c_p = 1010\,\mathrm{J\,kg^{-1}K^{-1}}$ (solid green and magenta curves) of given constant values of air's specific heat capacity (cf. Table 1). Clearly, in particular at elevated altitudes, the courses of $\theta_{1000}$ and $\theta_{1010}$ significantly deviate from the reference. To quantitatively evaluate the match between the different profiles, the relative difference of the profiles based on a constant $c_p$, with respect to the reference, i.e., $\Delta\theta/\theta_{\mathrm{ref}} = \left(\theta_{c_p} - \theta_{\mathrm{ref}}\right)/\theta_{\mathrm{ref}}$, is depicted in Figure 4b. The comparison demonstrates that the $\theta_{c_p}$ profiles significantly depart from the reference by about $\sim 300\,\mathrm{K}$ at $50\,\mathrm{km}$ altitude, corresponding to a relative difference of about $16\,\%$.

With both extremes of the recent constant values $c_p \in \left\{1000\,\mathrm{J\,kg^{-1}K^{-1}}, 1010\,\mathrm{J\,kg^{-1}K^{-1}}\right\}$, the relative error level of $0.1\,\%$ is

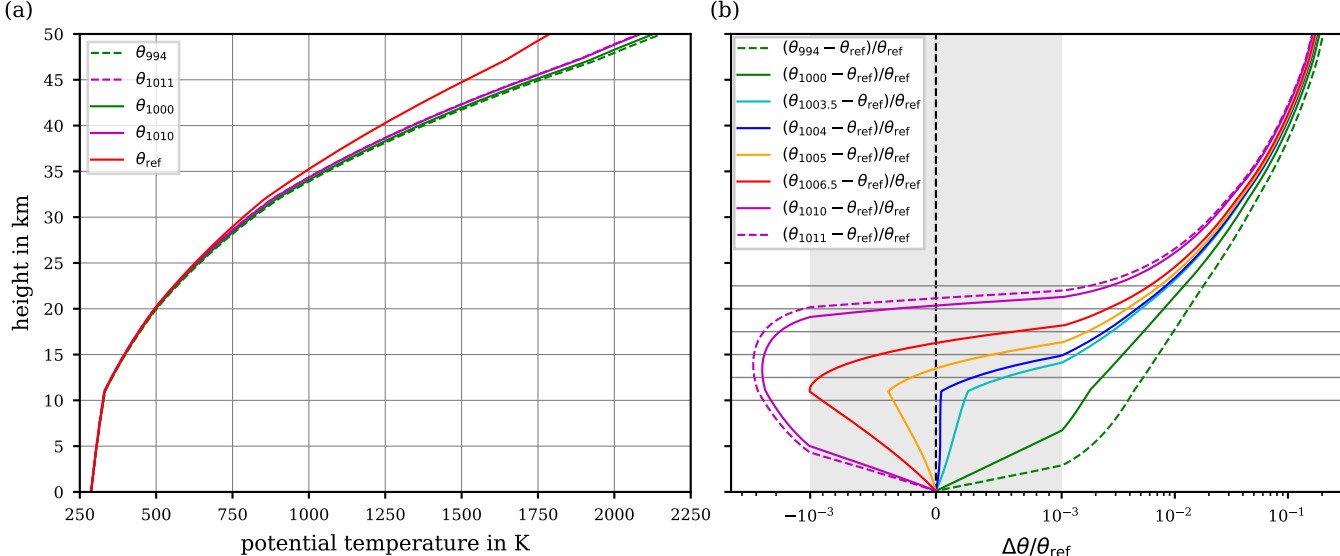

**Figure 4.** (a) Reference potential temperature $\theta_{\text{ref}}$ together with the potential temperatures $\theta_{994}, \theta_{1000}, \theta_{1010}$ and $\theta_{1011}$ relying on constant $c_p$ values (the dashed lines depict the historical extremes for $c_p$, cf. Table 1). (b) Relative differences $\left(\theta_{c_p} - \theta_{\text{ref}}\right)/\theta_{\text{ref}}$ for the same choices $c_p \in \left\{994\,\text{J}\,\text{kg}^{-1}\text{K}^{-1}, 1000\,\text{J}\,\text{kg}^{-1}\text{K}^{-1}, 1010\,\text{J}\,\text{kg}^{-1}\text{K}^{-1}, 1011\,\text{J}\,\text{kg}^{-1}\text{K}^{-1}\right\}$ as in the left panel between the reference potential temperature and the potential temperatures relying on constant $c_p$ values. For comparison, the relative difference $\left(\theta_{1005} - \theta_{\text{ref}}\right)/\theta_{\text{ref}}$ is displayed, for which $c_p = 1005\,\text{J}\,\text{kg}^{-1}\text{K}^{-1}$ corresponds to the WMO recommendation. In addition, also comparisons with $\theta_{1003.5}, \theta_{1004}, \theta_{1006.5}$ are included. All profiles are based on the values for temperature and pressure according to the US Standard Atmosphere. Note the linear axis-scaling inside and the logarithmic scaling outside of the grey-shaded area in panel (b).

exceeded at altitudes about $5\,\text{km}$. While $\theta_{1000}$ continues to increasingly deviate from the reference, $\theta_{1010}$ re-enters and crosses the $0.1\,\%$ relative error interval (grey-shaded area) at altitudes between $\sim 19\,\text{km}$ and $21\,\text{km}$, before it reaches similar errors to the other $\theta_{c_p}$ profiles that are based on a constant $c_p$. Although the extreme values $c_p \in \left\{1000\,\text{J}\,\text{kg}^{-1}\text{K}^{-1}, 1010\,\text{J}\,\text{kg}^{-1}\text{K}^{-1}\right\}$ appear in recent literature, these values may be considered unrealistic. For this reason, Figure 4b also shows the relative de-
viations for the values $c_p \in \left\{1003.5\,\text{J}\,\text{kg}^{-1}\text{K}^{-1}, 1004\,\text{J}\,\text{kg}^{-1}\text{K}^{-1}, 1005\,\text{J}\,\text{kg}^{-1}\text{K}^{-1}, 1006.5\,\text{J}\,\text{kg}^{-1}\text{K}^{-1}\right\}$, which include the recommended value of the WMO (1966) and a more realistic range, i.e. $c_p = 1005\,\text{J}\,\text{kg}^{-1}\text{K}^{-1} \pm 1.5\,\text{J}\,\text{kg}^{-1}\text{K}^{-1}$. Notably, up to an altitude of $15\,\text{km}$, the reference potential temperature is comparably well matched by both the recommended $\theta_{1005}$ and $\theta_{1004}$ (based on the frequently used alternative $c_p = 1004\,\text{J}\,\text{kg}^{-1}\text{K}^{-1}$, cf. Table 1). Until $15\,\text{km}$ altitude, both constant $c_p$ values lead to errors of calculated $\theta_{c_p}$ which remain comparatively small within the $0.1\,\%$ relative error interval. However, above
$\sim 17.5\,\text{km}$, both $\theta_{1004}$ and $\theta_{1005}$ exceed the $0.1\,\%$ relative error interval, and further aloft their relative error with respect to the reference $\theta_{\text{ref}}$ increases rapidly.

In the context of numerical models of the atmosphere, the energy balance equation is occasionally formulated based on the potential temperature $\theta$, thus $\theta$ constitutes a prognostic model variable. In such a case, the temperature $T$ needs to be calculated from a given pair of values of pressure $p$ and potential temperature $\theta$. Using once more the defining equation (22), for given $\theta$

a zero of the function

$$0 = -\int\limits_{T}^{\theta} \frac{c_p(T')}{T'} \, dT' - R_a \ln\left(\frac{p}{p_0}\right) \tag{27}$$

is to be computed. Since (27) corresponds to the function $F$ defined in (23) with the exception of a negative sign, the identical approximation procedure as outlined above in this section for the calculation of $(T, p) \mapsto \theta$ may be applied *mutatis mutandis* to calculate the transformation $(\theta, p) \mapsto T$.

In any case, a certain effort is required to implement the new formulation of the potential temperature in an atmospheric model, as this equation should be based on the implicit definition (22), and such a goal may be the subject of future endeavours.

## 5.2   Approximations of the reference potential temperature

Of course, the previously described procedure to compute the potential temperature may appear to be anything but practical. Indeed, due to the complications inherent with:

– the requirement to numerically solve the integral in the function $F(x)$ and

     – the need to use Newton's method for an iteration sequence to approach the zero of $F(x)$,

a convenient approach to re-assess the conventional definition of the potential temperature is not provided at all. This motivates the development of a more practical approximation of the reference potential temperature. To arrive at a practicable approximation procedure, the two principal steps in the suggested procedure are briefly outlined in the following, whereas the 465   comprehensive details and intermediate derivation steps are found in Appendix C.

     Proceeding from the definition (23) of the function $F(x)$, the computation of the integral $\int\limits_{x}^{T} \frac{c_p^0(T')}{T'} \, dT'$ becomes the first obstacle to a practical approximation. Therefore, a plausible initial step is to replace the integral by an expression that is easier to treat. This expression may be proposed as $f(T) - f(x)$, where the function $f$ is defined as $f(x) = b_0 + b_1 \ln(x - b_2) + b_3 x + b_4 x^2$ and which is recognisable as an approximated primitive of $\frac{c_p^0(T')}{T'}$, see Appendix C1. The choice of the functional form of $f$ is 470   motivated by the exact primitive of the integral in the case of a constant $c_p$.

     As previously discussed (cf. Section 5.1), the formulation of a new expression for the potential temperature based on the temperature-dependent specific heat capacity $c_p(T)$ requires finding the zero of the equation $0 = F(x)$, where the function $F(x)$ is defined in (23). Replacing the exact integral in (23) by the difference $f(T) - f(x)$ means that $F(x)$ is substituted by the function

$$\widehat{F}(x) = f(T) - f(x) - R_a \ln\left(\frac{p}{p_0}\right). \tag{28}$$

Consequently, the resulting approximated reference potential temperature, i.e., the respective zero of the function $\widehat{F}(x)$, is denoted as $\theta_{\text{ref}}^{\text{approx}}$.

     The difference between the approximation result and the reference, i.e.,

$$\theta_{\text{ref}} - \theta_{\text{ref}}^{\text{approx}}, \tag{29}$$

is then referred to as the basic error of the approximation. Note that the replacement of the function $F$ by $\widehat{F}$ only circumvents the integration in $F$; the root-finding problem $0 = \widehat{F}(x)$ for the approximated reference potential temperature $\theta_{\text{ref}}^{\text{approx}}$ remains analytically not solvable.

Therefore, the second move towards a practical approximation procedure is to construct approximations $\theta^{(k)}$ to the zero of $\widehat{F}(x)$ by using Newton's method, see Appendix C2. Newton's method is an iterative procedure; the notation $\theta^{(k)}$ refers to the $k$-th computed iterate. Hence, $\theta^{(k)}$ constitutes an approximation to $\theta_{\text{ref}}^{\text{approx}}$, and, in the limit $k \to \infty$, the approximation error

$$\theta_{\text{ref}}^{\text{approx}} - \theta^{(k)} \tag{30}$$

vanishes. Two formulations of Newton's method are distinguished in Appendix C2, i.e., the principal application of Newton's method, and its further derivative, called Householder's method. Both formulations require the stipulation of one of the iterates $\theta^{(k)}$ as sufficient to obtain a result of appropriate accuracy. The higher the number of iterations, of course, the smaller is the error (30), whereas the basic error (29) remains unaffected by the number of iterations. Hence, in any case, the basic error (29) is to be accepted as at least implied in the final approximation, even though a well-chosen $\theta^{(k)}$ could result in an approximation error $\theta_{\text{ref}} - \theta^{(k)}$ that is smaller than the basic error.

The various errors implied in the proposed approximation procedure combining for the approximation's total error, as well as accompanying details, are discussed in Appendix D. In brief, Figure 5a illustrates the basic error (29) based on the pressure and temperature profiles of the US Standard Atmosphere, as these provide atmospherically meaningful averages of realistic temperature-pressure data pairs. Based on the parameters of the US Standard Atmosphere, the basic error inherent with the approximation remains below $1.25\,\text{K}$ up to altitudes of $50\,\text{km}$. Thus, regarding the subsequent iteration process, a substantial improvement of the error compared to $\sim 1.5\,\text{K}$ is not to be expected for the total error of approximating the reference potential temperature.

An error analysis exclusively based on the US Standard Atmosphere is constrained to specific combinations of the air's pressure and temperature, potentially suppressing latent errors that may emerge if certain fluctuations of the real atmosphere's temperature and pressure profiles are considered. Thus, the error analysis is extended to an atmospheric pressure ($p$) and temperature ($T$) range, from $1000\,\text{hPa}$ to $0.5\,\text{hPa}$ and from $180\,\text{K}$ to $300\,\text{K}$, such that the conditions within the entire troposphere and stratosphere, including the stratopause, are covered. Figure 5b illustrates the absolute basic error (29) for the extended ranges of pressure and temperature while Figure 5c illustrates the relative basic error $|\theta_{\text{ref}} - \theta_{\text{ref}}^{\text{approx}}| / \theta_{\text{ref}}$. The contours in Figures 5b and 5c mainly highlight two regions: at $\sim 100\,\text{hPa}$ where $\Delta\theta$ never rises above $0.75\,\text{K}$ which corresponds to a maximum relative basic error of $0.15\,\%$, and in a pressure range from $\sim 5\,\text{hPa}$ to $1\,\text{hPa}$ where a $\Delta\theta$ of $1.25\,\text{K}$ is never exceeded, corresponding to relative errors of at most $0.1\,\%$. Note that the entire $\Delta\theta$ scale ranges up to $3\,\text{K}$, which may only be reached at pressures below $0.8\,\text{hPa}$ combined with temperatures above $280\,\text{K}$.

As previously discussed, the basic error is unavoidable and is to be accepted when applying the suggested substitution for the integral in the definition of the function $F(x)$ in (23). However, as outlined in Appendix C2, the second iterate $\theta^{(2)}$ of Newton's method (principal application), may thoroughly suffice for the final approximation to the reference potential temperature $\theta_{\text{ref}}$, as this iteration level already features an approximation error (30) which is negligibly small. Figure 6a illustrates the total

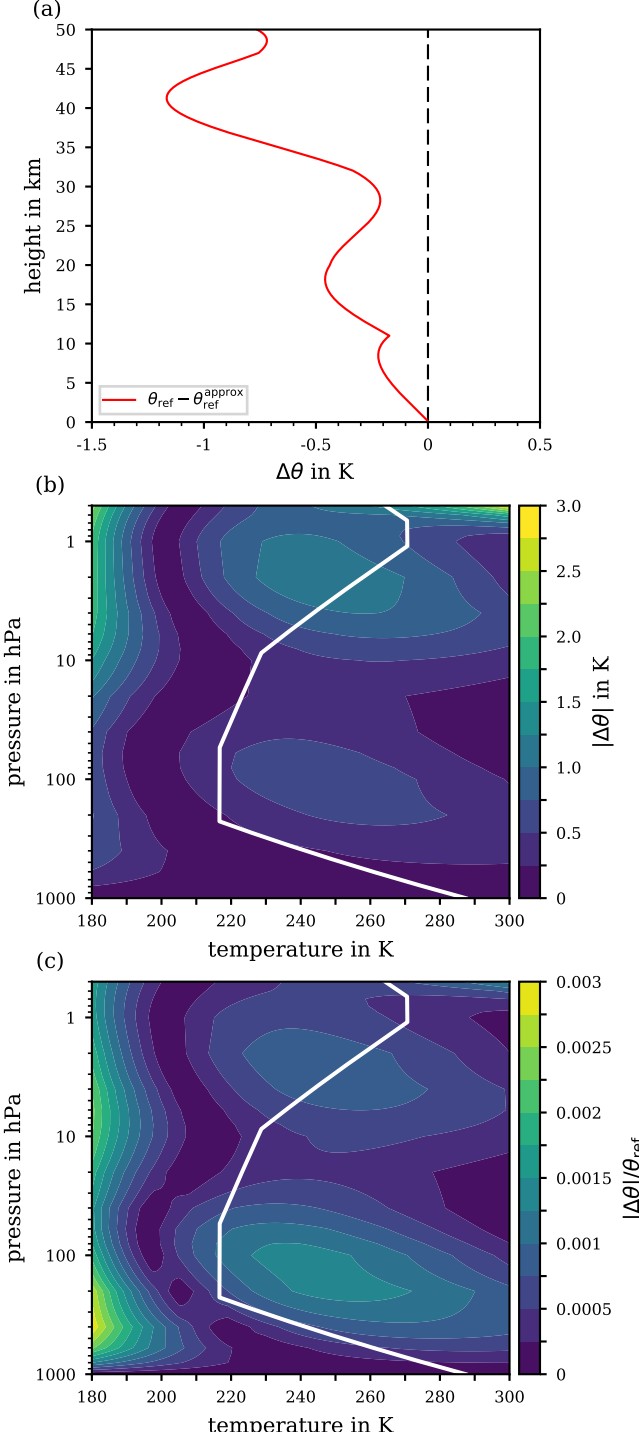

**Figure 5.** Absolute basic error $\Delta\theta = \theta_{\mathrm{ref}} - \theta_{\mathrm{ref}}^{\mathrm{approx}}$, cf. (29), from approximating the reference potential temperature along the US Standard Atmosphere (a) and for the extended pressure range $1000\,\mathrm{hPa}$ to $0.5\,\mathrm{hPa}$ and temperature range $180\,\mathrm{K}$ to $300\,\mathrm{K}$ (b). For orientation, the white solid line indicates the $p$-$T$-profile from the US Standard Atmosphere. The relative basic error $|\Delta\theta|/\theta_{\mathrm{ref}}$ is shown in panel (c) for the extended pressure and temperature range.

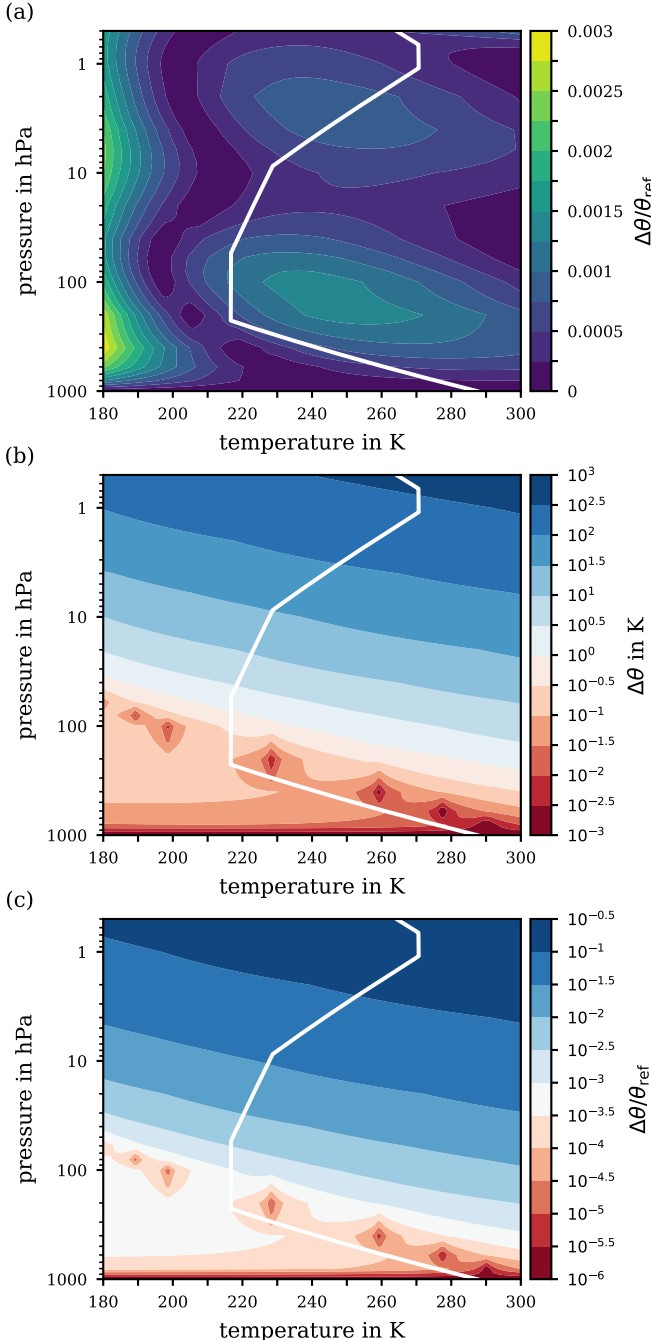

**Figure 6.** (a) Relative error $\Delta\theta/\theta_{\mathrm{ref}} = \left|\theta^{(2)} - \theta_{\mathrm{ref}}\right|/\theta_{\mathrm{ref}}$ of the second iterate $\theta^{(2)}$, obtained with Newton's method for the ranges of pressure and temperature from $1000\,\mathrm{hPa}$ to $0.5\,\mathrm{hPa}$ and from $180\,\mathrm{K}$ to $300\,\mathrm{K}$, respectively. Panels (b) and (c) exhibit the difference $\Delta\theta = \left|\theta_{1005} - \theta_{\mathrm{ref}}\right|$ and relative difference $\Delta\theta/\theta_{\mathrm{ref}}$, respectively, on a logarithmic scale between the reference potential temperature $\theta_{\mathrm{ref}}$ and the potential temperature $\theta_{1005}$ based on a constant specific heat capacity ($c_p = 1005\,\mathrm{J\,kg^{-1}K^{-1}}$). For orientation, the white solid line indicates the $p$-$T$-profile from the US Standard Atmosphere.

relative error of the suggested approximation $\theta^{(2)}$ with respect to the ultimate reference $\theta_{\mathrm{ref}}$ for the extended ranges of pressure and temperature. Indeed, the contour pattern in Figure 6a and the basic relative approximation error shown in Figure 5c are remarkably similar. Thus, the iteration process itself imparts only a minor contribution to the total error compared to the basic approximation error.

The total approximation error, which is

$$\theta_{\mathrm{ref}} - \theta^{(2)} = \left(\theta_{\mathrm{ref}} - \theta_{\mathrm{ref}}^{\mathrm{approx}}\right) + \left(\theta_{\mathrm{ref}}^{\mathrm{approx}} - \theta^{(2)}\right), \tag{31}$$

is dominated by the unavoidable basic error (first bracket) and augmented by a negligible error inherent to the iteration (second bracket), also supporting the conclusion that the second iterate of Newton's method is an appropriate approximation procedure. Figure 7 presents step-wise instructions for the computation of the second iterate approximation to the reference potential temperature, and may serve as a guide to follow the numerous equations and intermediate analytical steps described throughout the derivations in Appendix C.

For completeness, Figures 6b and 6c exhibit a final comparison by means of the logarithmic difference and the logarithmic relative difference between the reference potential temperature $\theta_{\mathrm{ref}}$ and the conventional definition $\theta_{c_p}$ (WMO, 1966) based on a constant specific heat capacity $c_p = 1005\,\mathrm{J\,kg^{-1}K^{-1}}$. Notably, over a wide altitude range within the troposphere (i.e., for atmospheric pressures greater than $\sim 100\,\mathrm{hPa}$), the absolute error $\Delta\theta = |\theta_{1005} - \theta_{\mathrm{ref}}|$ remains below $1\,\mathrm{K}$, cf. Figure 6b, corresponding to a relative error $\Delta\theta/\theta_{\mathrm{ref}}$ of at most $0.1\,\%$. However, in the pressure range below $\sim 100\,\mathrm{hPa}$, deviations of the real atmospheric conditions from those of the US Standard Atmosphere could increase the absolute error $\Delta\theta$ from a few K to up to $10\,\mathrm{K}$, corresponding to an increase of the relative error to $1\,\%$. Further critical pressure levels are at $\sim 20\,\mathrm{hPa}$ and $\sim 5\,\mathrm{hPa}$, where the error's magnitude increases to several tens and several hundreds of K, respectively. At a pressure of $0.5\,\mathrm{hPa}$, an absolute error $\Delta\theta$ of up to $500\,\mathrm{K}$ is reached, which corresponds to a relative error of $10\,\%$ or even more.

## 5.3   Implementation aspects

The use of the new reference potential temperature $\theta_{\mathrm{ref}}$ in a numerical model requires additional computational effort to perform corresponding calculations. Hereafter, two aspects are briefly discussed: (i) the formulation of the model equations, which include $\theta_{\mathrm{ref}}$ and (ii) the calculation of $\theta_{\mathrm{ref}}$.

Although it is beyond the scope of the present study to provide a general derivation of an appropriate energy equation based on $\theta_{\mathrm{ref}}$ for atmospheric models, a formulation of the total derivative of $\theta_{\mathrm{ref}}$ is given by

$$c_p(\theta_{\mathrm{ref}})\frac{\mathrm{d}\theta_{\mathrm{ref}}}{\theta_{\mathrm{ref}}} = c_p(T)\frac{\mathrm{d}T}{T} - R_a\frac{\mathrm{d}p}{p}, \tag{32}$$

where the details of its derivation are given in Appendix E. The total derivative of $\theta_{\mathrm{ref}}$ may be useful, since the governing equations are commonly formulated as differential equations.

The calculation of both the reference potential temperature $\theta_{\mathrm{ref}}$ and its approximation $\theta_{\mathrm{ref}}^{\mathrm{approx}}$ on the basis of given values of pressure $p$ and temperature $T$ requires an iterative procedure. The additional computational effort inherent with these calculations depends on the number of iterations. If, however, the second iteration $\theta^{(2)}$ already represents an appropriate approximation of $\theta_{\mathrm{ref}}$ (cf. Section 5.2), then the flowchart in Figure 7 immediately conveys the additional computational effort to

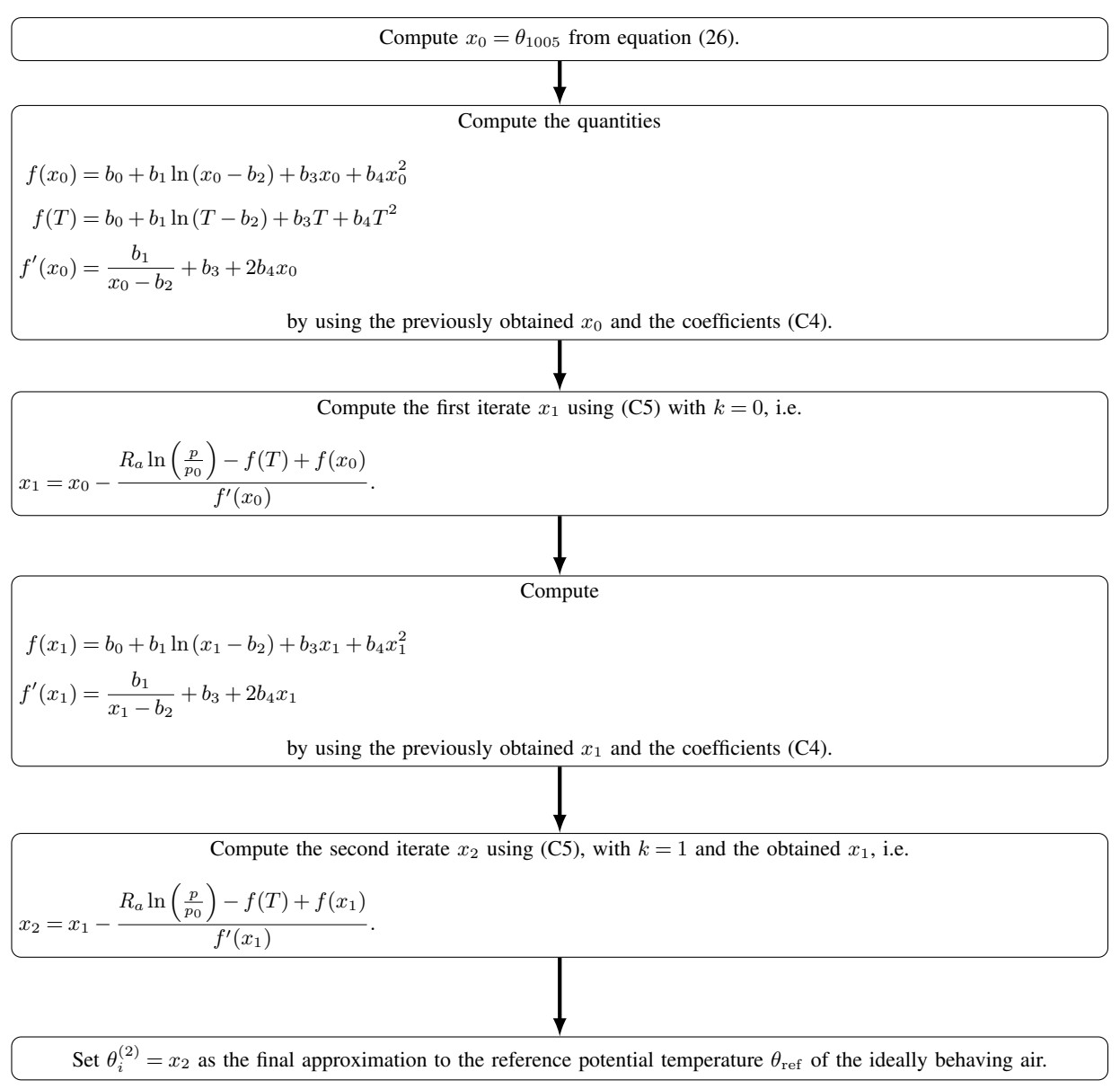

**Figure 7.** Flowchart guiding through the process of computing the approximation $\theta^{(2)}$ by using Newton's formulation (C5) until its second iteration, wherein $T$ (in K) and $p$ (in hPa) are the atmospheric air conditions in terms of temperature and pressure, respectively, and $p_0$ is set to 1000 hPa (WMO, 1966). Table C1 collects values of $\theta_{\mathrm{ref}}$ and the approximation $\theta^{(2)}$ together with intermediate results for selected pairs of temperature and pressure to verify a computation according to this instruction.

be expected. The calculation of the starting value $x_0$ is identical to computing $\theta_{1005}$. An additional effort results from the evaluation of the functions $f$ (three times) and $f'$ (two times), respectively, and the combination (two times) of obtained values to determine $x_1$ and $x_2$. Since each of these evaluations causes additional numerical steps, the computational effort to obtain $\theta^{(2)}$ is in total about seven times more than the calculation of the conventional $\theta_{1005}$, while the algorithmic complexity is constant.

## 6 The potential temperature for air as a real gas

To account for real-gas effects (that cause a behaviour other than that of an ideal gas cf. Section 4) on the potential temperature, we use the model embedded in *REFPROP* (Lemmon et al., 2018), a standard reference database from NIST. This model treats air as a mixture and employs state-of-the art reference equations of state for pure nitrogen (Span et al., 2000), oxygen (Schmidt and Wagner, 1985), and argon (Tegeler et al., 1999). The mixing rule and binary interaction parameters are taken from the GERG-2008 model (Kunz and Wagner, 2012). From its definition in terms of an isentropic process, the potential temperature $\theta_{\mathrm{real}}(T, p)$ is defined implicitly by

$$s(\theta_{\mathrm{real}}, p_0) = s(T, p), \tag{33}$$

where $s$ is the specific entropy. Calculating $\theta_{\mathrm{real}}(T, p)$ is a two-step process. First, the specific entropy $s$ is computed at temperature $T$ and pressure $p$. Then, the temperature $\theta_{\mathrm{real}}$ is found that gives the same entropy $s$ at the ground pressure $p_0$. This is an iterative calculation, but it is accomplished automatically within the *REFPROP* software (Lemmon et al., 2018).

One caveat should be mentioned regarding the computed potential temperatures. The range of validity of the equations of state for the air components (Span et al., 2000; Schmidt and Wagner, 1985; Tegeler et al., 1999) extends only up to $2000\,\mathrm{K}$. At very high altitudes, computed values of $\theta_{\mathrm{real}}$ exceed this limit. While all the equations extrapolate in a physically realistic way, their quantitative accuracy is less certain above $2000\,\mathrm{K}$. This caveat also applies to the ideal-gas calculations; the correlations for $c_p^0(T)$ for $N_2$ and $O_2$ are extrapolations beyond $2000\,\mathrm{K}$. However, since the same ideal-gas values are used in the real-gas calculations, any inaccuracy in $c_p^0(T)$ will cancel when evaluating the difference between ideal-gas and real-gas values of $\theta$.

Figure 8 illustrates the comparison between the real-gas potential temperature $\theta_{\mathrm{real}}$ and the ideal-gas reference potential temperature $\theta_{\mathrm{ref}}$. Figure 8a shows the difference $\theta_{\mathrm{real}} - \theta_{\mathrm{ref}}$ along the $p$-$T$-profile of the US Standard Atmosphere and Figure 8b accounts for any $p$-$T$-combination of extended range but shows the relative difference instead. The difference between $\theta_{\mathrm{real}}$ and $\theta_{\mathrm{ref}}$ never exceeds $0.1\,\mathrm{K}$ for the absolute difference or $30 \cdot 10^{-5} = 0.03\,\%$ for the relative difference. As may be anticipated from the deviation of $c_p^0$ shown in Figure 3 at low temperatures both from the experimentally determined values (which may be inaccurate) as well as from the *REFPROP* data, the real-gas effect on the specific heat capacity of dry air tends to increase towards the coldest gas temperatures. However, the difference between the real- and ideal-gas approaches results in essentially no substantial difference between the resulting $\theta$'s, neither at ground conditions (for any temperature at $\sim 1000\,\mathrm{hPa}$) nor at very high altitudes (at pressures below $\sim 1\,\mathrm{hPa}$). While the negligible difference between $\theta_{\mathrm{real}}$ and $\theta_{\mathrm{ref}}$ near ground levels is less surprising, the diminished difference at higher altitudes reflects that in this region the potential temperature reaches such high values that the difference between the real-gas and the ideal-gas specific heat capacity becomes insignificant. Within the

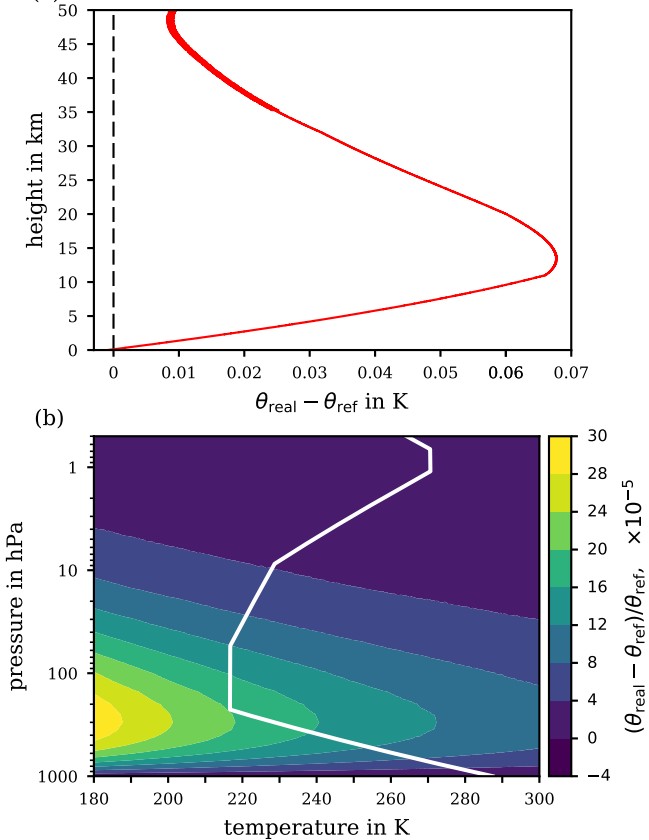

**Figure 8.** Difference $\theta_{\mathrm{real}} - \theta_{\mathrm{ref}}$ reflecting the deviation of the potential temperature $\theta_{\mathrm{real}}$, based on the properties of air behaving as a real gas under variable temperature and pressure, from the herein derived potential temperature expression $\theta_{\mathrm{ref}}$ for the ideal-gas limit of the air's specific heat capacity $c_p^0(T)$. (a) Difference along the profile of the US Standard Atmosphere. (b) Relative difference in $p$-$T$-coordinates covering any combination of atmospherically relevant temperatures and pressures.

intermediate (stratospheric) region, the low pressures (and thus the low air densities) cause the ideal-gas assumption to be an
accurate approximation even at low temperatures. In general, the degree to which a gas can be treated as ideal is primarily a
function of the (molar) density. For an ideal gas, the density is proportional to the quotient $\frac{p}{T}$; this is almost true also for real
air. Hence, declining pressures together with rising temperatures both make the air's behaviour increasingly close to ideal.

## 7  Implications on the use of the potential temperature

As previously shown, the newly defined reference potential temperature $\theta_{\mathrm{ref}}$ deviates most from the WMO-defined potential
temperature $\theta_{1005}$ at stratospheric altitudes and above (cf. Figure 6). More particularly, not only do the values from both $\theta$
definitions differ, but also their vertical derivatives, i.e., $\frac{\partial \theta_{\mathrm{ref}}}{\partial z}$ and $\frac{\partial \theta_{1005}}{\partial z}$. Whether such deviations have a significant effect on

an application is very case-dependent and requires detailed examination and specific appraisal. Below, four typical applications of the potential temperature were selected and are examined regarding the quantitative effect on the results due to deviations of the introduced reference potential temperature compared to the conventional and commonly used $\theta_{1005}$. The purpose of this examination is to document the magnitude of errors to allow a well-founded, individual decision for each application of the potential temperature whether it is worth applying the more rigorous calculation in the particular context.

### 7.1 The Brunt-Väisälä frequency

The formula for the (squared) Brunt-Väisälä frequency $N^2$ is often given in the form of (2), i.e., a formula involving the potential temperature $\theta$. The substitution of $\theta$ in equation (2) by the new reference potential temperature $\theta_{\mathrm{ref}}$ may be tempting, but it is erroneous and the resulting quantity is denoted as $N^2_{\mathrm{false}}$. The Brunt-Väisälä frequency is not defined by equation (2), since this formula results from various simplifications in its derivation, e.g., by assuming hydrostatic conditions and a constant specific heat capacity. Consequently, the substitution of $\theta_{\mathrm{ref}}$ in equation (2) leads to a wrong formula for the Brunt-Väisälä frequency that does not correctly consider the temperature dependence of dry air's specific heat capacity.

The Brunt-Väisälä frequency is the oscillation frequency of an air parcel due to a local density perturbation (see, e.g., Durran and Klemp, 1982; Marquet and Geleyn, 2013; Wallace and Hobbs, 2006; Ambaum, 2010). Retaining the assumption of hydrostatic conditions, the defining formula yields

$$N^2 = \frac{g}{T} \left( \frac{\partial T}{\partial z} + \frac{g}{c_p(T)} \right) \tag{34}$$

where the temperature-dependent specific heat capacity $c_p(T)$ was implied, and which quantifies the balance between the actual temperature stratification $\frac{\partial T}{\partial z}$ and the dry adiabatic lapse rate $-\frac{g}{c_p(T)}$ (e.g., Holton, 2004).

To illustrate the deviation of $N^2_{\mathrm{false}}$ from $N^2$, vertical profiles of both variables were calculated based on the temperature profiles shown in Figure 9a. The temperature data are taken from the Upper Atmosphere Research Satellite Reference Atmosphere Project (URAP, see Swinbank and Ortland, 2003) data and are assumed as typical at mid-latitudes during June and December. The temperature profiles extend up to altitudes of $85\,\mathrm{km}$ and thus cover the entire stratosphere and most of the mesosphere. The hydrostatic assumption allowed for computing pressure profiles along the URAP values for the vertical temperature distribution. Subsequently, the reference potential temperature $\theta_{\mathrm{ref}}$ and its vertical derivative were calculable. The resulting vertical profiles for $N^2_{\mathrm{false}}$ and the true Brunt-Väisälä frequency $N^2$ are shown in Figure 9b. Evidently, the values of $N^2_{\mathrm{false}}$ (dashed lines) deviate significantly from $N^2$ (solid lines) and increasingly so towards higher altitudes above $15\,\mathrm{km}$. However, the absolute deviation $|N^2 - N^2_{1005}|$, using $N^2_{1005}$ as calculated with $\theta_{1005}$ in accordance with Equation (2), does not exceed $1.6 \cdot 10^{-6}\,\mathrm{s}^{-2}$ (not shown), indicating that $N^2_{1005}$ is a good representation of $N^2$ along these temperature profiles.

For equations involving the potential temperature, however, it should be emphasised that the substitution of $\theta$ by $\theta_{\mathrm{ref}}$ rarely succeeds and that instead the entire derivation of the equations requires careful consideration of the assumptions, such as the constancy of $c_p$, to avoid aberrations and erroneous conclusions.

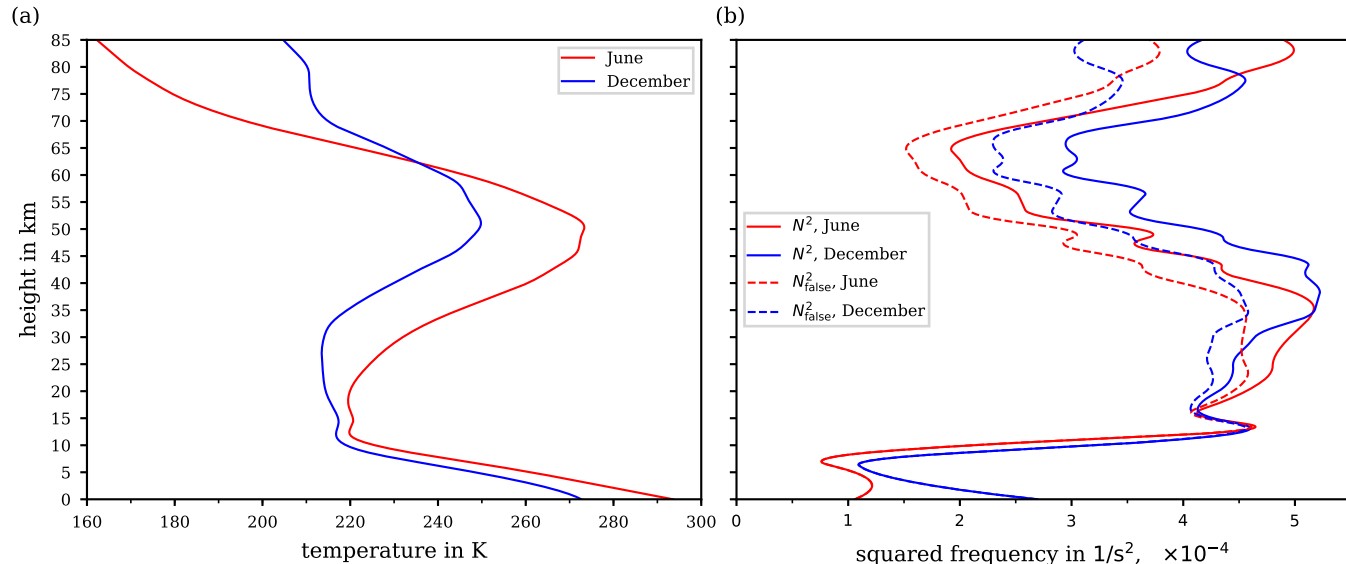

**Figure 9.** Vertical profiles of (a) the temperature up to $85\,\mathrm{km}$ altitude as typical for mid-latitudes in June (red curve) and December (blue curve). (b) Resulting wrong Brunt-Väisälä frequency $N_{\mathrm{false}}^2$ (dashed lines) and the true Brunt-Väisälä frequency $N^2$ (solid lines) for the two temperature profiles from panel (a).

### 7.2 The Potential Vorticity

Ertel's potential vorticity (e.g., Ertel, 1942; Hoskins et al., 1985; Schubert et al., 2004; Holton, 2004) may be defined as the potential vorticity of the dry air potential temperature by

$$PV(\theta) = \frac{1}{\rho} \left( 2\boldsymbol{\Omega} + \nabla \times \boldsymbol{u} \right) \cdot \nabla \theta. \tag{35}$$

In this definition, $2\boldsymbol{\Omega} + \nabla \times \boldsymbol{u}$ is the absolute vorticity, $\boldsymbol{\Omega}$ denotes Earth's angular velocity, $\boldsymbol{u}$ the three-dimensional wind vector, and $\rho$ the air density (see, e.g., Hoskins et al., 1985; Cotton et al., 2011; Marquet, 2014). Since (35) represents the defining equation for Ertel's potential vorticity, the two potential vorticities

$$\begin{aligned} PV_{\mathrm{ref}} &= PV(\theta_{\mathrm{ref}}), \\ PV_{1005} &= PV(\theta_{1005}) \end{aligned} \tag{36}$$

based on the new reference potential temperature $\theta_{\mathrm{ref}}$ and $\theta_{1005}$, respectively, are considered. To provide a first comparison of these potential vorticities, $\boldsymbol{u} = 0$ is assumed, i.e., an atmosphere at rest. Additionally, the potential temperature is assumed as horizontally constant. Consequently, (35) reduces to

$$PV(\theta) = \frac{2\sin(\phi)}{\rho} \frac{2\pi}{t_{\mathrm{E}}} \frac{\partial \theta}{\partial z} \tag{37}$$

for a position on Earth with geographical latitude $\phi$ and $t_{\mathrm{E}} = 24\,\mathrm{h}$, the duration of one rotation of the Earth.

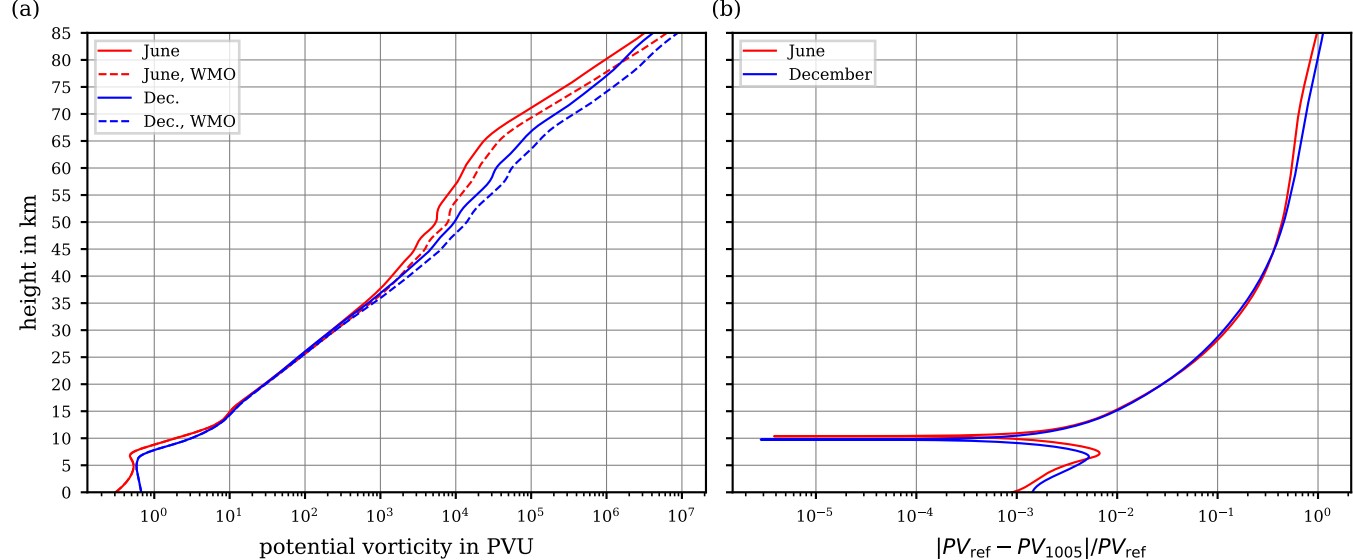

**Figure 10.** (a) Vertical profiles of the potential vorticitiy $PV_{\text{ref}}$ computed with $\theta_{\text{ref}}$ (solid lines), and $PV_{1005}$ computed with $\theta_{1005}$ (dashed lines), for an atmosphere at rest along the temperature profiles from Figure 9a for June (red lines) and December (blue lines). Since the temperature profiles are representative for mid-latitudes on the northern hemisphere, the geographical latitude in (37) was set to $52°\text{N}$. (b) Relative deviation $|PV_{\text{ref}} - PV_{1005}|/PV_{\text{ref}}$ of the potential vorticity profiles from panel (a).

Using the temperature profiles from Figure 9a together with the values of the potential temperatures $\theta_{\text{ref}}$ and $\theta_{1005}$, the evaluation of the two potential vorticities (36) and (37) yields the potential vorticity profiles shown in Figure 10a while their relative deviations are shown in Figure 10b. Since the temperature profiles are representative for the north-hemispheric mid-latitudes, the geographical latitude $\phi$ in (37) was set to $52°\text{N}$. At tropospheric altitudes, the relative deviation between $\theta_{\text{ref}}$ and $\theta_{1005}$ is small and never exceeds $\sim 1\%$, while it continuously increases towards higher altitudes. According to these profiles, the relative deviation exceeds $10\%$ at $30\,\text{km}$ and reaches $100\%$ at the highest altitudes.

It is noteworthy, however, that the computations of both $N^2$ (cf. Section 7.1 and Figure 9b) and PV (Figure 10b) are based on the specific temperature profiles from URAP (cf. Section 7.1 and Figure 9a) and thus are not of general validity. The selection of these temperature profiles was entirely arbitrary and exclusively aimed at illustrating possible implications of the use of the developed reference potential temperature. The resulting and indicated deviations are ultimately subject to individual assessment on applying $\theta_{\text{ref}}$.

## 7.3 Vertical sorting of data

For atmospheric investigations, e.g., in the region of the upper troposphere and lower stratosphere (UT/LS), it is common practice to set vertical profiles of atmospheric parameters in relation to the potential temperature as vertical coordinate. This way, the increasingly isentropic stratification of the atmosphere above the UT is taken into account. The transport of an air mass

along isentropic surfaces, i.e., surfaces of constant potential temperature and entropy, is to be regarded as adiabatic. Hence, the air's composition and properties within the same isentrope interval, regardless of the observation location, is better comparable than it would be if based on other isopleths (i.e., height or pressure coordinates). Investigations of air mass compositions over time and from different regions at the same $\theta$-level largely exclude that, during its transport history, the air had experienced vertical displacement and/or diabatic processes (radiative heating, condensation/evaporation) which would result in energy conversion. The tropopause height is often used as a reference height in the $\theta$ coordinate system in connection with the vertical sorting of observational data, whereby the assignment of tropospheric and stratospheric processes is made, or exchanges across the tropopause are investigated (Holton et al., 1995; Stohl et al., 2003). Consequently, the tropopause height is also determined by the potential vorticity (e.g., Gettelman et al., 2011, and cf. Section 7.2), if the conventional tropopause definitions (cold point or lapse rate, WMO, 1957) do not allow for clearly determining the tropopause height, e.g., in the Asian Monsoon Anticyclone (cf. Höpfner et al., 2019) or in the polar winter vortex (Wilson et al., 1989; Weigel et al., 2014). The conventional definition of $\theta$ implies a systematic error in the vertical sorting of observational data in the $\theta$ coordinate system, independent of the measurement platform. Investigations with high-altitude research aircraft such as the G-550 HALO (e.g., Wendisch et al., 2016; Voigt et al., 2017), the NASA WB-57 or ER-2 (e.g., Murphy et al., 2007; Dessler, 2002), the M-55 Geophysica (Curtius et al., 2005; Borrmann et al., 2010; Frey, 2011), balloon-borne platforms (Lary et al., 1995; Vernier et al., 2018), or satellite-based vertical profiles (e.g., Davies et al., 2006; Spang et al., 2005), require consideration of the systematic error in $\theta$ if calculated as $\theta_{c_p}$ in compliance with the definition by the WMO (1966). The possibly inconsistent use of a constant $c_p$ value of $1004\,\mathrm{J\,kg^{-1}K^{-1}}$ or $1005\,\mathrm{J\,kg^{-1}K^{-1}}$ (or any other) in different and compared data sets, which could be due to different literature references for this value (cf. Table 1), will not be explored here. At altitudes between $15$ and $20\,\mathrm{km}$ (ceiling of high-altitude research aircraft), an overestimation by about $0.1-0.5\,\%$ is to be expected for the potential temperature according to the conventional definition, cf. Figure 4b. At altitudes of $30-35\,\mathrm{km}$, an overestimation by up to $2-5\%$ results. Whether this error is significant or small compared to the uncertainty of ambient temperature and pressure measurement aboard the respective aircraft is left to individual judgement in the course of data processing. In the case of spacecraft-bound vertical soundings (e.g., from ASTROSPAS, SCIAMACHY, or ENVISAT), the error in the potential temperature determined by $\theta_{c_p}$ exceeds $10\,\%$ at altitudes above $40\,\mathrm{km}$, as shown in Figure 4b. Finally, we note that the specified errors apply exclusively along the vertical profile of the US standard atmosphere, and that deviations of the actual temperature profile from the US standard atmosphere, e.g., warmer temperatures, could lead to larger errors (cf. Figure 6).

## 7.4 Diabatic heating rates

Diabatic heating rates refer to the rate of energy $\frac{\mathrm{d}q}{\mathrm{d}t}$ supplied to a given air parcel, e.g., by radiative heating, and are given in units of $\mathrm{J\,kg^{-1}s^{-1}}$. This energy supply causes a temperature change of an air parcel at a rate which hereafter is referred to as

the *absolute heating rate*,

$$\text{AHR}_{\text{ref}}\left(\frac{\mathrm{d}q}{\mathrm{d}t}\right) = \frac{\mathrm{d}T}{\mathrm{d}t} = \frac{1}{c_p^0(T)}\frac{\mathrm{d}q}{\mathrm{d}t},$$

$$\text{AHR}_{1005}\left(\frac{\mathrm{d}q}{\mathrm{d}t}\right) = \frac{\mathrm{d}T}{\mathrm{d}t} = \frac{1}{1005\,\mathrm{J\,kg^{-1}K^{-1}}}\frac{\mathrm{d}q}{\mathrm{d}t}. \qquad (38)$$

Again, the distinction was made between the temperature-dependent $c_p^0(T)$ and the constant $c_p = 1005\,\mathrm{J\,kg^{-1}K^{-1}}$ specific heat capacity. From the defining equations (38), the relative difference between these absolute heating rates, where $x$ designates an arbitrary diabatic heating rate, is

$$\frac{\text{AHR}_{1005}(x) - \text{AHR}_{\text{ref}}(x)}{\text{AHR}_{\text{ref}}(x)} = \frac{c_p^0(T)}{1005\,\mathrm{J\,kg^{-1}K^{-1}}} - 1. \qquad (39)$$

Apart from the absolute heating rates for the change of absolute temperature, the change of potential temperature due to a diabatic heating rate $\frac{\mathrm{d}q}{\mathrm{d}t}$ is of interest. For example, it is the change of potential temperature that modifies the altitude of modelled trajectories in Lagrangian chemical transport models based on isentropic coordinates rather than the change in absolute temperature (e.g., the SLIMCAT (Chipperfield, 2006) or CLaMS model (Pommrich et al., 2014)).

Taking the relation $T\,\mathrm{d}s = \mathrm{d}q$ for the specific entropy into account, Gibbs' equation (8) may be rewritten as

$$\frac{\mathrm{d}q}{T} = \frac{c_p(T)}{T}\mathrm{d}T - R_a\frac{\mathrm{d}p}{p}. \qquad (40)$$

Comparing the right-hand side of this equation to the total derivative of the new reference potential temperature $\theta_{\text{ref}}$ (see Appendix E for the detailed computation and Equation (E6) for the result) equation (40) amounts to

$$\frac{\mathrm{d}q}{T} = c_p(\theta_{\text{ref}})\frac{\mathrm{d}\theta_{\text{ref}}}{\theta_{\text{ref}}}. \qquad (41)$$

Consequently, the following two diabatic heating rates

$$\frac{\mathrm{d}\theta_{\text{ref}}}{\mathrm{d}t} = \frac{\theta_{\text{ref}}}{c_p^0(\theta_{\text{ref}})T}\frac{\mathrm{d}q}{\mathrm{d}t} = \text{HR}_{\text{ref}}\left(\frac{\mathrm{d}q}{\mathrm{d}t}\right),$$

$$\frac{\mathrm{d}\theta_{1005}}{\mathrm{d}t} = \frac{\theta_{1005}}{(1005\,\mathrm{J\,kg^{-1}K^{-1}})\cdot T}\frac{\mathrm{d}q}{\mathrm{d}t} = \text{HR}_{1005}\left(\frac{\mathrm{d}q}{\mathrm{d}t}\right) \qquad (42)$$

for the potential temperatures $\theta_{\text{ref}}$ and $\theta_{1005}$ may be defined. Denoting again by $x$ an arbitrary diabatic heating rate, the relative difference between the heating rates (42) is

$$\frac{\text{HR}_{1005}(x) - \text{HR}_{\text{ref}}(x)}{\text{HR}_{\text{ref}}(x)} = \frac{\theta_{1005}}{\theta_{\text{ref}}}\frac{c_p^0(\theta_{\text{ref}})}{1005\,\mathrm{J\,kg^{-1}K^{-1}}} - 1. \qquad (43)$$

In order to judge the magnitudes of the relative differences (39) and (43), the monthly averaged temperature profiles from ERA-Interim (Dee et al., 2011) data for $52°\mathrm{N}$ geographical latitude are used, see Figure 11a. The relative differences of the absolute heating rates (39) are shown in Figure 11b and the difference appears to be small. However, the relative differences of the heating rates (43) in Figure 11c are much larger, as relative deviations exceeding $50\,\%$ are reached in the upper stratosphere

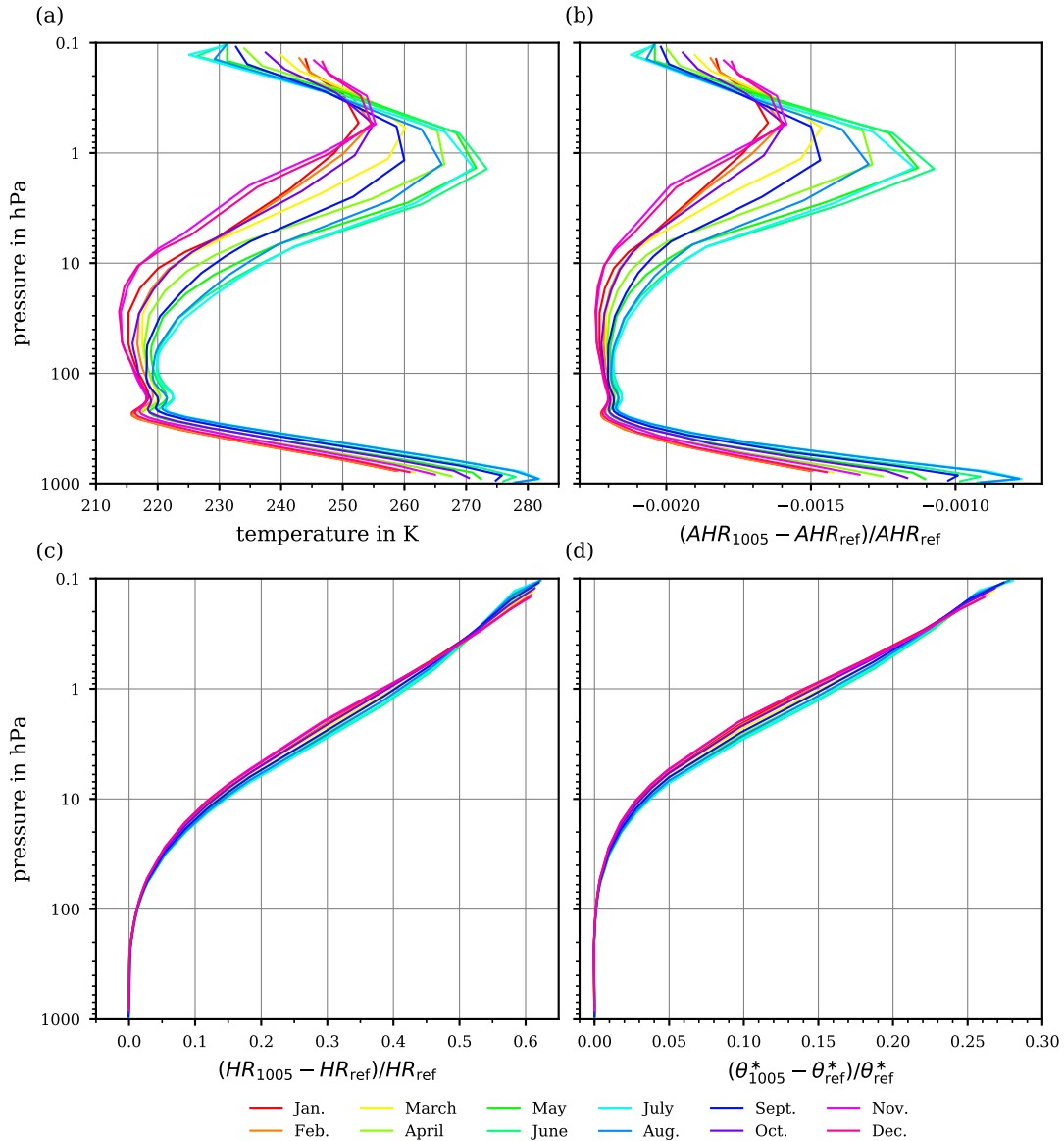

**Figure 11.** (a) Monthly averaged temperatures profiles for $52°$N. (b) The relative differences between the absolute heating rates, defined in (39). (c) The relative differences between the heating rates (43) for the potential temperatures $\theta_{1005}$ and $\theta_{\text{ref}}$. (d) The resulting potential temperatures $\theta^*_{1005}, \theta^*_{\text{ref}}$ after $24\,\text{h}$ of heating with constant diabatic heating $\frac{\text{d}q}{\text{d}t}$ and the resulting heating rates $\text{HR}_{\text{ref}}, \text{HR}_{1005}$ at constant pressure.

and lower mesosphere (at pressures below $1\,\mathrm{hPa}$). Additionally, the temperatures were computed that resulted after $24\,\mathrm{h}$ of heating with a constant heating rate $\frac{\mathrm{d}q}{\mathrm{d}t}$ as given in the (averaged) dataset, where a constant pressure is assumed for simplicity. As may be anticipated from the small deviations in Figure 11b, the difference in the final absolute temperatures by using the absolute heating rates $\mathrm{AHR}_{1005}$ or $\mathrm{AHR}_{\mathrm{ref}}$ are smaller than $0.044\,\mathrm{K}$. However, the differences in the potential temperatures $\theta^*_{1005}, \theta^*_{\mathrm{ref}}$, computed with the heating rates $\mathrm{HR}_{\mathrm{ref}}, \mathrm{HR}_{1005}$, are much larger (Figure 11d), and amount to about $3\,\%$ at $10\,\mathrm{hPa}$ and about $15\,\%$ at $1\,\mathrm{hPa}$. For transport calculations done in isentropic coordinates, these differences are of the same order of magnitude as the deviations resulting from the use of the temperature-dependent instead of the constant $c_p$. It remains to be decided on individual application whether this additional effect in the calculation is significant.

A standard diagnostic for the speed of the stratospheric circulation is the time lag of the upward propagating seasonal signal in tropical stratospheric water vapour (the so-called tape recorder, Mote et al., 1996). Here, differences between calculations (done in isentropic coordinates) based on different current meteorological reanalysis data sets amount to about $10-30\,\%$ for the signal's upward propagation below about $10\,\mathrm{hPa}$ (Tao et al., 2019), such that the additional deviation from using the temperature-dependent $c_p$ is comparably small. However, in cases of smaller inter-model differences the additional $c_p$-related uncertainty needs to be assessed.

Note, the determination of absolute temperatures $T^*_{1005}, T^*_{\mathrm{ref}}$ which correspond to the resulting potential temperatures $\theta^*_{1005}, \theta^*_{\mathrm{ref}}$ after $24\,\mathrm{h}$ differ by less than $0.014\,\mathrm{K}$ (not shown).

## 8 Summary and Conclusions

Under the assumption that dry air is an ideal gas, a re-assessment of computing the potential temperature was introduced that accounts for the hitherto unconsidered temperature dependence of air's specific heat capacity. The new reference potential temperature $\theta_{\mathrm{ref}}$ was introduced, which is thermodynamically consistent and based on a state-of-the-art parameterisation of the ideal-gas specific heat capacity of dry air from the National Institute of Standards and Technology (NIST). This reference potential temperature was compared to a potential temperature $\theta_{\mathrm{real}}$ wherein the real-gas behaviour of dry air is considered. In the range of temperatures from $180\,\mathrm{K}$ to $300\,\mathrm{K}$ and the range of pressures from $1000\,\mathrm{hPa}$ to $0.5\,\mathrm{hPa}$, covering the atmospheric conditions of roughly the entire troposphere and stratosphere, the relative differences between $\theta_{\mathrm{ref}}$ and $\theta_{\mathrm{real}}$ are smaller than $0.03\,\%$ and may be considered negligible. Consequently, $\theta_{\mathrm{ref}}$ even provides a reasonable approximation to the potential temperature of the real gas.

The difference between the newly derived reference potential temperature $\theta_{\mathrm{ref}}$ and the conventionally determined potential temperature $\theta_{c_p}$ (with constant $c_p = 1005\,\mathrm{J\,kg^{-1}K^{-1}}$, as recommended by the World Meteorological Organisation, WMO, 1966) increases with altitude, e.g., $\Delta\theta \geq 1\,\mathrm{K}$ at pressures $p \leq 60\,\mathrm{hPa}$.

Derivation of a potential temperature that is consistent with thermodynamics and that accounts for the ideal-gas properties of dry air requires the integration of Gibbs' equation and the subsequent solution of the resulting nonlinear equation. With a constant $c_p$, both analytical steps are straightforward, resulting in the conventional expression (13) as suggested by WMO (1966). However, if instead the temperature dependence of air's specific heat capacity $c_p(T)$ is considered, the integrals as

well as the equations are not analytically solvable and, thus, the solution must be approximated. Both approximations were performed and described in detail. The integral was treated with the basic approximation and the solution of the nonlinear equation was approximated by the second iterate of Newton's method. As an alternative to Newton's classical method, a modified formulation of Householder's iteration method is provided, featuring accelerated convergence properties.

The suggested approximation steps to obtain a reference potential temperature have two main sources of error: the error $\theta_{\text{ref}} - \theta_{\text{ref}}^{\text{approx}}$ inherent in the integral's basic approximation and the error $\theta_{\text{ref}}^{\text{approx}} - \theta^{(k)}$ of the $k$-th Newton iterate. The latter error approaches zero as $k \to \infty$, whereas the error resulting from the basic approximation remains well below $0.1\,\%$ (along the US Standard Atmosphere) for values of $\theta_{\text{ref}}$ of up to $\sim 2000\,\text{K}$, hence up to stratopause altitudes. To keep this low error level also for $\theta_{\text{ref}} > 2000\,\text{K}$, the approximation may require an extension by means of a higher-order polynomial.

One of the foremost implications of the re-assessed potential temperature's definition concerns the use of $\theta$ as a vertical co-ordinate for the sorting, grouping, and comparison of (measured) data, e.g., along or across isentropes. Thereby, the re-assessed potential temperature constitutes a more accurate consideration of the air's actual properties. This particularly concerns, e.g., the specific heat capacity which is conventionally assumed as constant and for which various values are given depending on the textbook consulted (offering a range from $1000\,\text{J}\,\text{kg}^{-1}\text{K}^{-1}$ to $1010\,\text{J}\,\text{kg}^{-1}\text{K}^{-1}$, see Table 1).

Significant errors and biases may arise if, for instance, the conventional derivation of $\theta$ (WMO, 1966) is used together with values for air's specific gas constant ($R_a$) or air's specific heat capacity ($c_p$) which better comply with the most recent state-of-knowledge. Moreover, the use of the standard pressure $1013.25\,\text{hPa}$ instead of $1000\,\text{hPa}$ as defined by WMO (1966) and consistently used herein as ground level pressure ($p_0$) may cause an additional deviation of the resulting $\theta$. Thus, the re-assessment of $\theta$'s definition could largely diminish such errors and biases and improve the comparability of data.

In addition to the vertical sorting of data, implications of the new reference potential temperature were discussed for several other applications in which the potential temperature is used. On the one hand, results may appear mostly unaffected by using $\theta_{\text{ref}}$ instead of the convential $\theta_{1005}$, such as the values of the Brunt-Väisälä frequency or the temperature change of air parcels due to diabatic heating. On the other hand, it was illustrated that any formula which involves the potential temperature needs to be carefully reviewed to see if its derivation relies on the assumed constancy of the specific heat capacity. If this is the case, substituting $\theta_{\text{ref}}$ for all occurrences of $\theta$ within the particular formula may lead to a wrong computation.

In contrast, examples were shown where the computation of Ertel's potential vorticity and the rate of change of potential temperature in response to diabatic heating yields different results by the use of $\theta_{\text{ref}}$ instead of $\theta_{1005}$. The differences increased with altitude, hence they become more important for applications within the stratosphere and above.

It should be emphasised that all these examples were based on assuming particular profiles of temperature and pressure together with other assumptions. Moreover, only a limited number of examples could be investigated, while the applications of potential temperature are numerous. Consequently, a well-founded, individual decision is required for each application of the potential temperature as to whether it is worth applying the more rigorous calculation in the particular context.

On the one hand, such a re-assessment could take into account the current state of knowledge regarding the accuracy of thermodynamic variables and substance-related properties. On the other hand, this way, the conceptional abstractness already inherent in $\theta$ is not further complicated by a misleading selection of parameters or reputed constants. There is no doubt

that the conventional method is suitable for the description of most processes occurring within the troposphere. However, at stratospheric or even mesospheric altitudes, the neglect of the temperature dependence of the ideal-gas heat capacity in the conventional definition increasingly distorts the resulting absolute values as well as the vertical course of the potential temperature. Ultimately, it seems obvious to profit from the computing capacities available today and from the known higher accuracy of physical variables and atmospheric parameters to carry out a reappraisal of the potential temperature, a useful (but not always consistently used) meteorological quantity.

**Appendix A: Derivation of the specific heat capacity from thermodynamics**

In the following, the derivation of the air's specific heat capacities $C_V, C_p$ (capital letters indicate molar units) at constant volume and pressure, respectively, is summarised, mainly following the textbook exposition by Kondepudi and Prigogine (1998). We start with the ideal gas law

$$pV = NRT, \tag{A1}$$

with $p$ the pressure, $V$ the volume of the system, $N$ the amount of gas within the volume, $T$ the temperature, and $R$ the universal gas constant. Additionally, the first law of thermodynamics is

$$dU = dQ - p\,dV, \tag{A2}$$

with the internal energy $U$ of the system and $dQ$ specifies the change of heat. Insertion of the total derivative of the internal energy $U$ in (A2), and assuming the system as thermodynamically closed, i.e., the molar amount $N$ remains conserved ($dN = 0$), leads to

$$dQ - p\,dV = \left.\frac{\partial U}{\partial T}\right|_{V,N} dT + \left.\frac{\partial U}{\partial V}\right|_{T,N} dV, \tag{A3}$$

and subsequently

$$dQ = \left.\frac{\partial U}{\partial T}\right|_{V,N} dT + \left(p + \left.\frac{\partial U}{\partial V}\right|_{T,N}\right) dV. \tag{A4}$$

If the system's volume is held constant, equation (A4) represents the definition of the constant-volume heat capacity $C_V$ in molar units, i.e.,

$$dQ = \left.\frac{\partial U}{\partial T}\right|_{V,N} dT = C_V(p,T)\,dT. \tag{A5}$$

Alternatively, assuming the system's pressure as constant, its volume is variable with total derivative

$$dV = \left.\frac{\partial V}{\partial T}\right|_{p,N} dT + \left.\frac{\partial V}{\partial p}\right|_{T,N} \underbrace{dp}_{=0} = \left.\frac{\partial V}{\partial T}\right|_{p,N} dT \tag{A6}$$

and, therefore

$$
\begin{aligned}
\mathrm{d}Q &= \left.\frac{\partial U}{\partial T}\right|_{V,N} \mathrm{d}T + \left(p + \left.\frac{\partial U}{\partial V}\right|_{T,N}\right) \mathrm{d}V \\
&= \left.\frac{\partial U}{\partial T}\right|_{V,N} \mathrm{d}T + \left(p + \left.\frac{\partial U}{\partial V}\right|_{T,N}\right)\left(\left.\frac{\partial V}{\partial T}\right|_{p,N} \mathrm{d}T\right) \\
&= \left[\left.\frac{\partial U}{\partial T}\right|_{V,N} + \left(p + \left.\frac{\partial U}{\partial V}\right|_{T,N}\right)\left.\frac{\partial V}{\partial T}\right|_{p,N}\right] \mathrm{d}T \\
&= C_p(p,T)\,\mathrm{d}T,
\end{aligned}
\tag{A7}
$$

defining the isobaric molar heat capacity $C_p$. In general, this quantity depends on pressure as well as on temperature. However, if the gas is assumed as ideal, an important conclusion from the statistical description of an ideal gas is the fact that the internal energy $U$ must be independent of the pressure (see, e.g., Fay, 1965).

Using this result, together with (A7) and the ideal gas law (A1), it follows

$$
\begin{aligned}
C_p &= \left.\frac{\partial U}{\partial T}\right|_{V,N} + \left(p + \left.\frac{\partial U}{\partial V}\right|_{T,N}\right)\left.\frac{\partial V}{\partial T}\right|_{p,N} \\
&= \left.\frac{\partial U}{\partial T}\right|_{V,N} + p \left.\frac{\partial V}{\partial T}\right|_{p,N} \\
&= \left.\frac{\partial U}{\partial T}\right|_{V,N} + \left.\frac{\partial}{\partial T}\left(pV\right)\right|_{p,N} \\
&= \left.\frac{\partial U}{\partial T}\right|_{V,N} + \left.\frac{\partial}{\partial T}\left(NRT\right)\right|_{p,N} \\
&= \left.\frac{\partial U}{\partial T}\right|_{V,N} + NR.
\end{aligned}
\tag{A8}
$$

In the previous computations, there is no restriction on the temperature dependence of the internal energy $U(T)$. Therefore, even by assuming ideal-gas behaviour, the specific heat capacity $C_p$ in (A8) is in general a function of temperature.

### Appendix B: Sensitivity of the conventional definition of $\theta$ to perturbations of $c_p$

This section explores, from a mathematical perspective, the sensitivity of the potential temperature formulation (13) based on a constant specific heat capacity. Considering the specific heat capacity $c_p$ as a variable, the sensitivity of $\theta_{c_p}$ (13) to a small perturbation $\delta$ of $c_p$ is described by its Taylor expansion

$$
\begin{aligned}
\theta_{c_p+\delta} &= \theta_{c_p} + \frac{\partial \theta_{c_p}}{\partial c_p}\delta + \mathcal{O}\left(\delta^2\right) \\
&= \theta_{c_p} - \theta_{c_p}\frac{R_a}{c_p^2}\ln\left(\frac{p_0}{p}\right)\delta + \mathcal{O}\left(\delta^2\right).
\end{aligned}
\tag{B1}
$$

For any constant value of the specific heat capacity $c_p$ and for a minor perturbation $\delta$, the second summand within the expansion (B1) remains small for small values of $\ln\left(\frac{p_0}{p}\right)$. If the interval between the two pressure levels is very narrow, i.e., $p \approx p_0$, the

expression $\ln\left(\frac{p_0}{p}\right)$ approximately equals $\ln(1) = 0$. Contrarily, if the pressure approaches very low values, i.e., $p \to 0\,\mathrm{Pa}$, the logarithmic expression diverges to negative infinity, i.e., $\ln\left(\frac{p_0}{p}\right) \to -\infty$, implying that the impact of the second summand intensifies with decreasing pressure, i.e., for increasing altitudes. Moreover, this may explain why the deviation between $\theta_{1000}$ and $\theta_{1010}$, as illustrated in Figure 2b, remains comparatively small within the troposphere and systematically increases with rising altitude, i.e., decreasing pressure levels.

### Appendix C: Approximate computation of the reference potential temperature

This section summarises the detailed steps of approximating the function $F(x)$, defined in (23), by $\widehat{F}(x)$, defined in (28) (Section C1), as well as the approximations of the solutions of the resulting nonlinear equations by Newton's method (Section C2).

### C1   Reformulating the function $F(x)$

Proceeding from the definition of a function $h(x)$

$$h(x) = \int_{T_1}^{x} \frac{c_p(T')}{T'}\,\mathrm{d}T', \tag{C1}$$

with $T_1 = 180\,\mathrm{K}$, the function $F(x)$ may be rearranged as

$$
\begin{aligned}
F(x) &= \int_{x}^{T} \frac{c_p(T')}{T'}\,\mathrm{d}T' - R_a \ln\left(\frac{p}{p_0}\right) \\
&= h(T) - h(x) - R_a \ln\left(\frac{p}{p_0}\right).
\end{aligned}
\tag{C2}
$$

The advantage of this reformulation of $F(x)$ is the inclusion of $h(x)$, consisting of an integral with fixed lower bound and a sole variable upper bound. This way, the function $h(x)$ is numerically solvable, and subsequently $h(x)$ can be substituted by an approximation $f(x)$ that is defined as

$$f(x) = b_0 + b_1 \ln(x - b_2) + b_3 x + b_4 x^2. \tag{C3}$$

Notably, if $c_p$ is constant, this function reduces to an exact primitive of the integrand $\frac{c_p}{T'}$ with $b_3 = b_4 = 0$. Moreover, in this case, the resulting root-finding problem $0 = F(x)$ is exactly solvable and finally leads to the known conventional definition (13) of the potential temperature.

As a further step, the function $h(x)$ is numerically approximated, while $c_p(T)$ in (C1) is replaced by the ideal-gas limit of air's specific heat capacity $c_p^0(T)$. The integration interval $[T_1, x]$ with $T_1 \le x \le 2000\,\mathrm{K}$ is traversed in steps of at most $0.001\,\mathrm{K}$ while each step of the integration process is carefully approximated by using Simpson's rule.

By solving a least-squares problem, the coefficients in (C3) for the approximation of $h(x)$ by the function $f(x)$ are estimated as

$$b_0 = -4072.2121328563667,$$

$$b_1 = 797.09247926609601,$$

$$b_2 = 29.587047521428016, \qquad\qquad\qquad\qquad\qquad\qquad\qquad\qquad\qquad\qquad\qquad (C4)$$

$$b_3 = 0.41981158226925142,$$

$$b_4 = -5.1008025097060311 \cdot 10^{-5}.$$

In Figure C1a the function $h(x)$ is graphed together with the approximation $f(x)$, as well as the respective deviations $h(x) - f(x)$ in Figure C1b. Evidently, the absolute error inherent to the approximations is comparatively small as, over the entire temperature range above $190\,\mathrm{K}$, the approximation error never exceeds $\pm 1\,\mathrm{J\,kg^{-1}K^{-1}}$. Exclusively at temperatures below $190\,\mathrm{K}$, the approximation error rapidly rises above $1\,\mathrm{J\,kg^{-1}K^{-1}}$, bearing in mind that such absolute temperatures are only occasionally found in the atmosphere within a relatively narrow altitude interval at the cold point tropopause. Moreover, the difference between $f(x)$ and $h(x)$ appears negligible as the profiles almost ideally coincide (cf. Figure C1a).

## C2 Finalised approximation of the reference potential temperature

As discussed in Section 5.1, the new formulation of the potential temperature based on the temperature-dependent specific heat capacity $c_p(T)$ requires solving the root-finding problem $0 = F(x)$, where the function $F(x)$ is defined in (23). However, since $F(x)$ contains an integral that complicates the root-finding process, this integral is substituted by the difference $f(T) - f(x)$, where $f$ is given in Section C1. Therefore, $F(x)$ is replaced by the function $\widehat{F}(x)$ as defined in (28) and the zero of the equation $0 = \widehat{F}(x)$ is denoted as $\theta_{\mathrm{ref}}^{\mathrm{approx}}$.

The equation $0 = \widehat{F}(x)$ is still not analytically solvable, so Newton's method is once more required. Using again $x_0 = \theta_{1005}$ as the initial guess, cf. (26), the iteration sequence for Newton's method is given by the recursion

$$x_{k+1} = x_k - \frac{\widehat{F}(x_k)}{\widehat{F}'(x_k)} = x_k - \frac{f(T) - f(x_k) - R_a \ln\left(\frac{p}{p_0}\right)}{-f'(x_k)}$$

$$\qquad\qquad = x_k - \frac{R_a \ln\left(\frac{p}{p_0}\right) - f(T) + f(x_k)}{f'(x_k)}. \qquad\qquad\qquad\qquad\qquad\qquad\qquad (C5)$$

Instead of this standard formulation of Newton's method (C5), Householder's formulation

$$x_{k+1} = x_k - \frac{\widehat{F}(x_k)}{\widehat{F}'(x_k)} - \frac{\widehat{F}''(x_k)}{2\widehat{F}'(x_k)}\left[\frac{\widehat{F}(x_k)}{\widehat{F}'(x_k)}\right]^2$$

$$\qquad = x_k - \frac{R_a \ln\left(\frac{p}{p_0}\right) - f(T) + f(x_k)}{f'(x_k)} \qquad\qquad\qquad\qquad\qquad\qquad\qquad (C6)$$

$$\qquad\quad - \frac{f''(x_k)}{2f'(x_k)}\left[\frac{R_a \ln\left(\frac{p}{p_0}\right) - f(T) + f(x_k)}{f'(x_k)}\right]^2$$

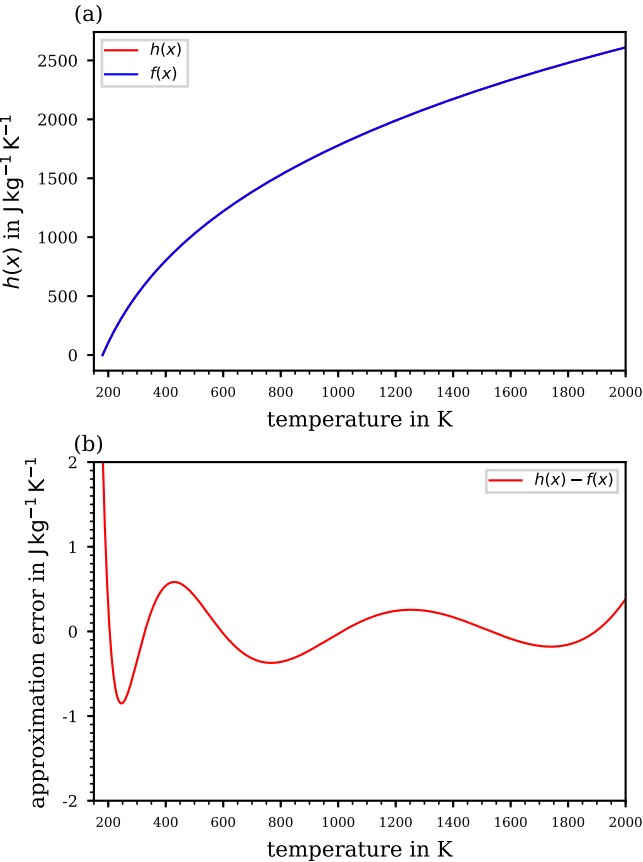

**Figure C1.** (a) Numerically evaluated function $h(x)$ together with its approximation $f(x)$; (b) the absolute approximation error $h(x) - f(x)$.

may be used, which allows for reducing the computation time due to its accelerated convergence speed. For completeness, the required derivatives $f'$, $f''$ in the recursion formulas (C5) and (C6) are

$$f'(x) = \frac{b_1}{x - b_2} + b_3 + 2b_4 x,$$

$$f''(x) = 2b_4 - \frac{b_1}{(x - b_2)^2}.$$

(C7)

The final step on the way to formulate a new expression for the potential temperature requires defining one of the iterates $x_k$ as appropriate enough for the approximations that result from applying the different methods:

– the standard of Newton's method (C5), simply referred to as Newton's method in the sequel, or

– Householder's method (C6).

While the mathematical expressions in (C5) and (C6) are of increasing complexity, the convergence rate of the approximating sequence increases with rising mathematical complication. The preferred method is determined by the accuracy required, i.e.,

| $z$ in m | $T$ in K | $p$ in Pa | $\theta_{\mathrm{ref}}$ in K | $\theta^{(1)}$ in K | $\theta^{(2)}$ in K | $\theta^{(1)}_{\mathrm{Householder}}$ in K |
|---|---|---|---|---|---|---|
| 5500 | 252.4 | 50506.8 | 306.837 | 307.016 | 307.016 | 307.016 |
| 11000 | 216.65 | 22632.1 | 331.337 | 331.510 | 331.510 | 331.510 |
| 20000 | 216.65 | 5474.89 | 494.940 | 495.376 | 495.378 | 495.378 |
| 32000 | 228.65 | 868.019 | 855.324 | 855.172 | 855.656 | 855.660 |
| 47000 | 270.65 | 110.906 | 1637.052 | 1620.463 | 1637.726 | 1638.974 |

**Table C1.** Values of the new reference potential temperature $\theta_{\mathrm{ref}}$, together with the first two iterates $\theta^{(1)}$, $\theta^{(2)}$ using Newton's method and the first iterate $\theta^{(1)}_{\mathrm{Householder}}$ using Householder's method for five pairs of temperature and pressure along the US Standard Atmosphere. The computed values are rounded to three digits.

better accuracy is necessarily associated with elevated computational effort for the approximation method. A discussion of the approximation errors is found in Appendix D.

Table C1 collects values of the new reference potential temperature $\theta_{\mathrm{ref}}$, together with the first two iterates $\theta^{(1)}$, $\theta^{(2)}$ using
Newton's method (C5) and the first iterate $\theta^{(1)}_{\mathrm{Householder}}$ using Householder's method (C6) for five pairs of temperature and pressure along the US Standard Atmosphere, cf. Figure 1, which allows verification of computations. The first height is chosen midway along the linearly decreasing temperature profile within the troposphere, while the other heights correspond to the kinks of the temperature profile.

## Appendix D:  Approximation error for the reference potential temperature

The following aims at a comprehensive investigation of the errors inherent with approximating the ultimate reference potential temperature $\theta_{\mathrm{ref}}$. As discussed in Section 5.2, the total error is a combination of the basic error $\theta_{\mathrm{ref}} - \theta_{\mathrm{ref}}^{\mathrm{approx}}$ and the approximation error that results from the approximation sequence $\theta_{\mathrm{ref}}^{\mathrm{approx}} - \theta^{(k)}$, where $\theta^{(k)}$ denotes the $k$-th iterate of the approximation sequence which is computed in accordance with either Newton's or Householder's method. The formulations of Newton's (C5) and Householder's (C6) method require replacing the function $F(x)$ by $\widehat{F}(x)$, and the approximation sequences
$\theta^{(k)}$ converge to $\theta_{\mathrm{ref}}^{\mathrm{approx}}$ for $k \to \infty$. Consequently, the approximation error $\theta_{\mathrm{ref}}^{\mathrm{approx}} - \theta^{(k)}$ tends to zero for $k \to \infty$.

The analysis of the approximation error is initially based on the pressure and temperature profiles of the US Standard Atmosphere. Figure D1 shows the total relative errors $\left(\theta_{\mathrm{ref}} - \theta^{(1)}\right)/\theta_{\mathrm{ref}}$ of the first iterate (Figure D1a) and $\left(\theta_{\mathrm{ref}} - \theta^{(2)}\right)/\theta_{\mathrm{ref}}$ of the second iterate (Figure D1b), computed with Newton's or Householder's method. The first iterate still causes the approximation to have significant errors, especially at altitudes above $35\,\mathrm{km}$. However, the second iterate with either Newton's or
Householder's method yields results with negligible approximation error. Hence, the total error of the approximation procedure is dominated by the unavoidable basic error, and may be deduced from the provided figures whenever the total error profile nearly congruently follows the profile of the basic error (cf. Figures D1b and 5a).

It may be noted that Householder's method achieves a significantly lower error level than Newton's method due to its accelerated rate of convergence. Compared to the first iterate approximations, computation up to the second iterate (cf. Figure

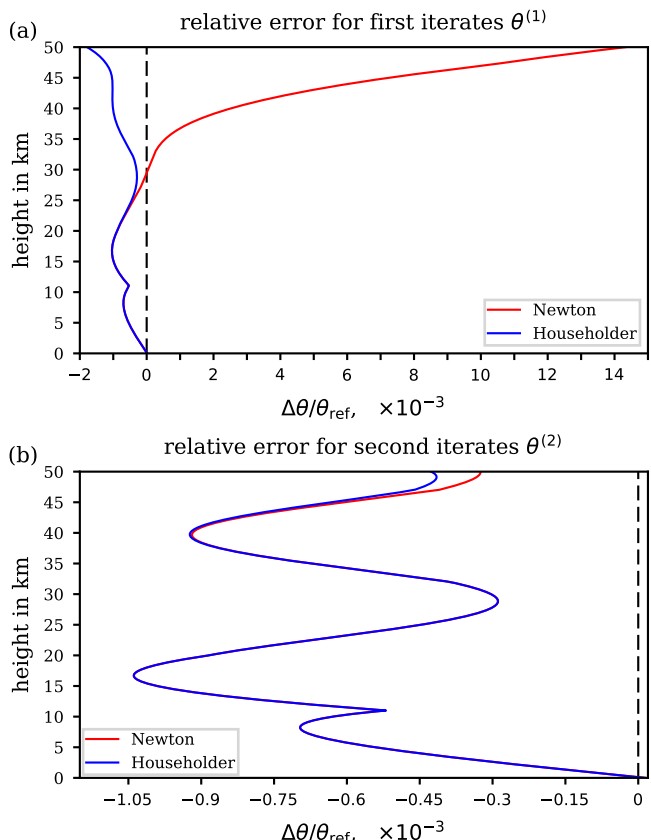

**Figure D1.** Total relative error along the US Standard Atmosphere arising from the iteration process by declaring (a) the first iterate $\theta^{(1)}$ or (b) the second iterate $\theta^{(2)}$ as the final approximation to the reference potential temperature $\theta_{\text{ref}}$. Red curves: iterates computed using Newton's method (C5); blue curves: iterates computed using Householder's method (C6). Note the different range of the abscissae.

D1b) achieves, in general, a considerable improvement for both methods, and both second iterate approximations approach the basic error quite closely (cf. Figure D1b). As is also evident from Figure D1b, compared to Householder's method, the second iterate with Newton's method results in a smaller total relative error $\left(\theta_{\text{ref}} - \theta^{(2)}\right)/\theta_{\text{ref}}$ relative to the ultimate reference potential temperature (indicated by a smaller distance to the dashed zero-line above $45\,\text{km}$ altitude). Nevertheless, the relative approximation error, $\left(\theta_{\text{ref}}^{\text{approx}} - \theta^{(2)}\right)/\theta_{\text{ref}}$, is larger compared to the second iterate with Householder's method. So, luckily, the second iterate with Newton's method provides a better approach to the reference potential temperature than that with Householder's method.

As with the discussion of the basic error in Section 5.2, the analysis of the total error should include all possible combinations of pressure and temperature in order to take into account fluctuations in the real atmosphere that deviate from the profile of the US Standard Atmosphere. Therefore, the extended analysis of the approximation error is summarised in Figure D2. The upper panels illustrate the total relative error of the second iterate for Newton's (Figure D2a) and Householder's method (Figure D2b).

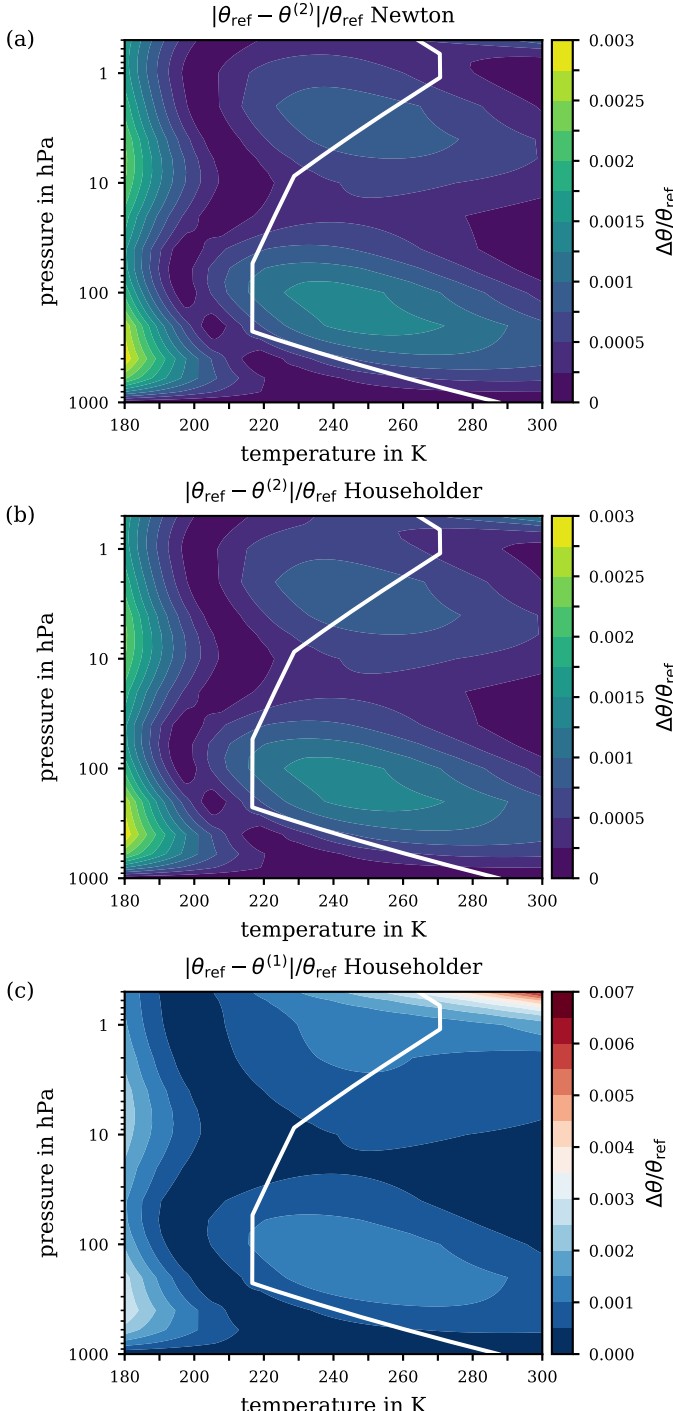

**Figure D2.** Relative error of the second iterates $\theta^{(2)}$ with (a) Newton's method and (b) Householder's method for the the ranges of pressure and temperature from $1000\,\text{hPa}$ to $0.5\,\text{hPa}$ and from $180\,\text{K}$ to $300\,\text{K}$, respectively. (c) The absolute error arising from the first iterate $\theta^{(1)}$ with Householder's method. The white solid line indicates the $p$-$T$-profile from the US Standard Atmosphere. Note the different ranges of the $\Delta\theta$ scales.

As previously shown, further iteration with either method does not improve the approximation quality. The contour patterns in these panels show a remarkable similarity to the contours for the relative error of the basic approximation in Figure 5c. Also here (upper panels of Figure D2), two regions are highlighted by the contours, i.e., at $\sim 100\,\text{hPa}$ and in a pressure range from $\sim 5\,\text{hPa}$ to $1\,\text{hPa}$, featuring the same impact on $\Delta\theta/\theta_{\text{ref}}$ of identical strength as the basic error. This result may not be surprising, since the second iteration step with both methods, Newton's and Householder's, was already proven to approach the approximation comparatively well, without worsening the total error level (cf. Figure D1b).

Consequently, concerning the required number of iterations and the method to use, the second iteration of Newton's method can be recommended to deliver appropriate results, with a relative error of less than $0.3\,\%$, up to the stratopause level ($\sim 50\,\text{km}$). Householder's method features an accelerated convergence rate, and its use up to its first iterate $\theta^{(1)}$ may be already appropriate for certain applications. According to the total error of Householder's method up to its first iterate $\theta^{(1)}$ (Figure D2c), the resulting relative error remains below $7\,\%$ to a pressure level of $\sim 50\,\text{hPa}$ and $\Delta\theta$ stays below $0.3\,\%$ to pressures of $\sim 2\,\text{hPa}$. Thus, Figure D2 may serve as guidance to decide how many iterations with one or the other method best meets the individual accuracy requirements.

## Appendix E: The derivative of the reference potential temperature

As discussed in Section 5.1, the new reference potential temperature is defined as the zero of the function

$$F(x, p, T) = \int_{x}^{T} \frac{c_p(T')}{T'}\,\mathrm{d}T' - R_a \ln\left(\frac{p}{p_0}\right) \tag{E1}$$

for given values of pressure $p$ and temperature $T$, see Equation (23). More precisely, for varying $p, T$, a function $(p, T) \mapsto \theta_{\text{ref}}(p, T)$ is implicitly defined by the equation

$$F(\theta_{\text{ref}}(p, T), p, T) = 0. \tag{E2}$$

According to the implicit function theorem (e.g., Protter and Morrey, 1985, chapter 7), equation (E2) is uniquely solvable for $\theta_{\text{ref}}(p, T)$, i.e., the function $(p, T) \mapsto \theta_{\text{ref}}(p, T)$ actually exists as a differentiable function of $(p, T)$, if the condition $\frac{\partial F}{\partial \theta} \neq 0$ holds. According to (E1), this partial derivative equals

$$\frac{\partial F}{\partial \theta}(\theta_{\text{ref}}, p, T) = -\frac{c_p(\theta_{\text{ref}})}{\theta_{\text{ref}}}, \tag{E3}$$

being strictly negative, since the specific heat capacity is always positive. Moreover, the implicit function theorem states that the derivatives of the implicit function $\theta_{\mathrm{ref}}(p, T)$ are given by

$$
\begin{aligned}
&\left[\frac{\partial \theta_{\mathrm{ref}}}{\partial p}(p, T), \frac{\partial \theta_{\mathrm{ref}}}{\partial T}(p, T)\right] \\
&= -\left(\frac{\partial F}{\partial \theta}(\theta_{\mathrm{ref}}, p, T)\right)^{-1}\left[\frac{\partial F}{\partial p}(\theta_{\mathrm{ref}}, p, T), \frac{\partial F}{\partial T}(\theta_{\mathrm{ref}}, p, T)\right] \\
&= \frac{\theta_{\mathrm{ref}}}{c_p(\theta_{\mathrm{ref}})}\left[-\frac{R_a}{p}, \frac{c_p(T)}{T}\right] \\
&= \left[-\frac{R_a}{c_p(\theta_{\mathrm{ref}})}\frac{\theta_{\mathrm{ref}}}{p}, \frac{\theta_{\mathrm{ref}}}{T}\frac{c_p(T)}{c_p(\theta_{\mathrm{ref}})}\right].
\end{aligned}
\tag{E4}
$$

Note, these partial derivatives coincide with the partial derivatives of $\theta_{c_p}$ in the case of a constant specific heat capacity. Using the partial derivatives (E4), the total differential of $\theta_{\mathrm{ref}}$ may be written as

$$
\begin{aligned}
\mathrm{d}\theta_{\mathrm{ref}} &= \frac{\partial \theta_{\mathrm{ref}}}{\partial p}\,\mathrm{d}p + \frac{\partial \theta_{\mathrm{ref}}}{\partial T}\,\mathrm{d}T \\
&= -\frac{R_a}{c_p(\theta_{\mathrm{ref}})}\frac{\theta_{\mathrm{ref}}}{p}\,\mathrm{d}p + \frac{\theta_{\mathrm{ref}}}{T}\frac{c_p(T)}{c_p(\theta_{\mathrm{ref}})}\,\mathrm{d}T
\end{aligned}
\tag{E5}
$$

or

$$
c_p(\theta_{\mathrm{ref}})\frac{\mathrm{d}\theta_{\mathrm{ref}}}{\theta_{\mathrm{ref}}} = c_p(T)\frac{\mathrm{d}T}{T} - R_a\frac{\mathrm{d}p}{p}.
\tag{E6}
$$

*Author contributions.* MB, RW, and PS conceived, designed, and carried out the main part of the research. UA contributed the URAP data and implications on breaking heights of gravity waves. AH gave advice about the heat capacity and performed the calculations of real-gas potential temperatures. FP contributed the implications on diabatic heating. MB and RW wrote the manuscript with contributions and reviews from all authors.

*Competing interests.* The authors declare that they have no conflict of interest.

*Acknowledgements.* We thank Eric W. Lemmon for his advice on the equation of state of dry air, Vera Bense for fruitful discussions on gravity wave breaking, Gergely Bölöni for providing us the URAP vertical profiles in Section 7, Miklós Szakáll for the hint concerning Titan's atmosphere, and Heini Wernli for his interest in an early version of the draft. We thank Pascal Marquet and an anonymous reviewer for their very valuable comments and hints which led to a significantly improved manuscript. Additionally, we gratefully acknowledge the contributions of Timothy Garret as editor in the review process of this article. Manuel Baumgartner and Peter Spichtinger acknowledge support by the Deutsche Forschungsgemeinschaft (DFG) within the Transregional Collaborative Research Centre TRR165 Waves to Weather, (www.wavestoweather.de), projects B7 and Z2. Ralf Weigel received financial support by the *Bundesministerium für Bildung und Forschung* (BMBF) under the joint ROMIC-project *SPITFIRE* (01LG1205A). Ulrich Achatz and Peter Spichtinger acknowledge partial support by the DFG through the research unit Multiscale Dynamics of Gravity Waves (MS-GWaves) and through grants AC 71/12-2 and SP 1163/5-2.

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
