# Peer review of "Reappraising the appropriate calculation of a common meteorological quantity: Potential Temperature"

_Atmospheric Chemistry and Physics, 2020_

## Referee Comment (RC1) · Pascal MARQUET (Referee) · 2 Jun 2020

**Review by Pascal Marquet of the paper "acp_2020_361"**

entitled: *Reappraising the appropriate calculation of a common meteorological quantity: Potential Temperature.*

by Manuel Baumgartner, Ralf Weigel, Ulrich Achatz, Allan H. Harvey, and Peter Spichtinger.

**1    General Major Comments / Recommendations**

The paper of Baumgartner, Weigel, Achatz, Harvey and Spichtinger submitted to the Atmospheric Chemistry and Physics examines the impact of temperature variation in specific heat capacity $c_p(T)$ on the calculation of the potential temperature $\theta$ and entropy of dry air.

The authors show that, through integration and a cumulative effect, the impacts of $c_p(T)$ on $\theta$ appear to be significant above 10 to 20 km height. The authors show that modified calculations of $\theta$ for dry air can induce "non-negligible" differences in predicting the altitude of gravity wave breaking, "although not excessive".

It is undeniable that the hypothesis of constant values of $c_p$ for dry air and water vapour is only a first approximation that deserves to be studied further, even if the variations of $c_p$ with temperature is by far greater for the liquid and solid phases of water.

It seems to me, however, that the authors should comment on and/or answer a series of questions that arise on reading their article.

**(1)** – the authors present in section 3 a range of possible dry-air values of $c_p$ that appear to be greatly exaggerated, ranging from 994 to 1011 J/K/kg. I show in this review that the uncertainty interval must be much smaller (1004.5 to 1007.5 J/K/kg), which must imply impacts on values of $\theta$ about 7 times smaller than those considered at high altitude in the document. The authors should modify sections 2 to 5 and Figures 2, 3 and 4, by reducing the uncertainty on $c_p$ and by retaining only the more recent and realistic values.

**(2)** – I show from copies of previously published papers, tables and figures that the observed values of $c_p(T)$ for $T < 320$ K contradict values above 1007.5 J/K/kg, those under 1004.5 J/K/kg and the (ideal gas) formulations of Lemmon et al. (2000) and Dixon (2007) considered in section 4 by the authors. Observed values of $c_p(T)$ for $T < 320$ K are rather consistent with the (real gas) NIST-REFPROP formulation considered in section 6 and with the IAPWS-TEOS10 formulation.

**(3)** – In this sense, the approach followed by the authors to calculate first values of $\theta_{\mathrm{ref}}$ from the ideal-gas formulation of $c_p(T)$ by Lemmon et al (2000), and then those of $\theta_{\mathrm{real}}$ for the real-gas NIST-REFPROP formulation, seems attractive, with however a comparison to irrelevant and too extreme constant values of 1011 and 994 J/K/kg in Figure 4 of the paper.

**(4)** – Moreover the results of your section 6 seem strange to me, because the comparison of $\theta_{\mathrm{ref}}$ deduced from the ideal-gas Lemmon's formulation (purple curve in your figure 3) with $\theta_{\mathrm{real}}$ deduced from the real-gas NIST-REFPROP's formulation (yellow discs in your figure

3) gives very small differences on figures 8. Indeed, the differences $\theta_{\text{real}} - \theta_{\text{ref}}$ of less than 0.05 K for $\theta > 700$ K above 20 km (less than 0.007%) seem unrealistic and not consistent with differences of 4.5 J/K/kg (or 4.5 %) for $c_p(T)$ at 200 K, 2.8 J/K/kg (or 2.8%) at 250K and 1.3 J/K/kg (or 1.3 %) at 300 K (values deduced from the yellow discs and the purple curve in your figure 3).

I guess that the relative differences $(\theta_{\text{real}} - \theta_{\text{ref}})/\theta_{\text{ref}}$ should be of the order of a few percent above 25 km and should increase with height, as indicated by a rough analysis of the differences between curves of your Figure 4b (to be checked by you, however, from direct computations and/or from a version with a linear scale of your figure 4b).

Differences of several percent between ideal-gas and real-gas formulations of $c_p(T)$ should lead to larger differences in the gap between $\theta_{\text{real}}$ and $\theta_{\text{ref}}$. This should result in a likely change in the conclusion in your section 6 and the use of formulations from IAPWS-TEOS10 (free) or INIST-REFPROP (to buy), rather than the analytical formula of Lemmon et al (2000, Eq.18, page 345) that is contradict by the values of $c_p(T)$ published in Table A2 (pages 366-367) of the same paper (see Fig.9 in section 3 bellow)

(5) – In fact, after reflection and analysis of this aspect (4), this is probably a false problem. Indeed, everything seems to be explained by the fact that the major differences for your $\theta$ come from values of $c_p$ for highest $T$ temperatures, say between 400 K and 2000 K. This aspect is not documented in your figure 3, where the values of $c_p$ are only plotted up to 485 K.

The fact that the values of your $\theta_{\text{real}}$ and $\theta_{\text{ref}}$ are very close must be explained by a low sensitivity of your $\theta$ to values of $c_p$ for ambient temperatures (let's say those ranging from 200 K to 320 K and which define how the physical parameterizations should influence the weather parameters), with, on the other hand, a strong sensitivity of your $\theta$ to values of $c_p$ for temperatures above 400 K (temperatures that are not observed in the real atmosphere but that intervene numerically in the calculation of your $\theta$ when passing from high altitudes where the pressure is very low and returning adiabatically towards the ground level through very high artificial temperatures).

Therefore, if you are interested in the values of $\theta$ calculated by an adiabatic evolution from a very low pressure $p$ to a (surface) pressure $p_0 = 1000$ hPa, you should better describe the accuracy of the values of $c_p(T)$ for $T > 400$ K.

(6) – Another aspect should be addressed in this article. One of the goals of our community is to provide efficient and applicable numerical methods for climate and numerical weather prediction models. In this sense, it would be useful to quantify the iterative processes designed and tested in this article: what is the extra cost (in CPU) for the calculation of $\theta_{\text{ref}}$ and $\theta_{\text{real}}$ compared to the direct calculation $\theta_{c_p}$ for a constant $c_p$? (make this evaluation for example for a set of vertical columns of standard atmosphere)

(7) – For me, the most problematic aspect concerns the application you chose in section 7, by assuming that the squared Brunt-Väisälä frequency could be

$$N^2 = \frac{g}{\theta} \frac{\partial \theta}{\partial z},$$
(1)

where $\theta$ would be calculated by the particle method (by an adiabatic evolution from a very low pressure $p$ to a surface pressure $p_0 = 1000$ hPa).

Differently, we recalled in Marquet and Geleyn (2013, MG13) that $N^2$ should be calculated from the local gradients of basic meteorological parameters (temperature and pressure if dry air is considered), and not from the variable $\theta$ that you study in your article (by an adiabatic evolution from a very low pressure $p$ to a surface pressure $p_0 = 1000$ hPa).

In fact $N^2$ corresponds to adiabatic fluctuations of the density, before anything else. Accordingly, equations (B2) and (1) of MG13 applied to dry air give the corresponding expression of $N^2$ as a function of local vertical gradients of density ($\rho$) and specific entropy ($s$):

$$N^2 \; = \; \frac{g}{\rho} \; \left.\frac{\partial \rho}{\partial z}\right|_s \; - \; \frac{g}{\rho} \; \frac{\partial \rho}{\partial z} \; = \; \left( - \; \frac{g}{\rho} \; \left.\frac{\partial \rho}{\partial s}\right|_p \right) \; \frac{\partial s}{\partial z} \; ,$$

where the first vertical derivative (of density with respect to $z$) is computed at constant entropy and the second vertical derivative (of density with respect to $s$) is computed at constant pressure. The local state equation $p = \rho\,R\,T$ and $\rho = p/(R\,T)$ with constant $R$ and $p$ implies

$$\left.\frac{\partial \rho}{\partial s}\right|_p \; = \; - \; \frac{\rho}{T} \; \left.\frac{\partial T}{\partial s}\right|_p \; .$$

The dry-air Gibbs equation writes $T\,ds \; = \; dh - dp/\rho$, with $dh = c_p(T)\,dT$ and with possibly $c_p(T)$ depending on absolute temperature. For constant pressure, this Gibbs equation reduces to $T \; \left. ds\right|_p \; = \; c_p(T) \; \left. dT\right|_p$, leading to $dT/\left. ds\right|_p \; = \; T/c_p(T)$, and thus to $d\rho/\left. ds\right|_p \; = \; - \; \rho/c_p(T)$. The squared dry-air Brunt-Väisälä frequency is therefore equal to

$$N^2 \; = \; \frac{g}{c_p(T)} \; \frac{\partial s}{\partial z} \; . \tag{2}$$

The dry-air Gibbs equation can then be used again to write $T\,\partial s/\partial z \; = \; c_p(T)\,\partial T/\partial z \; - \; (1/\rho)\partial p/\partial z$ which is valid for vertical oscillations. If moreover hydrostatic conditions prevail, then $\partial p/\partial z = -\rho\,g$, leading to

$$\boxed{N^2 \; = \; \frac{g}{T} \; \left( \frac{\partial T}{\partial z} \; + \; \frac{g}{c_p(T)} \right)} \; . \tag{3}$$

This equation corresponds to the dry-air version of (22) in MG13, and it is Equation (1a) in the previous famous paper of Durran and Klemp (1982) about computations of the Brunt-Väisälä frequency (a key paper that you do not cite).

An expected result is that $N^2 = 0$ for the dry-air adiabatic lapse rate $\partial T/\partial z = - g/c_p(T)$.

The important finding for your study is that there is no need to use the gradient of any potential temperature for computing $N^2$. Really, only the vertical gradient of $T$ has to be calculated in (3), where it is possible to take into account the variations of $c_p(T)$ with the temperature you want to study in your paper. It is thus "possible", but not "mandatory", to use (2) and a possible entropy formulation $s = c_p \ln(\theta) + const$ for the entropy to get the form (1) $N^2 \; = \; (g/\theta)\,\partial\theta/\partial z$ you have considered in your paper, but if and only if $c_p$ is a constant. And this is not possible if $c_p(T)$ depends on the temperature, with in this case the need to stick with the formulation (3) recalled above in terms of the gradient $\partial T/\partial z$.

The other important result here is that it is the local temperature that is involved in $c_p(T)$, so those between 1004.5 J/K/kg and 1007.5 J/K/kg for 200 K $< T <$ 320 K, and especially not the ones at the higher temperatures that you studied in your paper to calculate $\theta_{\rm ref}$ or $\theta_{\rm real}$, which are not needed for computing $N^2$ by (3).

It therefore seems to me that the application described in your section 7 is inaccurate, since the formulation (1) that you use for $N^2$ is not the right one (3). If so, can you show another application where values of your formulation of $\theta_{\text{ref}}$ or $\theta_{\text{real}}$ would intervene in meteorological science?

**(8)** – My recommendation is that the document deserves acceptance only if the impacts described in section 7 concerning gravity waves are real.

Therefore, the authors must provide evidence that it is indeed their formulations of $\theta_{\text{ref}}$ or $\theta_{\text{real}}$ (obtained by an adiabatic evolution between the pressures $p$ and $p_0$) that intervenes in the Brunt-Väisälä frequency formula, and not the local vertical gradients of temperature and pressure derived in Durran and Klemp (1982) and Marquet and Geleyn (2013).

If the authors can provide this evidence, then their paper would merit to be published subject to taking all the major recommendations and specific comments into account, or explaining why they do not need to take them into account.

**2  Specific Comments**

– **Line 1:** add *dry* in: "... it is conserved for *dry* air's adiabatic ..."

– **Lines 10 to 22:** I do not have access to Wegener's book (1911) and I confess that I was not aware of Köppen's oral contribution (1888). I have cited only the contributions of von Helmholtz and von Bezold in my papers (Marquet 2011, 2017, 2019b, Marquet and Dauhut 2018). I have been able to verify, however, Kutzbach's sentence (1979, page 143) in which Köppen's (1988) oral contribution is mentioned (see the excerpts in the Figure 1 in section 3 bellow). However, the title of the 1888 lecture of Köppen is written in Kutzbach (1979) as: "Ueber die Luftmischung und potentielle Temperatur", which might be different from the one in your bibliography: *"Über Luftmischung..."*? Moreover, I have not found the paper (or a copy of this lecture) of Köppen: do you have a copy of this lecture, or are you just citing the sentence of Kutzbach? Finally, I do not understand why you cite the URL: `http://snowcrystals.com/`?

– **Lines 10 to 28:** It would be useful to refer to the papers by Poisson (1833) and Thomson (1862-65) who had clearly imagined, before von Helmoltz and von Bezold in 1888, this idea of adiabatic variation on the vertical and the calculation of temperature for an air particle brought back to the surface (see section 5 of Marquet and Dauhut, 2018, and Marquet 2019b). I give copies of these articles on Figures 2 and 3.

– **Lines 10 to 28:** It would be useful to refer to Bauer's paper (1908-1910), where the link between entropy and the potential temperature of dry air is made for the first time (see citations in Marquet 2011, Marquet and Dauhut 2018 and Marquet 2019b). I give copies of this paper of Bauer on Figures 4 and 5

– **Lines 53 to 61:** You should mention the basic references for the definition and the use of $PV(\theta)$: Ertel (1940) and Hoskins (1987) at least (see also Schubert et al. 2004 cited in Marquet 2014).

– **Lines 65-67:** The studies of the moist-air entropy by Hauf and Höller (1987) and Marquet (2011) do not start from the Gibbs' equation "$Tds = dh - dp/\rho - \sum_n \mu_n dq_n$". On the contrary, they start from the moist-air entropy $s = \sum_n q_n s_n$ expressed as the weighted sum of the entropies $s_n$ for its $n = 0, ..., 3$ constituents (dry air, water vapour, liquid water and ice) with concentrations $q_n$ (specific contents).

– **Lines 68-69:** The assumption of "local equilibrium" and use of latent heats release (of vaporization $L_v$ and sublimation $L_s$) are also included in the definition of Hauf and Höller (1987) and Marquet (2011), not only in the formulation of Emanuel (1994).

– **Lines 69-70:** It is not true that "*These formulations always rely on the assumption of reversible processes (i.e. conserved entropy)*". On the contrary, the formulation $s(\theta_s)$ of Marquet (2011, ...) makes it possible to measure and quantify the losses or increases in moist-air entropy associated with irreversible processes such as the removal of precipitations that you mention. See in particular Eq.(59) in Marquet and Geleyn (2015), where the change in moist-air entropy associated with pseudo-adiabatic (von Bezold, 1888) processes writes:

$$ds \;=\; c_{pd} \, \frac{d\theta_s}{\theta_s} \;=\; (s - s_l) \left( \frac{-\, dr_{sw}}{1 + r_{sw}} \right) .$$

– **Lines 82-83:** You say: "*the potential temperature is commonly used as a prognostic variable in numerical models for the formulation of the energy equation*". Could you explain in which models $\theta$ is used as a prognostic variable? As far as I know, the prognostic variables associated with energy is either the temperature $T$ or the combination $c_p T$, with the moist-air definition for $c_p$. In particular, your reference to Richardson et al (2007) on line 105 seems incorrect, since page 25 of this article the equations are: "$DT/Dt = F_q$" or "$\partial T/\partial t = ... + F_q$" or "$\partial(\rho\, T)/\partial t = ... + \rho\, F_q$".

– **Lines 105:** You say: "*it was pointed out by Li and Chen (2019) that this approach could suffer from not accounting for the temperature dependence of the isobaric specific heat capacity $c_p$ of the respective atmospheres gas composition*". I spent some time checking this out in Li and Chen (2019), and I find (page 2): "*Furthermore, the expressions of potential temperature and equivalent potential temperature become complicated when the heat capacity of the atmosphere varies with temperature or when multiple condensing species exist in the atmosphere.* Here as elsewhere, could you quote the pages and/or equations corresponding to your citations, to help the reader find his way around in articles or books with very many pages?

– **Lines 117:** It is customary, at the end of the introduction, to present the outline of the article, with a summary of the content of each forthcoming sections. This should be included at the end of your Section 1.

– **Lines 134:** Your value for $R_a = R/M_{\mathrm{mol},a}$ is known with $R$ given up to $\pm 0.0001$ I believe? You could retain the value 8.31446 for example? Anyhow you have to give the resulting value $R_a = 287.115$ at least, with perhaps the associated precision $\pm 0.005$?

– **Lines 183:** It was indeed indicating by WMO that the variability of $c_p$ ranges from 994 J/K/kg to 1011 J/K/kg. But the real recommendation is rather a value close to 1005 J/K/kg, in line with the values presently used in most General Circulation (GCM) and Numerical Weather Prediction (NWP) models:

| $c_p$ (J/K/kg) | GCM and/or NWP models |
|---|---|
| 1005.0 | Unified-Model (UKMO, UK, from Adrian Lock) |
| 1005.0 | COSMO (DWD, Germany, from Dmitrii Mironov) |
| 1004.7 | IFS (ECMWF, Reading, UK, from "sucst.F90") |
| 1004.7 | ARPEGE (Meteo-France, Toulouse, France, from "sucst.F90") |
| 1004.7 | AROME (Meteo-France, Toulouse, France, from "sucst.F90") |
| 1004.7 | Meso-NH (L.A.+Meteo-France, Toulouse, France, from "sucst.F90") |
| 1004.7 | LMD-Z (IPSL, Paris, France, from "suphec.F90") |
| 1004.6 | ICON (DWD, Germany, from Dmitrii Mironov) |
| 1004.6 | GFS (USA, from "physcons.f") |

– **Lines 191-192:** These old WMO values of 994 J/K/kg and 1011 J/K/kg are too extreme and unrealistic, because they are not used in any current GCM and NWP model. Or could you indicate the models where these values might be used?

– **Page 8, Table 1:** The values of 994 J/K/kg, 1000 J/K/kg, 1003 J/K/kg and 1011 J/K/kg do not seem relevant.

- The value 994 J/K/kg comes from an old book I couldn't find, and the accuracy of the data obtained before 1933 can be questioned. This is like the measurement of the speed of light, the accuracy of which cannot be the "meeting of all possibilities", including for example the measurements of Romer and Huygens in 1675 (220,000 km/s), Bradley in 1729 (301,000 km/s), Fizeau in 1849 (315,000 km/s) or Foucault in 1862 (298,000 km/s)? It is the same for the measurement of the numerical values of $\gamma = c_p/c_v$ for diatomic gases, where the value of 1.421 retained by Poisson in 1833 or of 1.41 by Thomson in 1862 cannot be compared with the modern value of $7/5 = 1.40$? It is the same for the measurement of absolute scale of temperature, with a constant corresponding to 267 K in Gay-Lussac (1802) and Carnot (1824), to 273.22 K in Thomson (1848), before to be presently fixed to 273.15 K (see the review in Marquet, 2019a).
- The value of 1000 J/K/kg attributed to Valis (2009) seems to be easily questionable: see the legend in Figure 6 in section 3 bellow.
- I don't know where the value of 1003 J/K/kg published in Tripoli and Cotton (1981) comes from. But one can also have doubts about their values of $c_p$ for ice (2100 J/K/kg instead of 2106 J/K/kg) and liquid water (4187 J/K/kg instead of 4218 J/K/kg), with important differences for both dry air, liquid water and ice from the values commonly used in GCM and NWP model.
- Other than the mention in the WMO recommendations, I have never seen an application of the value 1011 J/K/kg. Could you indicate such an application of the value $c_p = 1011$ J/K/kg for dry air?

– **Lines 194-199 and Figure 2:** Assuming new extreme values of 1004.5 J/K/kg and 1007.5 J/K/kg (later demonstrated), I was able to redo your figures 2 (a) and (b) with the same US standard atmosphere profile: see Figure 7 in section 3 bellow, with indeed the same difference $\Delta\theta_{c_p} = \theta_{994} - \theta_{1011}$ (in black) as in your paper. The new differences $\theta_{1004.5} - \theta_{1007.5}$ (in red) are much smaller, by an order of magnitude or so (divided by a factor of about 5 to 7). The new differences are less than 2 K at 20 km, 5 K at 35 km and 14 K (instead of 75 K) at 50 km. These new differences $\theta_{1004.5} - \theta_{1007.5}$ may modified your comments and conclusions in your section 3.

– **Section 4, Lines 226-260 and Figure 3:** I disagree with many of the points you've drawn on your figure 3. So I redid your figure 3 by deleting the old and questionable data (see Figure 8 in section 3 bellow). I kept the values of 1004 J/K/kg, 1004.832 J/K/kg, 1005 J/K/kg and 1005.7 J/K/kg, the data of Vassermann et al (1966) and NIST-REFPROP as well as the two curves of Lemmon et al (2000) and Dixon (2007). This new figure shows that constant values of $c_p$ between 1004.5 J/K/kg and 1007.5 J/K/kg agree with the selected points for the range of temperatures observed in the atmosphere (say 200 to 320 K). The two curves of Lemmon et al (2000) and Dixon (2007) are retained here because they are valid for the approximation of ideal gases and allow to measure the differences with formulations for real gases, such as Vassermann et al (1966) and NIST-REFPROP. The impact of real gases properties on $c_p$ increases with decreasing values of $T$ bellow 260 K, and is larger than 4 J/K/kg at 200 K.

– **Lines 245, legend of Fig 3, lines 315-321 and Eqs.(18) and (19):** It should be mentioned that your formula (18) with the coefficients (19) of Lemmon et al (2000) disagrees with the observed values given in Table A2 of the same article Lemmon et al (2000). And indeed, while formula (18) leads to decreasing values of $c_p(T)$ for decreasing $T$, the values of $c_p(T)$ in Table A2 show a minimum around 250 K and become increasing for decreasing temperatures up to 81.72 K (see Figs.9 and 10 in section 3 bellow). It should also be mentioned that your equation (18) corresponds to equation (18) (page 345) in Lemmon et al (2000).

– **Lines 246 and 325-329:** You should mention that the equation of Dixon (2007, p.376) used to compute the dry-air value $c_p(T)$ plotted in your Fig.3 is
$$c_p(T) = 1002.5 + 275.\,10^{-6}\,(T - 200)^2 \text{ J/K/kg}$$
(see Figure 11 in section 3 bellow).

– **Lines 356 / Eq.(11):** The gaz constant "$R_a$" is missing before the integral $\int_{p_0}^{p} dp'/p'$

– **Lines 356 / Eq.(21), Lines 359 / Eq.(22), Lines 374 / Eq.(23), Lines 411 / Eq.(25), Line 426 and 429, Line 691 / Eq.(C1), Line 693 / Eq.(C2):** You should used the same dummy variable "$T'$" as in your Eq.(11) line 153 ($\int_{T_0}^{T} dT'/T'$) to write all the integrals of the kind $\int_{\theta}^{T} c_p(T')dT'/T'$. The use of the dummy variable "$z$" can lead to unfortunate confusion with the altitude variable, which is then used in the rest of your paper to describe the true vertical coordinate.

– **Page 11 / Fig 3:** I have plotted in Figure 12 (top, see section 3 bellow) the equivalent of your Figure 3, but with different formulations that correspond to observed ("real gases") values of $c_p(T)$, with a zoom (Figure 12 bottom) around the usual atmospheric temperatures.

I first reported (from your Fig.3) the points of your calculations made with the (paid) application of NIST-REFPROP. These NIST-REFPROP values are comparable to those I have computed with the (free) SIA software (http://www.teos-10.org/software.htm) corresponding to the IAPWS-2010 (Feistel et al, 2010) and TEOS-10 (Feistel, 2018) formulations. There is a similar minimum $c_p(T) \approx 1005.5$ to 1005.7 J/K/kg at around 250 K and with the same higher values of about 1007 J/K/kg at 320 K and 1006.7 J/K/kg at 200 K.

The values published in Table A2 of Lemmon et al. (2000) are fairly comparable to those of NIST-REFPROP and IAPWS-TEOS10, with a similar minimum of $c_p(T)$ at around 250 K.

The same applies to the values of $c_p(T)$ for N2 and O2 published in Marquet (2015), with the values for dry air completed with the values of $c_p(T)$ for Argon.

The minimum of $c_p(T)$ for N2 is at around 290 K in both Stewart and Jacobsen (1989, Table 5.73, see Fig.13 bellow) and Span et al. (2000, page 1410, see Fig.14 bellow). The resulting figure 5 for N2 published in Marquet (2015) is recalled in Fig.15 bellow.

The minimum of $c_p(T)$ for O2 is at around 220 K in Jacobsen et al. (1997, Table 5.79, see Figs.16 and 17). The resulting figure 4 for O2 published in Marquet (2015) is recalled in Fig.18 bellow.

Values of $c_p(T)$ for Argon increases for decreasing $T$ for both Tegeler et al. (1999, Table 34, see Fig.19) and Stewart and Jacobsen (1989, Table 15, Figs.20 and 21). The unpublished Fig.22 plotted bellow shows that values for Tegeler et al. (1999) and Stewart and Jacobsen (1989) fairly coincide for $150 < T < 300$ K.

The unpublished Fig.23 bellow shows that it is equivalent to use $c_p(T)$ computed for N2, O2 and H20 vapour by using Statistical and Quantum Physics (dashed lines) or by the "calorimetric method" (third law and integration of $c_p(T')/T'$ from 0 K to $T$, sum of $L(T_k)/T_k$ for all changes of phases at $T_k$, add the Pauling-Nagle residual entropy at 0 K for H2O). It thus appears that it is for these temperature-dependent values of $c_p(T)$ for gases that the agreement between the calorimetric and quantum methods can be obtained, an agreement which is not obtained with "ideal gas" formulations.

I have also plotted on Figure 12 bellow the constant values used in many GCM and NWP models (1004.6, 1004.7, 1005 J/K/kg, depicted by coloured horizontal dashed lines). It appears, considering all these values of $c_p$ constant or dependent on $T$, and in the range of atmospheric temperatures ($200 < T < 320$ K), that the imprecision on $c_p(T)$ is between 1004.5 and 1007.5 J/K/kg. These extreme values have been used earlier in this review to plot several figures, instead of the (old) WMO extreme values 994 and 1011 J/K/kg you used in your study.

Regarding the search for an accurate average value $c_p(T) \approx c_p^0$, it appears that $c_p^0 \approx$ 1005.8 J/K/kg could be more realistic (for $200 < T < 320$ K) than those presently used in GCM and NWP models (1004.6, 1004.7, 1005 J/K/kg).

However, the impact of these new formulations ($c_p(T)$ or $c_p^0$) should be small in our CMGs and NWP models. Moreover, taking into account the dependence of $c_p(T)$ on temperature, not only for dry air but also for water vapor, liquid water and ice, would greatly complicate

the writing of the physical parameterizations of these models, and would greatly increase the cost of these physical parameterizations.

**3 Additional Figures**

Thus Bezold reached the same conclusion as Hann.

In order to facilitate the discussion of this result and the analysis of similar processes, Bezold introduced a number of terms borrowed from physics into the meteorological vocabulary, the most important being "potential temperature."[89] This quantity stood for the heat content of air which in 1888 Helmholtz had defined as the temperature that the parcel of air assumes when being compressed or expanded adiabatically to a standard pressure.[90] It quickly became part of standard meteorological terminology. Using this convenient quantity, Bezold briefly and plainly described what he had expressed graphically:
* * *
[88] Bezold, *Gesammelte Abhandlungen*, p. 124.
[89] Also specific moisture, pseudo-adiabatic process, entropy.
[90] Helmholtz, "Ueber atmosphärische Bewegungen," *Sitzber. Ak. Berlin* (1888), 652–653, translated in Abbe's second collection (1891), 78–111. According to A. Wegener, (*Thermodynamik der Atmosphäre*, Leipzig, Barth, 1911, p. 111) early in 1888 (before publication of Bezold's paper) Köppen had already used the term potential temperature in a talk in Hamburg, entitled "Ueber die Luftmischung und potentielle Temperatur, in Anlehnung an die neueste Abhandlung von Herrn v. Helmholtz."

Figure 1: *Excerpts from Kutzbash (1979) page 143, with the title of the 1888 lecture of Köppen written as:* "Ueber die Luftmischung und potentielle Temperatur", which might be different from the one in your bibliography: *"Über Luftmischung..."*. Moreover, I have not found the paper (or a copy of this lecture) of Köppen, and I do not understand why you cite the URL: `http://snowcrystals.com/`.

[Figure]

[Figure]

Figure 2: *Excerpts from the book "Treatise on Mechanics" by Poisson (1833).* **On the top:** These equations (6) contain the "law of elastic force" and of "the temperature of gases", either compressed or expanded without any changes in their "quantity of heat" *(say adiabatic).* These laws are based on the sole hypothesis that the ratio $\gamma$ of the specific heat *(capacities, say $c_p/c_v$)* does not depend, for a given fluid, on both pressure and temperature *(here "$\theta + 266.67$" corresponds to the absolute temperature defined 15 years after in 1848 by W. Thomson, next Lord Kelvin, now set to: "$T = t + 273.15$ K").* With modern notations, these "adiabatic" laws are: $p' = p\,(\rho'/\rho)^\gamma$ and $T' = T\,(\rho'/\rho)^{\gamma-1}$. **On the bottom:** ... we can consider $\gamma$ as a constant ... Dulong found that, for a perfectly dry air, $\gamma \approx 1.421$ (which is not so different from the modern value 1.40 for the diatomic gases).

[Figure]

Figure 3: *Excerpts from the paper "On the convctive equilibrium of temperature in the atmosphere" by W. Thomson (lecture 1862, published 1865), next Lord Kelvin. This lecture was read a few years before Clausius' article (1865) in which he defined entropy. With modern notations, Eq.(1) writes $(p/p_0)^\kappa = T/\theta$, where $\theta$ is the (absolute) temperature when the parcel of temperature $T$ and pressure $p$ is brought to the surface pressure $p_0$ via an adiabatic transformation. This is the definition of the "potential temperature". Here $\kappa = 1 - 1/k = 1 - 1/\gamma \approx 0.291$, where $\gamma = c_p/c_v \approx 1.41$ is an improved value since Poisson (1833).*

**THE RELATION BETWEEN "POTENTIAL TEMPERA-TURE" AND "ENTROPY."[1]**

**By L. A. Bauer.**

IN 1888 the late Professor von Helmholtz incidentally introduced the term "waermegehalt" in connection with his investigations,[2] "On Atmospheric Motions." According to him the "waermegehalt" or the actual heat contained in a given mass of air is to be measured by the absolute temperature which the mass would assume if it were brought adiabatically to the normal or standard pressure. It remained for the late Professor von Bezold, however, to perceive the full significance of this term and to reveal its important bearing in the discussion of meteorological phenomena.

As the quantity really involved in this new term is not a quantity of heat, von Bezold suggested that the term be replaced by the evidently more appropriate one of "potential temperature."[3] This met with von Helmholtz's approval.

With the aid of this happy idea of "potential temperature" von Bezold was enabled to draw in a simple and beautiful manner a number of important conclusions governing thermodynamic phenomena taking place in the atmosphere. Thus, for example, he found that:

"Strict adiabatic changes of state in the atmosphere leave the potential temperature unchanged, whereas pseudo-adiabatic ones invariably increase the same, the increase being in proportion to the amount of aqueous evaporation."

Von Bezold called attention to the fact that this law bears a strik-

[1] Presented before the Philosophical Society of Washington, March 16, 1907.
[2] Sitzungsberichte Berliner Akademie, 1888, V. XLVI., p. 652, "Ueber atmospherische Bewegungen," see translation in Abbe's Mechanics of the Earth's Atmosphere, Washington, 1891, p. 83. The symbol $\theta$ is used to denote the "Waermegehalt."
[3] Sitzb. Berliner Akad., 1888, V. XLVI., p. 1189, "Zur Thermodynamik der Atmosphaere"; also in von Bezold's "Gesammelte Abhandlungen," Vieweg und Sohn, Braunschweig, 1906, p. 128. A translation will be found in Abbe's Mechanics, etc., p. 243.

ing resemblance to the well-known theorem of Clausius, now commonly known as the second law of thermodynamics, viz.: "that the entropy strives towards a maximum;" but, he says, "it is not identical with it."

The purpose of this paper is to examine into the precise relationship between the two functions "potential temperature" and "entropy" and to see whether any use could be made advantageously of the former in the treatment of certain thermodynamic problems as well as to ascertain wherein the potential temperature law fails to give full expression of the second law of thermodynamics. To my knowledge no application has as yet been made of the new term in treatises on thermodynamics. The substance of this paper was communicated to the American Association for the Advancement of Science at the Springfield meeting in 1895, but publication pending opportunity for further elaboration was deferred.

*The "potential temperature" of a body is defined as the absolute temperature assumed when the body is brought adiabatically to standard pressure.*

Defining the thermodynamic state per *unit of mass* of a body by the three variables, $T$, the absolute temperature, $v$, the volume per unit of mass, $p$, the pressure supposed uniform, the following characteristic equation subsists between them: $T = f(v, p)$.

If the body be brought now adiabatically to standard pressure $p_0$, then the temperature assumed at the end of the process is the so-called *potential temperature* as above defined and is designated by the symbol $\theta$. Hence,

$$\theta = f(v, p_0). \tag{1}$$

For a perfect gas, since $kT = pv$, $k$ being a constant for any particular gas,

$$\theta = \frac{p_0}{k} \cdot v = k_0 \cdot v, \tag{2}$$

*or the potential temperature for any particular gas is directly proportional to the volume* and, hence, as von Bezold showed, the potential temperature readily admits of a graphical representation on the usual $pv$ diagram, being simply proportional to the $v$ abscissæ of points of intersection of the line of standard pressure, $p = p_0$, with the adiabats.

Figure 4: *The first two pages of Bauer (lecture 1907; printed 1910). The origin of the name "potential temperature" is clearly credited to von Bezold with von Helmholtz's approval (and this is confirmed by the reading of von Bezold's papers of 1888), but without mention to the lecture of Köppen in this paper, nor in those of von Helmoltz and von Bezold.*

Figure 5: *The page 180 of Bauer (lecture 1907; printed 1910), where the link between (dry-air) entropy and potential temperature "$s = c_p \ln(\theta) + const$" appear for the first time in meteorological science.*

Figure 6: *Excerpts from the paper Valis (2009) pages 14 and 21. It is clearly explained that $R_d \approx 287$ J/K/kg and $c_p = (7/2)\,R_d$, leading to $c_p \approx 1004.5$ J/K/kg, and not $c_p \approx 1000$ J/K/kg as suggested in your Fig.3 and Table 1. This value $c_p \approx 1000$ J/K/kg is here only a rough indication, a very simple "order of magnitude".*

[Figure]

Figure 7: *The vertical profiles of $\theta_{1004.5}$ and $\theta_{1007.5}$, and the difference of them (in red), plotted for the same US Standard Atmosphere as used in your paper, and compared to the same difference $\Delta\theta_{c_p} = \theta_{994} - \theta_{1011}$ (in black) as in your paper.*

[Figure]

Figure 8: **Top:** *The copy of your Fig.3;* **Bottom:** *a modified version of it, where the "questionable data" are removed. The idealized curves of Lemmon (2000) and Dixon (2007) are kept here, although they should also be deleted and replaced by the real cases depicted in the next figure (IAPWS, TEOS10, Lemmon 2000 Table A2, Jacobsen et al. 1997, ...), as explained in the text.*

 **LEMMON *ET AL.***

TABLE A2. Thermodynamic properties of air

| Temperature (K) | Density (mol/dm³) | Internal energy (J/mol) | Enthalpy (J/mol) | Entropy J/(mol-K) | $c_v$ J/(mol-K) | $c_p$ J/(mol-K) | Speed of sound (m/s) |
|---|---|---|---|---|---|---|---|
| | | | 0.101 325 MPa isobar | | | | |
| 59.77 | 33.069 | −4713.1 | −4710.0 | 70.905 | 34.01 | 55.05 | 1030.6 |
| 60 | 33.036 | −4700.3 | −4697.2 | 71.119 | 33.96 | 55.05 | 1028.8 |
| 62 | 32.750 | −4590.2 | −4587.1 | 72.924 | 33.52 | 55.05 | 1012.6 |
| 64 | 32.462 | −4480.1 | −4477.0 | 74.672 | 33.09 | 55.07 | 996.3 |
| 66 | 32.171 | −4369.9 | −4366.8 | 76.367 | 32.69 | 55.11 | 979.6 |
| 68 | 31.878 | −4259.7 | −4256.5 | 78.013 | 32.30 | 55.17 | 962.7 |
| 70 | 31.581 | −4149.3 | −4146.1 | 79.614 | 31.92 | 55.25 | 945.5 |
| 72 | 31.281 | −4038.7 | −4035.5 | 81.172 | 31.56 | 55.37 | 928.1 |
| 74 | 30.978 | −3927.9 | −3924.6 | 82.691 | 31.22 | 55.51 | 910.4 |
| 76 | 30.670 | −3816.7 | −3813.4 | 84.173 | 30.89 | 55.68 | 892.3 |
| 78 | 30.358 | −3705.2 | −3701.9 | 85.622 | 30.57 | 55.88 | 874.0 |
| 78.90 | 30.215 | −3654.7 | −3651.4 | 86.266 | 30.43 | 55.99 | 865.6 |
| 81.72 | 0.155 27 | 1628.3 | 2280.9 | 160.41 | 21.73 | 31.56 | 177.2 |
| 82 | 0.154 67 | 1634.6 | 2289.8 | 160.52 | 21.71 | 31.52 | 177.5 |
| 84 | 0.150 53 | 1679.4 | 2352.5 | 161.28 | 21.58 | 31.26 | 180.0 |
| 86 | 0.146 63 | 1723.8 | 2414.8 | 162.01 | 21.48 | 31.04 | 182.4 |
| 88 | 0.142 95 | 1767.9 | 2476.7 | 162.72 | 21.40 | 30.85 | 184.8 |
| 90 | 0.139 47 | 1811.7 | 2538.2 | 163.41 | 21.33 | 30.69 | 187.1 |
| 92 | 0.136 17 | 1855.4 | 2599.5 | 164.09 | 21.27 | 30.55 | 189.4 |
| 94 | 0.133 03 | 1898.8 | 2660.4 | 164.74 | 21.22 | 30.42 | 191.7 |
| 96 | 0.130 05 | 1942.1 | 2721.2 | 165.38 | 21.17 | 30.32 | 193.9 |
| 98 | 0.127 21 | 1985.2 | 2781.7 | 166.00 | 21.13 | 30.22 | 196.1 |
| 100 | 0.124 49 | 2028.2 | 2842.1 | 166.61 | 21.09 | 30.13 | 198.2 |
| 102 | 0.121 90 | 2071.0 | 2902.2 | 167.21 | 21.06 | 30.05 | 200.3 |
| 104 | 0.119 42 | 2113.8 | 2962.3 | 167.79 | 21.03 | 29.98 | 202.4 |
| 106 | 0.117 04 | 2156.4 | 3022.2 | 168.36 | 21.01 | 29.92 | 204.5 |
| 108 | 0.114 76 | 2199.0 | 3082.0 | 168.92 | 20.98 | 29.86 | 206.5 |
| 110 | 0.112 57 | 2241.5 | 3141.6 | 169.47 | 20.96 | 29.81 | 208.6 |
| 112 | 0.110 47 | 2284.0 | 3201.2 | 170.01 | 20.94 | 29.76 | 210.5 |
| 114 | 0.108 44 | 2326.3 | 3260.7 | 170.53 | 20.93 | 29.72 | 212.5 |
| 116 | 0.106 49 | 2368.6 | 3320.1 | 171.05 | 20.91 | 29.68 | 214.5 |
| 118 | 0.104 62 | 2410.9 | 3379.4 | 171.56 | 20.90 | 29.64 | 216.4 |
| 120 | 0.102 81 | 2453.1 | 3438.7 | 172.05 | 20.89 | 29.61 | 218.3 |
| 122 | 0.101 06 | 2495.3 | 3497.8 | 172.54 | 20.87 | 29.58 | 220.2 |
| 124 | 0.099 377 | 2537.4 | 3557.0 | 173.02 | 20.86 | 29.55 | 222.1 |
| 126 | 0.097 748 | 2579.5 | 3616.0 | 173.50 | 20.85 | 29.52 | 223.9 |
| 128 | 0.096 174 | 2621.5 | 3675.1 | 173.96 | 20.84 | 29.50 | 225.8 |
| 130 | 0.094 650 | 2663.5 | 3734.0 | 174.42 | 20.84 | 29.48 | 227.6 |
| 132 | 0.093 175 | 2705.5 | 3793.0 | 174.87 | 20.83 | 29.45 | 229.4 |
| 134 | 0.091 747 | 2747.5 | 3851.9 | 175.31 | 20.82 | 29.43 | 231.2 |
| 136 | 0.090 363 | 2789.4 | 3910.7 | 175.75 | 20.82 | 29.42 | 232.9 |
| 138 | 0.089 021 | 2831.3 | 3969.5 | 176.18 | 20.81 | 29.40 | 234.7 |
| 140 | 0.087 718 | 2873.2 | 4028.3 | 176.60 | 20.81 | 29.38 | 236.4 |
| 142 | 0.086 455 | 2915.1 | 4087.1 | 177.02 | 20.80 | 29.37 | 238.2 |
| 144 | 0.085 227 | 2956.9 | 4145.8 | 177.43 | 20.80 | 29.35 | 239.9 |
| 146 | 0.084 035 | 2998.7 | 4204.5 | 177.83 | 20.79 | 29.34 | 241.6 |
| 148 | 0.082 877 | 3040.5 | 4263.1 | 178.23 | 20.79 | 29.33 | 243.3 |
| 150 | 0.081 750 | 3082.3 | 4321.8 | 178.62 | 20.78 | 29.32 | 244.9 |
| 155 | 0.079 065 | 3186.8 | 4468.3 | 179.59 | 20.78 | 29.29 | 249.1 |
| 160 | 0.076 553 | 3291.1 | 4614.7 | 180.51 | 20.77 | 29.27 | 253.1 |
| 165 | 0.074 198 | 3395.4 | 4761.0 | 181.41 | 20.76 | 29.25 | 257.1 |
| 170 | 0.071 985 | 3499.6 | 4907.2 | 182.29 | 20.76 | 29.23 | 261.1 |
| 175 | 0.069 902 | 3603.8 | 5053.3 | 183.13 | 20.75 | 29.22 | 264.9 |
| 180 | 0.067 937 | 3707.9 | 5199.3 | 183.96 | 20.75 | 29.20 | 268.7 |
| 185 | 0.066 081 | 3812.0 | 5345.3 | 184.76 | 20.75 | 29.19 | 272.5 |
| 190 | 0.064 324 | 3916.0 | 5491.2 | 185.54 | 20.75 | 29.18 | 276.2 |
| 195 | 0.062 659 | 4020.0 | 5637.1 | 186.29 | 20.75 | 29.17 | 279.8 |
| 200 | 0.061 079 | 4124.0 | 5782.9 | 187.03 | 20.74 | 29.16 | 283.4 |
| 210 | 0.058 147 | 4331.9 | 6074.5 | 188.45 | 20.74 | 29.15 | 290.5 |
| 220 | 0.055 486 | 4539.8 | 6365.9 | 189.81 | 20.74 | 29.14 | 297.4 |
| 230 | 0.053 059 | 4747.6 | 6657.3 | 191.11 | 20.74 | 29.13 | 304.1 |
| 240 | 0.050 836 | 4955.4 | 6948.6 | 192.35 | 20.75 | 29.13 | 310.7 |

Figure 9: *Values of $c_p(T)$ at 1013.25 hPa for dry air (above 81.72 K) from Lemmon et al. (2000, Table A2).*

Table A2. Thermodynamic properties of air—Continued

| Temperature (K) | Density (mol/dm³) | Internal energy (J/mol) | Enthalpy (J/mol) | Entropy J/(mol-K) | $c_v$ J/(mol-K) | $c_p$ J/(mol-K) | Speed of sound (m/s) |
|---|---|---|---|---|---|---|---|
| 250 | 0.048 793 | 5163.2 | 7239.9 | 193.53 | 20.75 | 29.13 | 317.1 |
| 260 | 0.046 908 | 5371.1 | 7531.1 | 194.68 | 20.76 | 29.13 | 323.4 |
| 270 | 0.045 164 | 5578.9 | 7822.4 | 195.78 | 20.76 | 29.13 | 329.6 |
| 280 | 0.043 546 | 5786.8 | 8113.7 | 196.84 | 20.77 | 29.13 | 335.6 |
| 290 | 0.042 040 | 5994.8 | 8405.1 | 197.86 | 20.78 | 29.14 | 341.5 |
| 300 | 0.040 634 | 6203.0 | 8696.5 | 198.85 | 20.80 | 29.15 | 347.4 |
| 310 | 0.039 320 | 6411.2 | 8988.1 | 199.80 | 20.81 | 29.16 | 353.1 |
| 320 | 0.038 089 | 6619.5 | 9279.8 | 200.73 | 20.83 | 29.18 | 358.7 |
| 330 | 0.036 932 | 6828.1 | 9571.6 | 201.63 | 20.85 | 29.19 | 364.2 |
| 340 | 0.035 844 | 7036.8 | 9863.6 | 202.50 | 20.87 | 29.21 | 369.7 |
| 350 | 0.034 818 | 7245.7 | 10 156.0 | 203.34 | 20.89 | 29.23 | 375.0 |

Figure 10:  *Values of $c_p(T)$ at 1013.25 hPa for dry air (above 81.72 K) from Lemmon et al. (2000, Table A2) (continued).*

The specific thermal capacity at constant pressure $c_P$ is given by the empirical expression

$$c_P = 1002.5 + 275 \times 10^{-6}(T_K - 200)^2 \text{ J/kg K}$$

Figure 11:  *The equation of Dixon (2007, p.376) used to compute the dry-air value $c_p(T)$ plotted in your Fig.3: $c_p(T) = 1002.5 + 275. \, 10^{-6} \, (T - 200)^2$ J/K/kg. You may help the reader by including this formula? Or helping the reader by giving the page 376? You may help the reader by indicating that the accuracy would be of 0.1 % from 200 K to 450 K?*

[Figure]

Figure 12: *The same as your Fig.3 and with your NIST-REFPROP (1013.25 hPa) datasets, but with other "real gases" formulations of $c_p(T)$ depending on $T$: IAPWS + TEOS10, Lemmon 2000 Table A2, Jacobsen et al. 1997 / Marquet 2015. Constant values used in many GCM and NWP models (1004.6, 1004.7, 1005) are depicted by coloured horizontal dashed lines, with the value 1005.8 a possible new "mean value"?*

**Table 5.73. Thermodynamic Properties of Nitrogen**

| Temperature (K) | Density (kg/m³) | Internal energy (kJ/kg) | Enthalpy (kJ/kg) | Entropy (kJ/kg-K) | $C_v$ (kJ/kg-K) | $C_p$ (kJ/kg-K) | Velocity of sound (m/s) |
|---|---|---|---|---|---|---|---|
| | | | 0.1 MPa isobar | | | | |
| 63.159 | 869.82 | −150.90 | −150.78 | 2.4232 | 1.117 | 2.019 | 1022. |
| 70.000 | 839.96 | −137.13 | −137.01 | 2.6303 | 1.094 | 2.015 | 934. |
| 77.237 | 807.14 | −122.47 | −122.34 | 2.8296 | 1.068 | 2.042 | 851. |
| 77.237 | 4.5655 | 54.738 | 76.642 | 5.4059 | 0.9373 | 1.340 | 172. |
| 80.000 | 4.3796 | 57.275 | 80.108 | 5.4500 | 0.8294 | 1.191 | 177. |
| 90.000 | 3.8449 | 65.241 | 91.250 | 5.5813 | 0.7536 | 1.081 | 191. |
| 100.00 | 3.4361 | 72.871 | 101.97 | 5.6943 | 0.7480 | 1.067 | 202. |
| 110.00 | 3.1089 | 80.440 | 112.61 | 5.7957 | 0.7465 | 1.060 | 212. |
| 120.00 | 2.8402 | 87.976 | 123.18 | 5.8877 | 0.7456 | 1.056 | 222. |
| 130.00 | 2.6153 | 95.490 | 133.73 | 5.9721 | 0.7450 | 1.053 | 231. |
| 140.00 | 2.4241 | 102.99 | 144.24 | 6.0500 | 0.7444 | 1.050 | 240. |
| 150.00 | 2.2593 | 110.47 | 154.73 | 6.1224 | 0.7440 | 1.048 | 249. |
| 160.00 | 2.1158 | 117.94 | 165.21 | 6.1900 | 0.7437 | 1.047 | 257. |
| 170.00 | 1.9896 | 125.41 | 175.67 | 6.2534 | 0.7435 | 1.046 | 265. |
| 180.00 | 1.8778 | 132.87 | 186.12 | 6.3132 | 0.7433 | 1.045 | 273. |
| 190.00 | 1.7780 | 140.32 | 196.57 | 6.3696 | 0.7431 | 1.044 | 281. |
| 200.00 | 1.6883 | 147.77 | 207.00 | 6.4232 | 0.7430 | 1.043 | 288. |
| 210.00 | 1.6073 | 155.22 | 217.44 | 6.4741 | 0.7429 | 1.043 | 295. |
| 220.00 | 1.5338 | 162.66 | 227.86 | 6.5226 | 0.7429 | 1.043 | 302. |
| 230.00 | 1.4667 | 170.11 | 238.29 | 6.5689 | 0.7428 | 1.042 | 309. |
| 240.00 | 1.4053 | 177.55 | 248.71 | 6.6133 | 0.7428 | 1.042 | 316. |
| 250.00 | 1.3488 | 184.99 | 259.13 | 6.6558 | 0.7428 | 1.042 | 322. |
| 260.00 | 1.2967 | 192.42 | 269.54 | 6.6966 | 0.7428 | 1.042 | 329. |
| 270.00 | 1.2485 | 199.86 | 279.96 | 6.7360 | 0.7428 | 1.041 | 335. |
| 280.00 | 1.2038 | 207.30 | 290.37 | 6.7738 | 0.7429 | 1.041 | 341. |
| 290.00 | 1.1621 | 214.74 | 300.78 | 6.8104 | 0.7430 | 1.041 | 347. |
| 300.00 | 1.1233 | 222.17 | 311.20 | 6.8457 | 0.7431 | 1.041 | 353. |
| 310.00 | 1.0870 | 229.61 | 321.61 | 6.8798 | 0.7433 | 1.041 | 359. |
| 320.00 | 1.0529 | 237.05 | 332.02 | 6.9129 | 0.7435 | 1.041 | 365. |
| 330.00 | 1.0210 | 244.49 | 342.44 | 6.9449 | 0.7438 | 1.042 | 370. |
| 340.00 | 0.99090 | 251.94 | 352.86 | 6.9760 | 0.7441 | 1.042 | 376. |
| 350.00 | 0.96254 | 259.39 | 363.28 | 7.0062 | 0.7445 | 1.042 | 381. |
| 360.00 | 0.93577 | 266.84 | 373.70 | 7.0356 | 0.7449 | 1.043 | 387. |
| 370.00 | 0.91045 | 274.29 | 384.13 | 7.0642 | 0.7454 | 1.043 | 392. |
| 380.00 | 0.88646 | 281.76 | 394.56 | 7.0920 | 0.7460 | 1.044 | 397. |
| 390.00 | 0.86371 | 289.22 | 405.00 | 7.1191 | 0.7467 | 1.044 | 402. |
| 400.00 | 0.84209 | 296.70 | 415.45 | 7.1456 | 0.7474 | 1.045 | 408. |
| 450.00 | 0.74847 | 334.20 | 467.80 | 7.2689 | 0.7523 | 1.050 | 432. |
| 500.00 | 0.67359 | 371.99 | 520.45 | 7.3798 | 0.7592 | 1.056 | 455. |
| 600.00 | 0.56131 | 448.82 | 626.98 | 7.5740 | 0.7781 | 1.075 | 496. |
| 700.00 | 0.48113 | 527.77 | 735.62 | 7.7414 | 0.8011 | 1.098 | 534. |

Figure 13:    *Values of $c_p(T)$ at 1000 hPa for Nitrogen. Dataset for vapour (N2, above 77.237 K) is from Jacobsen et al. (1997, Table 5.73).*

Thermodynamic properties of nitrogen

| T K | ρ mol/dm³ | u J/mol | h J/mol | s J/(mol K) | $c_v$ J/(mol K) | $c_p$ J/(mol K) | w m/s |
|---|---|---|---|---|---|---|---|
| | | | | 0.1 MPa | | | |
| 63.170 | 30.960 | −4222.8 | −4219.6 | 67.955 | 32.95 | 56.02 | 995.6 |
| 65 | 30.690 | −4120.2 | −4117.0 | 69.556 | 32.60 | 56.11 | 976.9 |
| 70 | 29.937 | −3839.0 | −3835.7 | 73.724 | 31.65 | 56.42 | 926.2 |
| 75 | 29.155 | −3555.9 | −3552.5 | 77.632 | 30.76 | 56.89 | 875.5 |
| 77.244 | 28.793 | −3428.0 | −3424.6 | 79.313 | 30.39 | 57.17 | 852.5 |
| 77.244 | 0.162 65 | 1544.3 | 2159.1 | 151.60 | 21.60 | 31.46 | 174.7 |
| 80 | 0.156 33 | 1605.7 | 2245.4 | 152.70 | 21.48 | 31.15 | 178.3 |
| 85 | 0.146 15 | 1715.8 | 2400.0 | 154.57 | 21.31 | 30.72 | 184.5 |
| 90 | 0.137 32 | 1824.5 | 2552.8 | 156.32 | 21.20 | 30.41 | 190.5 |
| 95 | 0.129 56 | 1932.4 | 2704.3 | 157.96 | 21.11 | 30.19 | 196.2 |
| 100 | 0.122 68 | 2039.6 | 2854.7 | 159.50 | 21.05 | 30.01 | 201.6 |
| 105 | 0.116 52 | 2146.2 | 3004.4 | 160.96 | 21.00 | 29.88 | 206.9 |
| 110 | 0.110 98 | 2252.5 | 3153.5 | 162.35 | 20.97 | 29.77 | 212.1 |
| 115 | 0.105 96 | 2358.4 | 3302.2 | 163.67 | 20.94 | 29.68 | 217.1 |
| 120 | 0.101 38 | 2464.0 | 3450.4 | 164.93 | 20.92 | 29.61 | 222.0 |
| 125 | 0.097200 | 2569.5 | 3598.3 | 166.14 | 20.90 | 29.55 | 226.7 |
| 130 | 0.093355 | 2674.8 | 3745.9 | 167.30 | 20.88 | 29.50 | 231.4 |
| 135 | 0.089809 | 2779.9 | 3893.3 | 168.41 | 20.87 | 29.46 | 235.9 |
| 140 | 0.086528 | 2884.9 | 4040.6 | 169.48 | 20.86 | 29.43 | 240.4 |
| 145 | 0.083482 | 2989.8 | 4187.6 | 170.51 | 20.85 | 29.40 | 244.7 |
| 150 | 0.080647 | 3094.6 | 4334.5 | 171.51 | 20.85 | 29.37 | 249.0 |
| 160 | 0.075524 | 3303.9 | 4628.0 | 173.40 | 20.84 | 29.33 | 257.3 |
| 170 | 0.071021 | 3513.1 | 4921.1 | 175.18 | 20.83 | 29.29 | 265.4 |
| 180 | 0.067029 | 3722.0 | 5213.9 | 176.85 | 20.82 | 29.27 | 273.2 |
| 190 | 0.063466 | 3930.8 | 5506.5 | 178.44 | 20.82 | 29.25 | 280.7 |
| 200 | 0.060265 | 4139.5 | 5798.9 | 179.93 | 20.81 | 29.23 | 288.1 |
| 210 | 0.057374 | 4348.2 | 6091.1 | 181.36 | 20.81 | 29.22 | 295.3 |
| 220 | 0.054749 | 4556.7 | 6383.2 | 182.72 | 20.81 | 29.21 | 302.3 |
| 230 | 0.052355 | 4765.2 | 6675.2 | 184.02 | 20.81 | 29.20 | 309.1 |
| 240 | 0.050162 | 4973.6 | 6967.1 | 185.26 | 20.81 | 29.19 | 315.8 |
| 250 | 0.048147 | 5182.0 | 7259.0 | 186.45 | 20.81 | 29.18 | 322.3 |
| 260 | 0.046287 | 5390.4 | 7550.8 | 187.60 | 20.81 | 29.18 | 328.7 |
| 270 | 0.044567 | 5598.7 | 7842.5 | 188.70 | 20.81 | 29.17 | 335.0 |
| 280 | 0.042970 | 5807.1 | 8134.3 | 189.76 | 20.81 | 29.17 | 341.2 |
| 290 | 0.041484 | 6015.4 | 8426.0 | 190.78 | 20.81 | 29.17 | 347.2 |
| 300 | 0.040098 | 6223.8 | 8717.7 | 191.77 | 20.82 | 29.17 | 353.2 |
| 310 | 0.038801 | 6432.2 | 9009.4 | 192.73 | 20.82 | 29.17 | 359.0 |
| 320 | 0.037586 | 6640.6 | 9301.2 | 193.65 | 20.83 | 29.18 | 364.7 |
| 330 | 0.036445 | 6849.1 | 9593.0 | 194.55 | 20.84 | 29.18 | 370.4 |
| 340 | 0.035371 | 7057.7 | 9884.8 | 195.42 | 20.85 | 29.19 | 375.9 |
| 350 | 0.034359 | 7266.3 | 10 177 | 196.27 | 20.86 | 29.20 | 381.4 |
| 400 | 0.030060 | 8311.6 | 11 638 | 200.17 | 20.94 | 29.27 | 407.5 |
| 450 | 0.026718 | 9362.1 | 13 105 | 203.63 | 21.08 | 29.40 | 431.8 |
| 500 | 0.024045 | 10 421 | 14 580 | 206.73 | 21.27 | 29.59 | 454.6 |
| 550 | 0.021858 | 11 490 | 16 065 | 209.57 | 21.51 | 29.84 | 476.0 |
| 600 | 0.020037 | 12 573 | 17 564 | 212.17 | 21.80 | 30.12 | 496.3 |
| 700 | 0.017175 | 14 785 | 20 607 | 216.86 | 22.44 | 30.76 | 533.9 |
| 800 | 0.015028 | 17 063 | 23 717 | 221.02 | 23.12 | 31.44 | 568.4 |
| 900 | 0.013359 | 19 409 | 26 894 | 224.76 | 23.78 | 32.10 | 600.7 |
| 1000 | 0.012023 | 21 817 | 30 135 | 228.17 | 24.39 | 32.70 | 631.1 |

J. Phys. Chem. Ref. Data, Vol. 29, No. 6, 2000

Figure 14:  *Values of $c_p(T)$ at 1000 hPa for Nitrogen. Dataset for vapour (N2, above 77.244 K) is from Span et al. (2000, Table p.1410).*

[Figure]

**Figure 5.** Specific heat capacity at constant pressure for $N_2$ corresponding to Table BII. Units of $c_p$ are $J\,K^{-1}kg^{-1}$. The latent heats are in units of $kJ\,kg^{-1}$.

| $N_2$ (vapour). Unit $J\,K^{-1}\,kg^{-1}$ | | | | | |
|---|---|---|---|---|---|
| 77.4 | 1340 | 120 | 1056 | 200 | 1043 |
| 80 | 1191 | 140 | 1050 | 250 | 1042 |
| 90 | 1081 | 160 | 1047 | 300 | 1041 |
| 100 | 1067 | 180 | 1045 | | |

Figure 15: *Values of $c_p(T)$ at 1000 hPa for Nitrogen (solids $\alpha$ and $\beta$, liquid, gas) from Marquet (2015).*

**Table 5.79. Thermodynamic Properties of Oxygen**

| Temperature (K) | Density (kg/m³) | Internal energy (kJ/kg) | Enthalpy (kJ/kg) | Entropy (kJ/kg-K) | $C_v$ (kJ/kg-K) | $C_p$ (kJ/kg-K) | Velocity of sound (m/s) |
|---|---|---|---|---|---|---|---|
| | | | 0.1 MPa isobar | | | | |
| 54.371 | 1306.1 | −193.61 | −193.53 | 2.0922 | 1.195 | 1.673 | 1124. |
| 55.000 | 1303.6 | −192.56 | −192.48 | 2.1114 | 1.176 | 1.671 | 1127. |
| 60.000 | 1282.1 | −184.21 | −184.13 | 2.2569 | 1.089 | 1.673 | 1128. |
| 65.000 | 1259.8 | −175.83 | −175.75 | 2.3910 | 1.046 | 1.677 | 1102. |
| 70.000 | 1237.1 | −167.44 | −167.36 | 2.5153 | 1.017 | 1.678 | 1067. |
| 75.000 | 1214.0 | −159.05 | −158.97 | 2.6311 | 0.9922 | 1.679 | 1028. |
| 80.000 | 1190.6 | −150.66 | −150.57 | 2.7395 | 0.9699 | 1.681 | 988. |
| 85.000 | 1166.7 | −142.24 | −142.15 | 2.8416 | 0.9490 | 1.688 | 947. |
| 90.000 | 1142.1 | −133.78 | −133.69 | 2.9383 | 0.9296 | 1.699 | 906. |
| 90.062 | 1141.8 | −133.67 | −133.58 | 2.9395 | 0.9293 | 1.699 | 905. |
| 90.062 | 4.4135 | 56.939 | 79.597 | 5.3065 | 0.6757 | 0.9705 | 177. |
| 95.000 | 4.1642 | 60.284 | 84.298 | 5.3573 | 0.6549 | 0.9413 | 183. |
| 100.00 | 3.9411 | 63.612 | 88.986 | 5.4054 | 0.6527 | 0.9352 | 188. |
| 105.00 | 3.7418 | 66.931 | 93.656 | 5.4510 | 0.6534 | 0.9332 | 193. |
| 110.00 | 3.5625 | 70.247 | 98.318 | 5.4944 | 0.6541 | 0.9316 | 198. |
| 115.00 | 3.4001 | 73.561 | 102.97 | 5.5357 | 0.6544 | 0.9298 | 203. |
| 120.00 | 3.2524 | 76.870 | 107.62 | 5.5753 | 0.6543 | 0.9280 | 207. |
| 125.00 | 3.1174 | 80.174 | 112.25 | 5.6131 | 0.6540 | 0.9262 | 212. |
| 130.00 | 2.9935 | 83.473 | 116.88 | 5.6494 | 0.6537 | 0.9246 | 216. |
| 135.00 | 2.8793 | 86.767 | 121.50 | 5.6843 | 0.6533 | 0.9231 | 220. |
| 140.00 | 2.7736 | 90.056 | 126.11 | 5.7178 | 0.6530 | 0.9218 | 225. |
| 145.00 | 2.6756 | 93.342 | 130.72 | 5.7502 | 0.6527 | 0.9207 | 229. |
| 150.00 | 2.5845 | 96.624 | 135.32 | 5.7814 | 0.6524 | 0.9196 | 233. |
| 155.00 | 2.4994 | 99.903 | 139.91 | 5.8115 | 0.6521 | 0.9188 | 237. |
| 160.00 | 2.4199 | 103.18 | 144.50 | 5.8407 | 0.6519 | 0.9180 | 241. |
| 165.00 | 2.3453 | 106.45 | 149.09 | 5.8689 | 0.6517 | 0.9173 | 244. |
| 170.00 | 2.2752 | 109.73 | 153.68 | 5.8963 | 0.6516 | 0.9167 | 248. |
| 175.00 | 2.2093 | 113.00 | 158.26 | 5.9228 | 0.6514 | 0.9162 | 252. |
| 180.00 | 2.1471 | 116.26 | 162.84 | 5.9486 | 0.6513 | 0.9158 | 255. |
| 185.00 | 2.0883 | 119.53 | 167.42 | 5.9737 | 0.6513 | 0.9154 | 259. |
| 190.00 | 2.0328 | 122.80 | 171.99 | 5.9981 | 0.6512 | 0.9151 | 262. |
| 195.00 | 1.9801 | 126.07 | 176.57 | 6.0219 | 0.6512 | 0.9148 | 266. |
| 200.00 | 1.9301 | 129.33 | 181.14 | 6.0451 | 0.6513 | 0.9146 | 269. |
| 205.00 | 1.8826 | 132.60 | 185.71 | 6.0676 | 0.6513 | 0.9144 | 273. |
| 210.00 | 1.8374 | 135.86 | 190.29 | 6.0897 | 0.6514 | 0.9143 | 276. |
| 215.00 | 1.7943 | 139.12 | 194.86 | 6.1112 | 0.6515 | 0.9142 | 279. |
| 220.00 | 1.7532 | 142.39 | 199.43 | 6.1322 | 0.6516 | 0.9142 | 283. |
| 225.00 | 1.7139 | 145.65 | 204.00 | 6.1527 | 0.6518 | 0.9142 | 286. |
| 230.00 | 1.6764 | 148.92 | 208.57 | 6.1728 | 0.6520 | 0.9143 | 289. |
| 235.00 | 1.6405 | 152.19 | 213.14 | 6.1925 | 0.6522 | 0.9144 | 292. |
| 240.00 | 1.6062 | 155.45 | 217.71 | 6.2118 | 0.6525 | 0.9145 | 295. |
| 250.00 | 1.5416 | 161.99 | 226.86 | 6.2491 | 0.6531 | 0.9150 | 301. |

Figure 16: *Values of $c_p(T)$ at 1000 hPa for Oxygen. Dataset for vapour (O2, above 90.062 K) is from Jacobsen et al. (1997, Table 5.79).*

**Table 5.79 (continued)**

| Temperature (K) | Density (kg/m³) | Internal energy (kJ/kg) | Enthalpy (kJ/kg) | Entropy (kJ/kg-K) | $C_v$ (kJ/kg-K) | $C_p$ (kJ/kg-K) | Velocity of sound (m/s) |
|---|---|---|---|---|---|---|---|
| 260.00 | 1.4820 | 168.54 | 236.02 | 6.2850 | 0.6539 | 0.9156 | 307. |
| 270.00 | 1.4269 | 175.09 | 245.17 | 6.3196 | 0.6549 | 0.9164 | 313. |
| 280.00 | 1.3757 | 181.65 | 254.34 | 6.3529 | 0.6560 | 0.9174 | 319. |
| 290.00 | 1.3281 | 188.23 | 263.52 | 6.3851 | 0.6573 | 0.9185 | 324. |
| 300.00 | 1.2837 | 194.81 | 272.71 | 6.4163 | 0.6587 | 0.9199 | 330. |

Figure 17: *Values of $c_p(T)$ at 1000 hPa for Oxygen. Dataset for vapour (O2, above 90.062 K) is from Jacobsen et al. (1997, Table 5.79, continued).*

[Figure]

**Figure 4.** Specific heat capacity at constant pressure for $O_2$ corresponding to Table BI. Units of $c_p$ are $J\,K^{-1}\,kg^{-1}$. The latent heats are in units of $kJ\,kg^{-1}$.

| $O_2$ (vapour). Unit $J\,K^{-1}kg^{-1}$ | | | | | |
|---|---|---|---|---|---|
| 90 | 970.5 | 135 | 923.1 | 240 | 914.5 |
| 95 | 941.3 | 140 | 921.8 | 250 | 915.0 |
| 100 | 935.2 | 145 | 920.7 | 260 | 915.6 |
| 105 | 933.2 | 150 | 919.6 | 270 | 916.4 |
| 110 | 931.6 | 170 | 916.7 | 280 | 917.4 |
| 115 | 929.8 | 190 | 915.1 | 290 | 918.5 |
| 120 | 928.0 | 210 | 914.3 | 300 | 919.9 |
| 125 | 926.2 | 230 | 914.3 | | |
| 130 | 924.6 | 235 | 914.4 | | |

Figure 18:   *Values of $c_p(T)$ at 1000 hPa for Oxygen (solids $\alpha$, $\beta$ and $\gamma$, liquid, gas) from Marquet (2015).*

TABLE 34. Thermodynamic properties of argon

| Temperature (K) | Density (kg m$^{-3}$) | Internal energy (kJ kg$^{-1}$) | Enthalpy (kJ kg$^{-1}$) | Entropy (kJ kg$^{-1}$ K$^{-1}$) | $c_v$ (kJ kg$^{-1}$ K$^{-1}$) | $c_p$ (kJ kg$^{-1}$ K$^{-1}$) | $w$ (m s$^{-1}$) |
|---|---|---|---|---|---|---|---|
| | | | 0.1 MPa isobar | | | | |
| 83.814[a] | 1416.80 | −276.61 | −276.54 | −2.5440 | 0.549 61 | 1.1156 | 862.52 |
| 85 | 1409.57 | −275.29 | −275.22 | −2.5283 | 0.544 83 | 1.1157 | 854.35 |
| 87.178[b] | 1396.16 | −272.86 | −272.79 | −2.5000 | 0.536 57 | 1.1171 | 839.20 |
| 87.178[b] | 5.7043 | −129.08 | −111.55 | −0.650 58 | 0.327 91 | 0.565 41 | 170.77 |
| 90 | 5.5077 | −128.12 | −109.97 | −0.632 65 | 0.325 70 | 0.560 00 | 173.85 |
| 95 | 5.1933 | −126.44 | −107.19 | −0.602 59 | 0.322 73 | 0.552 56 | 179.11 |
| 100 | 4.9152 | −124.78 | −104.44 | −0.574 39 | 0.320 58 | 0.547 02 | 184.16 |
| 105 | 4.6669 | −123.14 | −101.71 | −0.547 81 | 0.318 99 | 0.542 79 | 189.04 |
| 110 | 4.4436 | −121.51 | −99.008 | −0.522 64 | 0.317 78 | 0.539 48 | 193.76 |
| 115 | 4.2416 | −119.89 | −96.317 | −0.498 72 | 0.316 85 | 0.536 85 | 198.34 |
| 120 | 4.0577 | −118.28 | −93.639 | −0.475 92 | 0.316 12 | 0.534 72 | 202.80 |
| 125 | 3.8896 | −116.68 | −90.969 | −0.454 12 | 0.315 53 | 0.532 98 | 207.15 |
| 130 | 3.7352 | −115.08 | −88.308 | −0.433 25 | 0.315 06 | 0.531 53 | 211.39 |
| 135 | 3.5929 | −113.49 | −85.654 | −0.413 21 | 0.314 68 | 0.530 31 | 215.54 |
| 140 | 3.4613 | −111.90 | −83.005 | −0.393 95 | 0.314 36 | 0.529 28 | 219.60 |
| 145 | 3.3392 | −110.31 | −80.361 | −0.375 39 | 0.314 10 | 0.528 40 | 223.59 |
| 150 | 3.2255 | −108.72 | −77.721 | −0.357 49 | 0.313 88 | 0.527 64 | 227.49 |
| 155 | 3.1194 | −107.14 | −75.084 | −0.340 20 | 0.313 69 | 0.526 98 | 231.33 |
| 160 | 3.0202 | −105.56 | −72.451 | −0.323 48 | 0.313 53 | 0.526 41 | 235.10 |
| 165 | 2.9272 | −103.98 | −69.820 | −0.307 29 | 0.313 39 | 0.525 91 | 238.80 |
| 170 | 2.8398 | −102.40 | −67.192 | −0.291 59 | 0.313 27 | 0.525 46 | 242.45 |
| 175 | 2.7576 | −100.83 | −64.565 | −0.276 37 | 0.313 17 | 0.525 07 | 246.03 |
| 180 | 2.6800 | −99.254 | −61.941 | −0.261 58 | 0.313 08 | 0.524 72 | 249.57 |
| 185 | 2.6067 | −97.680 | −59.318 | −0.247 21 | 0.313 00 | 0.524 41 | 253.05 |
| 190 | 2.5374 | −96.108 | −56.697 | −0.233 23 | 0.312 94 | 0.524 12 | 256.48 |
| 195 | 2.4716 | −94.536 | −54.077 | −0.219 61 | 0.312 87 | 0.523 87 | 259.87 |
| 200 | 2.4093 | −92.964 | −51.458 | −0.206 35 | 0.312 82 | 0.523 64 | 263.21 |
| 210 | 2.2936 | −89.824 | −46.224 | −0.180 82 | 0.312 73 | 0.523 24 | 269.76 |
| 220 | 2.1885 | −86.686 | −40.993 | −0.156 48 | 0.312 66 | 0.522 91 | 276.15 |
| 230 | 2.0927 | −83.550 | −35.765 | −0.133 24 | 0.312 60 | 0.522 64 | 282.40 |
| 240 | 2.0050 | −80.415 | −30.540 | −0.111 01 | 0.312 55 | 0.522 40 | 288.50 |
| 250 | 1.9244 | −77.281 | −25.317 | −0.089 69 | 0.312 51 | 0.522 20 | 294.48 |
| 260 | 1.8500 | −74.149 | −20.096 | −0.069 21 | 0.312 48 | 0.522 02 | 300.33 |
| 270 | 1.7812 | −71.018 | −14.877 | −0.049 51 | 0.312 45 | 0.521 87 | 306.07 |
| 280 | 1.7174 | −67.887 | −9.6585 | −0.030 53 | 0.312 42 | 0.521 74 | 311.70 |
| 290 | 1.6579 | −64.757 | −4.4417 | −0.012 23 | 0.312 40 | 0.521 62 | 317.24 |
| 300 | 1.6025 | −61.628 | 0.77404 | 0.005 46 | 0.312 38 | 0.521 52 | 322.67 |
| 310 | 1.5507 | −58.499 | 5.9888 | 0.022 56 | 0.312 37 | 0.521 43 | 328.02 |
| 320 | 1.5021 | −55.371 | 11.203 | 0.039 11 | 0.312 35 | 0.521 35 | 333.27 |
| 330 | 1.4565 | −52.244 | 16.416 | 0.055 15 | 0.312 34 | 0.521 28 | 338.45 |
| 340 | 1.4135 | −49.116 | 21.628 | 0.070 71 | 0.312 33 | 0.521 21 | 343.55 |
| 350 | 1.3731 | −45.990 | 26.840 | 0.085 82 | 0.312 32 | 0.521 15 | 348.57 |
| 375 | 1.2814 | −38.174 | 39.867 | 0.121 77 | 0.312 30 | 0.521 03 | 360.81 |
| 400 | 1.2012 | −30.359 | 52.892 | 0.155 39 | 0.312 29 | 0.520 93 | 372.65 |

Figure 19: *Values of $c_p(T)$ at 1000 hPa for Argon vapour (above 87.178 K) from Tegeler, Span and Wagner (1999, Table 34).*

Table 15   Thermodynamic properties of argon–Continued

| Temperature | Density | Internal Energy | Enthalpy | Entropy | $c_v$ | $c_p$ | Velocity of Sound |
|---|---|---|---|---|---|---|---|
| K | mol/dm³ | J/mol | J/mol | J/mol K | J/mol K | J/mol K | m/s |

0.101325 MPa Isobar

| Temperature | Density | Internal Energy | Enthalpy | Entropy | $c_v$ | $c_p$ | Velocity of Sound |
|---|---|---|---|---|---|---|---|
| 84 | 35.449 | -4829.8 | -4826.9 | 53.39 | 21.31 | 42.63 | 852 |
| 86 | 35.150 | -4744.3 | -4741.4 | 54.39 | 21.06 | 42.90 | 839 |
| * 87.29 | 34.953 | -4688.7 | -4685.8 | 55.04 | 20.90 | 43.08 | 830 |
| * 87.29 | 0.14460 | 1044.8 | 1745.5 | 128.71 | | | 192 |
| 88 | 0.14337 | 1050.6 | 1757.3 | 128.84 | | | 186 |
| 90 | 0.13995 | 1071.3 | 1795.3 | 129.27 | | | 179 |
| 92 | 0.13664 | 1096.0 | 1837.5 | 129.74 | | | 177 |
| 94 | 0.13347 | 1122.5 | 1881.7 | 130.21 | | | 178 |
| 96 | 0.13044 | 1149.8 | 1926.6 | 130.68 | | | 179 |
| 98 | 0.12756 | 1177.4 | 1971.7 | 131.15 | | | 181 |
| 100 | 0.12481 | 1204.9 | 2016.8 | 131.60 | | | 183 |
| 102 | 0.12218 | 1232.3 | 2061.6 | 132.05 | | | 185 |
| 104 | 0.11967 | 1259.5 | 2106.2 | 132.48 | | | 187 |
| 106 | 0.11727 | 1286.5 | 2150.5 | 132.90 | | | 189 |
| 108 | 0.11497 | 1313.2 | 2194.6 | 133.31 | | | 191 |
| 110 | 0.11276 | 1339.8 | 2238.4 | 133.72 | 12.94 | 21.86 | 193 |
| 112 | 0.11065 | 1366.2 | 2282.0 | 134.11 | 12.89 | 21.76 | 195 |
| 114 | 0.10861 | 1392.5 | 2325.4 | 134.49 | 12.84 | 21.68 | 197 |
| 116 | 0.10665 | 1418.7 | 2368.7 | 134.87 | 12.79 | 21.60 | 199 |
| 118 | 0.10476 | 1444.7 | 2411.9 | 135.24 | 12.76 | 21.54 | 201 |
| 120 | 0.10295 | 1470.6 | 2454.9 | 135.60 | 12.72 | 21.48 | 203 |
| 122 | 0.10119 | 1496.5 | 2497.8 | 135.95 | 12.70 | 21.43 | 204 |
| 124 | 0.09950 | 1522.3 | 2540.6 | 136.30 | 12.67 | 21.38 | 206 |
| 126 | 0.09787 | 1548.0 | 2583.3 | 136.64 | 12.65 | 21.34 | 208 |
| 128 | 0.09629 | 1573.6 | 2626.0 | 136.98 | 12.64 | 21.31 | 210 |
| 130 | 0.09476 | 1599.2 | 2668.6 | 137.31 | 12.62 | 21.28 | 211 |
| 132 | 0.09328 | 1624.8 | 2711.1 | 137.64 | 12.61 | 21.25 | 213 |
| 134 | 0.09184 | 1650.3 | 2753.5 | 137.95 | 12.60 | 21.22 | 215 |
| 136 | 0.09046 | 1675.8 | 2796.0 | 138.27 | 12.59 | 21.20 | 216 |
| 138 | 0.08911 | 1701.3 | 2838.4 | 138.58 | 12.58 | 21.18 | 218 |
| 140 | 0.08780 | 1726.7 | 2880.7 | 138.88 | 12.57 | 21.16 | 220 |
| 142 | 0.08654 | 1752.1 | 2923.0 | 139.18 | 12.56 | 21.14 | 221 |
| 144 | 0.08531 | 1777.5 | 2965.3 | 139.48 | 12.56 | 21.13 | 223 |
| 146 | 0.08411 | 1802.8 | 3007.5 | 139.77 | 12.55 | 21.11 | 224 |
| 148 | 0.08295 | 1828.2 | 3049.7 | 140.06 | 12.55 | 21.10 | 226 |
| 150 | 0.08182 | 1853.5 | 3091.9 | 140.34 | 12.54 | 21.08 | 227 |
| 152 | 0.08072 | 1878.8 | 3134.0 | 140.62 | 12.54 | 21.07 | 229 |
| 154 | 0.07965 | 1904.1 | 3176.2 | 140.89 | 12.53 | 21.06 | 231 |

Figure 20:   *Values of $c_p(T)$ at 1013.25 hPa for Argon vapour (above 87.29 K) from Stewart and Jacobsen (1989, Table 15). Missing values between 87.29 K and 108 K can be computed from the diffeence in enthalpies $\Delta h = c_p \Delta T$, and thus $c_p = \Delta h/\Delta T$ (increasing from 15.9 J/K/mol at 87.29 K to 22.6 J/K/mol at 97−99 K, then decreasing toward 21.91 J/K/mol at 109 K).*

Table 15   Thermodynamic properties of argon–Continued

| Temperature
K | Density
mol/dm³ | Internal
Energy
J/mol | Enthalpy
J/mol | Entropy
J/mol K | $C_v$
J/mol K | $C_p$
J/mol K | Velocity
of Sound
m/s |
|---|---|---|---|---|---|---|---|
| 156 | 0.07861 | 1929.4 | 3218.3 | 141.17 | 12.53 | 21.05 | 232 |
| 158 | 0.07760 | 1954.6 | 3260.4 | 141.43 | 12.53 | 21.04 | 234 |
| 160 | 0.07661 | 1979.9 | 3302.5 | 141.70 | 12.53 | 21.03 | 235 |
| 162 | 0.07565 | 2005.1 | 3344.5 | 141.96 | 12.52 | 21.02 | 237 |
| 164 | 0.07471 | 2030.4 | 3386.6 | 142.22 | 12.52 | 21.02 | 238 |
| 166 | 0.07380 | 2055.6 | 3428.6 | 142.47 | 12.52 | 21.01 | 240 |
| 168 | 0.07291 | 2080.8 | 3470.6 | 142.72 | 12.52 | 21.00 | 241 |
| 170 | 0.07204 | 2106.0 | 3512.6 | 142.97 | 12.51 | 20.99 | 242 |
| 172 | 0.07119 | 2131.2 | 3554.6 | 143.22 | 12.51 | 20.99 | 244 |
| 174 | 0.07036 | 2156.4 | 3596.5 | 143.46 | 12.51 | 20.98 | 245 |
| 176 | 0.06955 | 2181.5 | 3638.5 | 143.70 | 12.51 | 20.98 | 247 |
| 178 | 0.06875 | 2206.7 | 3680.4 | 143.94 | 12.51 | 20.97 | 248 |
| 180 | 0.06798 | 2231.9 | 3722.4 | 144.17 | 12.51 | 20.96 | 250 |
| 185 | 0.06612 | 2294.7 | 3827.2 | 144.75 | 12.50 | 20.95 | 253 |
| 190 | 0.06436 | 2357.6 | 3931.9 | 145.30 | 12.50 | 20.94 | 256 |
| 195 | 0.06269 | 2420.4 | 4036.6 | 145.85 | 12.50 | 20.93 | 260 |
| 200 | 0.06111 | 2483.2 | 4141.2 | 146.38 | 12.50 | 20.92 | 263 |
| 205 | 0.05961 | 2545.9 | 4245.8 | 146.89 | 12.50 | 20.91 | 266 |
| 210 | 0.05818 | 2608.6 | 4350.3 | 147.40 | 12.49 | 20.91 | 270 |
| 215 | 0.05681 | 2671.3 | 4454.8 | 147.89 | 12.49 | 20.90 | 273 |
| 220 | 0.05551 | 2734.0 | 4559.3 | 148.37 | 12.49 | 20.89 | 276 |
| 225 | 0.05427 | 2796.7 | 4663.7 | 148.84 | 12.49 | 20.89 | 279 |
| 230 | 0.05308 | 2859.3 | 4768.2 | 149.30 | 12.49 | 20.88 | 282 |
| 235 | 0.05195 | 2922.0 | 4872.6 | 149.75 | 12.49 | 20.88 | 285 |
| 240 | 0.05086 | 2984.6 | 4976.9 | 150.19 | 12.49 | 20.87 | 288 |
| 245 | 0.04981 | 3047.2 | 5081.3 | 150.62 | 12.49 | 20.87 | 291 |
| 250 | 0.04881 | 3109.8 | 5185.6 | 151.04 | 12.49 | 20.86 | 294 |
| 255 | 0.04785 | 3172.4 | 5289.9 | 151.45 | 12.48 | 20.86 | 297 |
| 260 | 0.04693 | 3234.9 | 5394.2 | 151.86 | 12.48 | 20.86 | 300 |
| 265 | 0.04604 | 3297.5 | 5498.4 | 152.26 | 12.48 | 20.85 | 303 |
| 270 | 0.04518 | 3360.0 | 5602.7 | 152.64 | 12.48 | 20.85 | 306 |
| 275 | 0.04436 | 3422.6 | 5706.9 | 153.03 | 12.48 | 20.85 | 309 |
| 280 | 0.04356 | 3485.1 | 5811.2 | 153.40 | 12.48 | 20.84 | 312 |
| 285 | 0.04279 | 3547.6 | 5915.4 | 153.77 | 12.48 | 20.84 | 314 |
| 290 | 0.04205 | 3610.2 | 6019.6 | 154.13 | 12.48 | 20.84 | 317 |
| 295 | 0.04134 | 3672.7 | 6123.8 | 154.49 | 12.48 | 20.84 | 320 |
| 300 | 0.04065 | 3735.2 | 6227.9 | 154.84 | 12.48 | 20.83 | 323 |
| 310 | 0.03933 | 3860.2 | 6436.3 | 155.52 | 12.48 | 20.83 | 328 |
| 320 | 0.03810 | 3985.1 | 6644.6 | 156.19 | 12.48 | 20.83 | 333 |
| 330 | 0.03694 | 4110.1 | 6852.8 | 156.83 | 12.48 | 20.82 | 338 |
| 340 | 0.03585 | 4235.0 | 7061.0 | 157.45 | 12.48 | 20.82 | 344 |

Figure 21:   *Values of $c_p(T)$ at 1013.25 hPa for Argon vapour (above 87.29 K) from Stewart and Jacobsen (1989, Table 15, continued).*

[Figure]

Figure 22: *Unpublished figure plotted by me in 2016. Values of $c_p(T)$ for Argon at 1000 hPa (solid, liquid, gas). Datasets for vapour are from Tegeler, Span and Wagner (1999, TSW99, in red) and Stewart and Jacobsen (1989, SJ89, in blue). The asymptotic value 4.966 cal/K/mol corresponds to the "ideal-gas" value 520.3 = (5/2) 8.31432 / 0.039948 J/K/kg. Values of $c_p(T)$ for solid and liquid phases are from other papers and books.*

[Figure]

Figure 23:   *Unpublished figure plotted by me in 2020. The purple dashed curves computed for N2, O2 and H20 vapour by using Statistical and Quantum Physics compare well with the solid curves that correspond to the "calorimetric method" (third law and integration of $c_p(T')/T'$ from 0 K to T, sum of $L(T_k)/T_k$ for all changes of phases at $T_k$, add the Pauling-Nagle residual entropy at 0 K for H2O).*

**4    References**

• Bauer L.A. (1908). The relation between "potential temperature" and "entropy". *Phys. Rev.*, **26**, Series I, 177-183. (see: *The Mechanics of the Earth Atmosphere, a collection of translations by Cleveland Abbe*, 1910. Smithsonian Miscellaneous Collections. **Art. XXII**, 495-500).

• Durran D.R. and Klemp J.B. (1982). On the effects of moisture on the Brunt-Väisälä frequency *J. Atmos. Sci.*, **39**, 2152-2158.

• Ertel H. (1942). Ein neuer hydrodynamischer Wirbelsatz. *Meteorol. Z.*, **59**, 277-281.

• Feistel R., Wright D. G., Kretzschmar H.-J., Hagen E., Herrmann S., Span R. (2010). Thermodynamic properties of sea air. *Ocean Sci.*, **6**, 91-141. Doi: 10.5194/os-6-91-2010.

• Feistel R. (2018) Thermodynamic properties of seawater, ice and humid air: TEOS-10, before and beyond. *Ocean Sci.*, **14**, 471-502. Doi: 10.5194/os-14-471-2018.

• Hoskins BJ, McIntyre ME, Robertson AW. (1985). On the use and significance of isentropic potential vorticity maps. *Q. J. R. Meteorol. Soc.*, **111**, 877-946.

• Jacobsen R.T., Penoncello S.G., Lemmon E.W. (1997). *Thermodynamic Properties of Cryogenic Fluids*. The International Cryogenics Monograph Series. Springer, New York, Pp.312.

• Marquet P., Geleyn J.-F. (2013). On a general definition of the squared BruntVäisälä frequency associated with the specific moist entropy potential temperature. *Q. J. R. Meteorol. Soc.*, **139**, 85-100. Doi:10.1002/qj.1957, `https://arxiv.org/abs/1401.2379`

• Marquet P. (2014) . On the definition of a moist-air potential vorticity. *Q. J. R. Meteorol. Soc.*, **140**, 917-929 Doi:10.1002/qj.2182, `https://arxiv.org/abs/1401.2006`

• Marquet P. (2015). On the computation of moist-air specific thermal enthalpy. *Q. J. R. Meteorol. Soc.*, **141**, 67-84. Doi: 10.1002/qj.2335, `https://arxiv.org/abs/1401.3125`

• Marquet P., Geleyn J.-F. (2015). Formulations of moist thermodynamics for atmospheric modelling. Chapter 22 in: Parameterization of Atmospheric Convection (COST-ES0905 book) in R. S. Plant and J.-I. Yano, editors. Vol. 2: Current issues and new theories. *World Scientific, Imperial College Press*, October 2015.
Doi: `10.1142/9781783266913_0026`,
`http://www.worldscientific.com/doi/abs/10.1142/9781783266913_0026`,
`https://arxiv.org/abs/1510.03239`

• Marquet P. (2017). A third-law isentropic analysis of a simulated hurricane. *J. Atmos. Sci.*, **74**, 3451-3471. Doi: 10.1175/JAS-D-17-0126.1, `https://arxiv.org/abs/1704.06098`

• Marquet P. and Dauhut Th. (2018). Reply to the Comments of Olivier Pauluis to the paper "A Third-Law Isentropic Analysis of a Simulated Hurricane". *J. Atmos. Sci.*, **75**, p.3735-3747 Doi: 10.1175/JAS-D-18-0126.1, `https://arxiv.org/abs/1805.00834`

• Marquet P. (2019a). Le troisième principe de la thermodynamique ou une définition absolue de lentropie. Partie 1 : Origines et applications en thermodynamique. *La Météorologie*, **107**, 45-52. Doi: 10.4267/2042/70558 (see the English translation: "The third law of thermodynamics or an absolute definition for Entropy. Part 1 : the origin and applications in thermodynamics" `https://arxiv.org/abs/1904.11696`)

• Marquet P. (2019b). Le troisième principe de la thermodynamique ou une définition absolue de lentropie. Partie 2 : Définitions et applications en météorologie et en climat. *La Météorologie*, **107**, 45-52. Doi: 10.4267/2042/70559. (see the English translation: "The third law of thermodynamics or an absolute definition for Entropy. Part 2 : definitions and applications in Meteorology and Climate" `https://arxiv.org/abs/1904.11699`)

• Poisson S.D. (1833). *Traité de mécanique*. Seconde édition, Bachelier, Paris, 782 p.

• Schubert W, Ruprecht E, Hertenstein R, Nieto-Ferreira R, Taft R, Rozoff C, Ciesielski P, Kuo H-C. 2004. English translations of twenty-one of Ertels papers on geophysical fluid dynamics. *Meteorol. Z.*, **13**, 527-576.

• Stewart R.B., Jacobsen (1989). Thermodynamic properties of Argon from the triple point to 1200 K with pressures to 1000 MPa. *J. Phys. Chem. Ref. Data*, **18**, 639-798.

• Thomson W., (1862). On the convective equilibrium of temperature in the atmosphere. *Manch. Lit. Philos. Soc.*, **2**, 170-176.

---

## Referee Comment (RC2) · Anonymous Referee #2 · 13 Jul 2020

This manuscript constitutes a thorough and very well-presented revisit of a classical variable widely used in astmospheric science: potential temperature. The presentation is very pedagogical, with a brief and welcome account of the history of this quantity. The examination of the different effects and approximations are remarkably well exposed and discussed. The resulting text is perhaps a bit long and sometimes technical, but this is likely to become an important reference, at least for applications (such as middle atmospheric circulation) where a more exact defintion of potential temperature introduces significant differences. The study is very complete, with each approximation well explained, and with solutions for an approximate caluclation of the 'reference' potential temperature described for practical implementation. The conclusion include

the (expected) judgement that the conventional definition is quite acceptable for most uses, especially in the troposphere, but the possible cases where this reappraisal is relevant and useful are well and convincingly exposed. Overall, this is a well-written, rigorous investigation of a central quantity in dynamic meteorology. Publication after very minor revisions is advised.

- Abstract, line 1: 'changes of state' → should this be changed to or complemented by 'motions'; changes of state in my understanding mostly refers to thermodynamic changes of state, e.g. for a mixture of air and water. Potential temperature is already extremely useful for dry air undergoing displacements in the atmosphere, or even simply experiencing pressure changes.

- l42-44: meaning not clear, although I believe I know what is meant; the formulation is somewhat confusing

- l59: "Occasionally" means "on occasion, now and then", according to the Merriam-Webster dictionnary; it does not seem appropriate for this sentence. Suggestion: "Examples of definitions based on the potential vorticity include..."

- l185: has $\xi$ been introduced before, or does it make sense only upon reding Weigel et al 2016? If that is the case, perhaps it is sufficient to mention 'a coefficient factor, cf Weigel et al 2016'?

- l192 and 194: it seems odd that the same reference (WMO, 1966) both suggests the value of 1005 and 1011 J/(kg K)

- l224: the range of uncertainty displayed in figure 2 is an upper bound, obtained using the extreme values one may find in textbooks for $c_p$. A more plausible interval is probably 1004 - 1006, with key references like Holton (2004) and Emmanuel (1994) serving as classical references for one and the other extreme. Perhaps

the authors could indicate how this more limited range modifies the $\Delta c_p$ at 50 km (from 75 K down to ..?)

- l347: seductive $\rightarrow$ appealing? attractive? tempting?

- l495: should the authors recall what effects are dominant in the difference between ideal and real gas, or would this be too redundant with the first sections?

- l618: '. . . depending on the textbook consulted.' Perhaps recall the range of values, or at least refer to the table so the reader can quickly find the range of values (this table is useful and thought-provoking).
* * *

---

## Author Comment (AC1) · 26 Aug 2020

**Response to Reviewer Comments**

Manuel Baumgartner[1,2], Ralf Weigel[2], Allan H. Harvey[3],
Felix Plöger[4,5], Ulrich Achatz[6], and Peter Spichtinger[2]

[1]*Zentrum für Datenverarbeitung, Johannes Gutenberg University Mainz, Germany*
[2]*Institute for Atmospheric Physics, Johannes Gutenberg University Mainz, Germany*
[3]*Applied Chemicals and Materials Division, National Institute of Standards and Technology, Boulder, CO, USA*
[4]*Forschungszentrum Jülich GmbH, Institute of Energy and Climate Research (IEK-7), Jülich, Germany*
[5]*Institute for Atmospheric and Environmental Research, University of Wuppertal, Wuppertal, Germany*
[6]*Institut für Atmosphäre und Umwelt, Goethe-Universität Frankfurt, Frankfurt am Main, Germany*

August 26, 2020

**Contents**

**1 General Response**

We thank both reviewers for the effort they spent in preparing their reviews, their helpful comments and suggestions which led to a significantly improved manuscript.

This document contains our responses to the reviewer comments together with the output of "latexdiff" to highlight the changes made to the manuscript (blue text is added, red text is removed). After some general responses to points raised by the reviewers, we list all individual comments (pasted to this document in blue) together with our responses (in black).

1. We agree with Pascal Marquet, that not all values of $c_p$ from literature, as presented in our Table 1, are recent values, in particular the value $c_p = 994 \, \text{J} \, \text{kg}^{-1} \, \text{K}^{-1}$ from Wegener and Wegener (1935) is comparatively old and may appear outdated. However, the main goal of this table is to provide the reader a (naturally non-complete) synopsis of values that he or she may encounter in the literature. In our literature search, we focused on textbooks in atmospheric science, which are expected to be frequently used by researchers in this field in looking up values for (physical) constants. Thus, the minimum and maximum value within this table serve to encompass the range of suggested literature values, including older sources. We also agree that in particular the old values appear greatly exaggerated, however also more recent textbooks contain values that deviate significantly from the recommended value $1005 \, \text{J} \, \text{kg}^{-1} \, \text{K}^{-1}$ by the WMO (1966), where the lower bound $c_p = 1000 \, \text{J} \, \text{kg}^{-1} \, \text{K}^{-1}$ stems from Roedel and Wagner (2011) and the upper bound $c_p = 1010 \, \text{J} \, \text{kg}^{-1} \, \text{K}^{-1}$ from Chang et al. (2006), Tiwary and Williams (2019), and Brusseau et al. (2019). Arguably, these values are also rather extreme, but they are found in recently published literature sources, hence these values should be included in our synopsis.

   We adapted Figure 2b and included the absolute difference $\theta_{1000} - \theta_{1010}$ in addition to the difference $\theta_{994} - \theta_{1011}$, where the latter corresponds to the original curve shown in the initially submitted manuscript and is now designated as "historical".

   We reformulated the text to emphasize, that, on the one hand, the difference plots in Figure 2b only serve to indicate the possible range of values of the potential temperatures, depending on which value for $c_p$ is chosen or found in literature. On the other hand, these plots also illustrate the sensitivity of $\theta_{c_p}$ on comparatively small changes in $c_p$. We also emphasize, that the extreme values $\theta_{994}, \theta_{1000}, \theta_{1010}, \theta_{1011}$ are only

included for illustration purposes, while all comparisons in the sequel are made to $\theta_{1005}$, based on the recommended value $c_p = 1005\,\mathrm{J\,kg^{-1}\,K^{-1}}$ by the WMO (1966).

2. The main concept of the (dry air) potential temperature is the imaginary (dry air) adiabatic descent of an air parcel at absolute temperature $T$ and pressure $p$ down to pressure level $p_0$, where the air parcel would attain the potential temperature as its absolute temperature. The reviewer is correct that the absolute temperature of an air parcel is limited to "atmospheric values", e.g. not larger than about $350\,\mathrm{K}$. However, its potential temperature can attain significantly larger values as is already illustrated by the curves in Figure 2a, where values of up to $\sim 2000\,\mathrm{K}$ are visible for air parcels in the stratosphere. Although the curves in Figure 2a are based on the (historical) extreme values, the curve for $\theta_{1005}$ (corresponding to the recommended value) is in between the curves shown, hence also $\theta_{1005}$ attains values of up to $\sim 2000\,\mathrm{K}$ in the stratosphere.

Since our reference potential temperature $\theta_{\mathrm{ref}}$ is determined as the upper bound of the integral

$$\int\limits_{T}^{\theta_{\mathrm{ref}}} \frac{c_p^0(T')}{T'}\,\mathrm{d}T' \tag{1}$$

within the equation

$$R_a \log\left(\frac{p_0}{p}\right) = \int\limits_{T}^{\theta_{\mathrm{ref}}} \frac{c_p^0(T')}{T'}\,\mathrm{d}T'\,, \tag{2}$$

values of the integrand $\frac{c_p^0(T')}{T'}$ are required for temperatures up to $\sim 2000\,\mathrm{K}$, because we also need to expect that $\theta_{\mathrm{ref}}$ will attain comparably large values as $\theta_{1005}$. As specified in Lemmon et al. (2000), the parameterization $c_p^0(T')$ is accurate and valid up to $2000\,\mathrm{K}$. Therefore, the resulting potential temperatures $\theta_{\mathrm{ref}} \leq 2000\,\mathrm{K}$ are expected to be accurate. However, due to the division of the specific heat capacity $c_p^0(T')$ by the temperature $T'$ in the integrand, the influence of $c_p^0(T')$ on the integrand $\frac{c_p^0(T')}{T'}$ is diminished for large values of $T'$. Moreover, the parameterization $c_p^0(T')$ should extend in a physically consistent way to temperatures above $2000\,\mathrm{K}$. In summary, although the accuracy of the reference potential temperature for values above $2000\,\mathrm{K}$ is not guaranteed by Lemmon et al. (2000), we nevertheless expect these values to be close to the correct value.

3. Due to the major comment #7 by Pascal Marquet, we rewrote and restructured Section 7 completely (please also see our response to the respective major comment #7). As Section 7 within the original submission, also the new and restructured Section 7 is focused on the use of our new reference potential temperature $\theta_{\mathrm{ref}}$. Four typical uses of potential temperature are highlighted, where we show the (dis-)agreement between computations using either $\theta_{\mathrm{ref}}$ or the conventional $\theta_{1005}$.

- Section 7.1 is motivated by the very helpful major comment #7 by Pascal Marquet on our initial submission and illustrates a typical pitfall in the use of our new reference potential temperature $\theta_{\mathrm{ref}}$: a simple substitution of the occurence of $\theta$ by $\theta_{\mathrm{ref}}$ in a formula might lead to a wrong formula, if the derivation of this particular formula is based on the assumption of the constancy of $c_p$. Hence we showcase this pitfall at the example of the computation of the Brunt-Väisälä frequency and also indicate that we do not observe any significant deviations in the values of the Brunt-Väisälä frequency in using $\theta_{\mathrm{ref}}$ (with the correct formula) over the conventional potential temperature $\theta_{1005}$.

- Section 7.2 illustrates the (dis-)agreement of the values of Ertel's potential vorticity in a special case, where one computation uses $\theta_{\mathrm{ref}}$ and the other $\theta_{1005}$.

- Section 7.3 provides a general remark on the vertical sorting of (measurement) data with respect to intervals of potential temperature. Here, the difference $\theta_{\mathrm{ref}} - \theta_{1005}$ might mainly affect data at high altitudes, e.g. at stratospheric altitudes and beyond.

- Section 7.4 explores the effect of diabatic heating on the absolute and potential temperature of an air parcel. Only small differences are observed for the change of absolute temperature, either computed with the temperature dependent or the constant specific heat capacity. However, significant differences are observed in the corresponding rates for the change in the potential temperature. This difference could be significant for Lagrangian models based on isentropic coordinates, which are used for chemical transport models for the stratosphere.

We emphasize in the new Section 7, that a general assessment of the impact of using $\theta_{\mathrm{ref}}$ instead of $\theta_{1005}$ is not possible due to the wide range of possible applications of potential temperature in atmospheric science. Instead, our list is meant to show possible deviations in typical applications

and help the reader to make an informed decision whether it is worth to adopt the new potential temperature in the respective application.

**2 Response to Pascal Marquet**

**2.1 General response**

We thank Pascal Marquet for his thorough review, including numerous valuable suggestions and hints.

**2.2 Response to General Major Comments / Recommendations**

1. the authors present in section 3 a range of possible dry-air values of $c_p$ that appear to be greatly exaggerated, ranging from 994 to 1011 J/K/kg. I show in this review that the uncertainty interval must be much smaller (1004.5 to 1007.5 J/K/kg), which must imply impacts on values of $\theta$ about 7 times smaller than those considered at high altitude in the document. The authors should modify sections 2 to 5 and Figures 2, 3 and 4, by reducing the uncertainty on $c_p$ and by retaining only the more recent and realistic values.
   We agree that the values $994\,\mathrm{J\,kg^{-1}\,K^{-1}}$ and $1011\,\mathrm{J\,kg^{-1}\,K^{-1}}$ are rather extreme, but we included these for illustration purposes only. We reformulated certain parts of the text to emphasize this fact. Please also see our general response #1.

2. I show from copies of previously published papers, tables and figures that the observed values of $c_p(T)$ for T < 320 K contradict values above 1007.5 J/K/kg, those under 1004.5 J/K/kg and the (ideal gas) formulations of Lemmon et al. (2000) and Dixon (2007) considered in section 4 by the authors. Observed values of $c_p(T)$ for T < 320 K are rather consistent with the (real gas) NIST-REFPROP formulation considered in section 6 and with the IAPWS-TEOS10 formulation.
   We include our response to this item in the response to your general comment #5 below.

3. In this sense, the approach followed by the authors to calculate first values of $\theta_{\mathrm{ref}}$ from the ideal-gas formulation of $c_p(T)$ by Lemmon et al (2000), and then those of $\theta_{\mathrm{real}}$ for the real-gas NIST-REFPROP formulation, seems attractive, with however a comparison to irrelevant and too extreme constant values of 1011 and 994 J/K/kg in Figure 4 of the paper.

We include our response to this item in the response to your general comment #5 below.

4. Moreover the results of your section 6 seem strange to me, because the comparison of $\theta_{ref}$ deduced from the ideal-gas Lemmon's formulation (purple curve in your figure 3) with $\theta_{real}$ deduced from the real-gas NIST-REFPROP's formulation (yellow discs in your figure 3) gives very small differences on figures 8. Indeed, the differences $\theta_{real} - \theta$ref of less than 0.05 K for $\theta > 700$ K above 20 km (less than 0.007%) seem unrealistic and not consistent with differences of 4.5 J/K/kg (or 4.5 %) for $c_p(T)$ at 200 K, 2.8J/K/kg (or 2.8%) at 250K and 1.3 J/K/kg (or 1.3 %) at 300 K (values deduced from the yellow discs and the purple curve in your figure 3).

I guess that the relative differences $(\theta_{real} - \theta_{ref})/\theta_{ref}$ should be of the order of a few percent above 25 km and should increase with height, as indicated by a rough analysis of the differences between curves of your Figure 4b (to be checked by you, however, from direct computations and/or from a version with a linear scale of your figure 4b).

Differences of several percent between ideal-gas and real-gas formulations of $c_p(T)$ should lead to larger differences in the gap between $\theta_{real}$ and $\theta_{ref}$. This should result in a likely change in the conclusion in your section 6 and the use of formulations from IAPWS-TEOS10 (free) or INIST-REFPROP (to buy), rather than the analytical formula of Lemmon et al (2000, Eq.18, page 345) that is contradict by the values of $c_p(T)$ published in Table A2 (pages 366-367) of the same paper (see Fig.9 in section 3 bellow)
We include our response to this item in the response to your general comment #5 below.

5. In fact, after reflection and analysis of this aspect (4), this is probably a false problem. Indeed, everything seems to be explained by the fact that the major differences for your $\theta$ come from values of $c_p$ for highest $T$ temperatures, say between 400 K and 2000 K. This aspect is not documented in your figure 3, where the values of $c_p$ are only plotted up to 485 K.

The fact that the values of your $\theta_{real}$ and $\theta_{ref}$ are very close must be explained by a low sensitivity of your $\theta$ to values of $c_p$ for ambient temperatures (let's say those ranging from 200 K to 320 K and which define how the physical parameterizations should influence the weather parameters), with, on the other hand, a strong sensitivity of your $\theta$ to values of $c_p$ for temperatures above 400 K (temperatures that are not

observed in the real atmosphere but that intervene numerically in the calculation of your $\theta$ when passing from high altitudes where the pressure is very low and returning adiabatically towards the ground level through very high artificial temperatures).

Therefore, if you are interested in the values of $\theta$ calculated by an adiabatic evolution from a very low pressure $p$ to a (surface) pressure $p_0 = 1000$ hPa, you should better describe the accuracy of the values of $c_p(T)$ for $T > 400$ K.

Most of the issues (primarily expressed in your major comment #4) arise from the differences between real-gas and ideal-gas heat capacities shown in Figure 3. The key point is that the "real gas" (REFPROP) numbers shown on this graph are at 1013.25 hPa. The deviation between real gas and ideal gas is roughly proportional to the pressure, so at much lower pressures (for example, in the stratosphere) the real-gas effects are much smaller. Real-gas effects are also smaller at high temperatures (as can already be seen in Figure 3), so that, by the time a calculation reaches a pressure near sea level where nonideality could be significant, the (potential) temperature is high enough to make the behavior close to an ideal gas. This is why the real-gas effect on these calculations is small, as explained in the last paragraph of Section 6.

With regard to various formulations including those in REFPROP and TEOS-10, the paper of Lemmon et al. (2000) contains *two* real-gas air models (both of which use the same ideal-gas heat capacity that we use here). The most rigorous model treats air as a mixture of nitrogen, oxygen, and argon, using the reference EOS for each pure fluid. That model (implemented in REFPROP) is the real-gas model used here. Lemmon et al. (2000) also presents a simpler model where air is treated as a pseudo-pure fluid, and that model was used in the TEOS-10 package developed primarily for oceanographers. Since we use the more rigorous real-air model for our calculations in Section 6, we prefer not to introduce confusion by discussing alternative models such as the pseudo-pure fluid approach (especially since we find that real-gas effects are negligible in this context).

On his general comment #5, the reviewer is correct that the high-temperature behavior of $c_p^0$ is important. We already state the range of validity of the $c_p^0$ formulation (up to 2000 K) below Equation (19), but because of the importance we add the additional sentence: **Because the underlying calculations are based on rigorous statistical mechanics and accurate spectroscopic data, $c_p^0$ should be accurate**

**to within** $0.01\,\%$ **throughout this range, as discussed by Span et al. (2000).**

In addition, we correct an error in the caption of Figure 3 that may have caused confusion, changing $c_p$ to $c_p^0$ for the indicated parameterizations.

Concerning the influence of large values of $\theta$ on our computations, please also see our general response #1.

6. Another aspect should be addressed in this article. One of the goals of our community is to provide efficient and applicable numerical methods for climate and numerical weather prediction models. In this sense, it would be useful to quantify the iterative processes designed and tested in this article: what is the extra cost (in CPU) for the calculation of $\theta_{\mathrm{ref}}$ and $\theta_{\mathrm{real}}$ compared to the direct calculation $\theta_{c_p}$ for a constant $c_p$? (make this evaluation for example for a set of vertical columns of standard atmosphere)

   We appreciate this suggestion and added a short subsection on "Implementation aspects", being now Section 5.3. The additional computational overhead of computing $\theta_{\mathrm{ref}}$ highly depends on the number of iterations made with Newton's method. Restricting the attention to our suggested approximation $\theta^{(2)}$, seven additional computations, i.e. evaluations of mathematical expressions and function evaluations, need to be done. In this respect, a single computation of $\theta^{(2)}$ introduces an overhead of about seven, but the algorithmic complexity is constant, i.e. for all input values the required computational effort does not change.

7. For me, the most problematic aspect concerns the application you chose in section 7, by assuming that the squared Brunt-Väisälä frequency could be

   $$N^2 = \frac{g}{\theta}\frac{\partial\theta}{\partial z},$$

   where $\theta$ would be calculated by the particle method (by an adiabatic evolution from a very low pressure $p$ to a surface pressure $p_0 = 1000$ hPa).

   Differently, we recalled in Marquet and Geleyn (2013, MG13) that $N^2$ should be calculated from the local gradients of basic meteorological parameters (temperature and pressure if dry air is considered), and not from the variable $\theta$ that you study in your article (by an adiabatic evolution from a very low pressure $p$ to a surface pressure $p_0 = 1000$ hPa).

   In fact $N^2$ corresponds to adiabatic fluctuations of the density, before anything else. Accordingly, equations (B2) and (1) of MG13 applied to

dry air give the corresponding expression of $N^2$ as a function of local vertical gradients of density ($\rho$) and specific entropy ($s$):

$$N^2 = \frac{g}{\rho} \left.\frac{\partial \rho}{\partial z}\right|_s - \frac{g}{\rho}\frac{\partial \rho}{\partial z} = \left(-\frac{g}{\rho} \left.\frac{\partial \rho}{\partial s}\right|_p\right)\frac{\partial s}{\partial z},$$

where the first vertical derivative (of density with respect to $z$) is computed at constant entropy and the second vertical derivative (of density with respect to $s$) is computed at constant pressure. The local state equation $p = \rho R T$ and $\rho = p/(RT)$ with constant $R$ and $p$ implies

$$\left.\frac{\partial \rho}{\partial s}\right|_p = -\frac{\rho}{T} \left.\frac{\partial T}{\partial s}\right|_p.$$

The dry-air Gibbs equation writes $Tds = dh - dp/\rho$, with $dh = c_p(T)dT$ and with possibly $c_p(T)$ depending on absolute temperature. For constant pressure, this Gibbs equation reduces to $Tds|_p = c_p(T) \, dT|_p$, leading to $dT/ds|_p = T/c_p(T)$, and thus to $d\rho/ds|_p = -\rho/c_p(T)$. The squared dry-air Brunt-Väisälä frequency is therefore equal to

$$N^2 = \frac{g}{c_p(T)}\frac{\partial s}{\partial z}.$$

The dry-air Gibbs equation can then be used again to write $T\frac{\partial s}{\partial z} = c_p(T)\frac{\partial T}{\partial z} - (1/\rho)\frac{\partial p}{\partial z}$ which is valid for vertical oscillations. If moreover hydrostatic conditions prevail, then $\frac{\partial p}{\partial z} = -\rho g$, leading to

$$N^2 = \frac{g}{T}\left(\frac{\partial T}{\partial z} + \frac{g}{c_p(T)}\right).$$

This equation corresponds to the dry-air version of (22) in MG13, and it is Equation (1a) in the previous famous paper of Durran and Klemp (1982) about computations of the Brunt-Väisälä frequency (a key paper that you do not cite).

An expected result is that $N^2 = 0$ for the dry-air adiabatic lapse rate $\frac{\partial T}{\partial z} = -g/c_p(T)$.

The important finding for your study is that there is no need to use the gradient of any potential temperature for computing $N^2$. Really, only the vertical gradient of $T$ has to be calculated in (3), where it is possible to take into account the variations of $c_p(T)$ with the temperature you want to study in your paper. It is thus "possible", but not "mandatory",

to use (2) and a possible entropy formulation $s = c_p \ln(\theta) + const$ for the entropy to get the form (1) $N^2 = (g/\theta)\frac{\partial\theta}{\partial z}$ you have considered in your paper, but if and only if $c_p$ is a constant. And this is not possible if $c_p(T)$ depends on the temperature, with in this case the need to stick with the formulation (3) recalled above in terms of the gradient $\frac{\partial T}{\partial z}$.

The other important result here is that it is the local temperature that is involved in $c_p(T)$, so those between 1004.5 J/K/kg and 1007.5 J/K/kg for 200 K < T < 320 K, and especially not the ones at the higher temperatures that you studied in your paper to calculate $\theta_{\text{ref}}$ or $\theta_{\text{real}}$ , which are not needed for computing $N^2$ by (3).

It therefore seems to me that the application described in your section 7 is inaccurate, since the formulation (1) that you use for $N^2$ is not the right one (3). If so, can you show another application where values of your formulation of $\theta_{\text{ref}}$ or $\theta_{\text{real}}$ would intervene in meteorological science?

We thank the reviewer for this excellent comment. Indeed, we were misled by the often used formula

$$N^2 = \frac{g}{\theta}\frac{\partial\theta}{\partial z} \tag{3}$$

to compute the Brunt-Väisälä frequency, where we substituted our new $\theta_{\text{ref}}$ without reviewing the precise derivation of formula (3). In fact, after applying the correct formula

$$N^2 = \frac{g}{T}\left(\frac{\partial T}{\partial z} + \frac{g}{c_p(T)}\right), \tag{4}$$

the differences in the Brunt-Väisälä frequency profiles, and therefore also in the gravity wave breaking heights, vanish, which we described in Section 7 of our original submission. To highlight the impact of using formula (4) instead of the incorrect formula (3), we recreated Figure 9 from our manuscript; please find the new version as Figure 1 within this document below. Since the differences in the gravity wave breaking heights disappear, the initially documented changes were an artifact of using the incorrect formula.

This motivated us to offer the reader a warning on simply substituting $\theta_{\text{ref}}$ into a formula that contains the potential temperature, see the new Section 7.1. Moreover, we rewrote Section 7 completely to highlight other aspects of the use of potential temperature in atmospheric science, where we provide examples of the (dis-)agreement of computations using $\theta_{\text{ref}}$ instead of $\theta_{1005}$. Please see our general response #3 for further details.

8. My recommendation is that the document deserves acceptance only if the impacts described in section 7 concerning gravity waves are real.

   Therefore, the authors must provide evidence that it is indeed their formulations of $\theta_{\text{ref}}$ or $\theta_{\text{real}}$ (obtained by an adiabatic evolution between the pressures $p$ and $p_0$ ) that intervenes in the Brunt-Väisälä frequency formula, and not the local vertical gradients of temperature and pressure derived in Durran and Klemp (1982) and Marquet and Geleyn (2013).

   If the authors can provide this evidence, then their paper would merit to be published subject to taking all the major recommendations and specific comments into account, or explaining why they do not need to take them into account.

   Even though the impact on the gravity wave breaking height was not real, we are convinced that our re-assessed potential temperature might prove useful for other upper atmospheric applications, i.e. within the stratosphere and beyond. In our newly written Section 7, we indicate three typical applications of the potential temperature, which might be affected by using the new reference potential temperature. Please see our general response #3 for details on the rewritten Section 7 and our response to your general comment #7 above.

**2.3 Response to Specific Comments**

- **Line 1:** add *dry* in: "...it is conserved for *dry* air's adiabatic..."
  We added the word "dry".

- **Lines 10 to 22:** I do not have access to Wegener's book (1911) and I confess that I was not aware of Köppen's oral contribution (1888). I have cited only the contributions of von Helmholtz and von Bezold in my papers (Marquet 2011, 2017, 2019b, Marquet and Dauhut 2018). I have been able to verify, however, Kutzbach's sentence (1979, page 143) in which Köppen's (1988) oral contribution is mentioned (see the excerpts in the Figure 1 in section 3 bellow). However, the title of the 1888 lecture of Köppen is written in Kutzbach (1979) as: "Ueber die Luftmischung und potentielle Temperatur", which might be different from the one in your bibliography: "Über Luftmischung..."? Moreover, I have not found the paper (or a copy of this lecture) of Köppen: do you have a copy of this lecture, or are you just citing the sentence of Kutzbach? Finally, I do not understand why you cite the URL: http://snowcrystals.com/?
  An online version of Wegener's book from 1911 (written in german)

[Figure]

**Figure 1:** The same as Figure 9 from the originally submitted manuscript, where now the Brunt-Väisälä frequency is computed using the correct equation (4).

is available at `https://ia800404.us.archive.org/24/items/bub_gb_kWtUAAAAMAAJ/bub_gb_kWtUAAAAMAAJ.pdf`. Within the footnote on page 111 of this book, Köppen's oral contribution is mentioned similarly as in Kutzbach's book. Wegener cites the title of Köppens talk as "Über Luftmischung und potentielle Temperatur, in Anlehnung an die neueste Abhandlung von Herrn v. Helmholtz" and we refer to his citation. Given that "die" is a German article which does not modify the meaning of the sentence and we also do not have a copy of Köppen's talk, we prefer to adopt the title as written by Wegener. The citation of the url `http://snowcrystals.com/` is due to a false entry in the BibTeX file; we eliminated this URL.

- **Lines 10 to 28:** It would be useful to refer to the papers by Poisson (1833) and Thomson (1862-65) who had clearly imagined, before von Helmoltz and von Bezold in 1888, this idea of adiabatic variation on the vertical and the calculation of temperature for an air particle brought back to the surface (see section 5 of Marquet and Dauhut, 2018, and Marquet 2019b). I give copies of these articles on Figures 2 and 3.
  We are grateful for the references to the works of Poisson and Thomson which we were not aware of. We added these works to our text.

- **Lines 10 to 28:** It would be useful to refer to Bauer's paper (1908-1910), where the link between entropy and the potential temperature of dry air is made for the first time (see citations in Marquet 2011, Marquet and Dauhut 2018 and Marquet 2019b). I give copies of this paper of Bauer on Figures 4 and 5
  We included the reference to Bauer's work.

- **Lines 53 to 61:** You should mention the basic references for the definition and the use of $PV(\theta)$: Ertel (1940) and Hoskins (1987) at least (see also Schubert et al. 2004 cited in Marquet 2014).
  We added these references.

- **Lines 65-67:** The studies of the moist-air entropy by Hauf and Höller (1987) and Marquet (2011) do not start from the Gibbs' equation "$Tds = dh - dp/\rho - \sum_n n\mu_n dq_n$". On the contrary, they start from the moist-air entropy $s = \sum_n q_n s_n$ expressed as the weighted sum of the entropies $s_n$ for its $n = 0, ..., 3$ constituents (dry air, water vapour, liquid water and ice) with concentrations $q_n$ (specific contents).
  We thank for this clarification. We reformulated the sentence to clarify that the first law of thermodynamics together with Gibbs' equation is

used to derive the equation for the moist-air entropy, as stated in the introduction of Hauf and Höller (1987).

- **Lines 68-69:** The assumption of "local equilibrium" and use of latent heats release (of vaporization $L_v$ and sublimation $L_s$ ) are also included in the definition of Hauf and Höller (1987) and Marquet (2011), not only in the formulation of Emanuel (1994).
  We reformulated the sentences accordingly.

- **Lines 69-70:** It is not true that "These formulations always rely on the assumption of reversible processes (i.e. conserved entropy)". On the contrary, the formulation $s(\theta_s)$ of Marquet (2011, ...) makes it possible to measure and quantify the losses or increases in moist-air entropy associated with irreversible processes such as the removal of precipitations that you mention. See in particular Eq.(59) in Marquet and Geleyn (2015), where the change in moist-air entropy associated with pseudo-adiabatic (von Bezold, 1888) processes writes:

$$ds = c_{pd}\frac{d\theta_s}{\theta_s} = (s - s_l)\left(\frac{-dr_{sw}}{1 + r_{sw}}\right).$$

  We reformulated the paragraph accordingly to clarify that also the case of irreversible processes is included in this formulation.

- **Lines 82-83:** You say: "the potential temperature is commonly used as a prognostic variable in numerical models for the formulation of the energy equation". Could you explain in which models $\theta$ is used as a prognostic variable? As far as I know, the prognostic variables associated with energy is either the temperature $T$ or the combination $c_pT$, with the moist-air definition for $c_p$. In particular, your reference to Richardson et al (2007) on line 105 seems incorrect, since page 25 of this article the equations are: "$DT/Dt = F_q$" or "$\frac{\partial T}{\partial t} = ... + F_q$" or "$\frac{\partial \rho T}{\partial t} = ... + \rho F_q$".
  Indeed, the equations on page 25 of Richardson et al. (2007) are formulated without indicating the potential temperature. However, in their Section 2, the authors refer to the WRF model where the potential temperature is used as a prognostic variable (also see Skamarock et al., 2005; Skamarock and Klemp, 2008). We contacted the developers of the "planetWRF" model and they confirmed that the governing model equations were not changed, thus the symbol $T$ used on page 25 of Richardson et al. (2007) appears to be a typographical error. Apart from the WRF-model, also the governing equations within the ICON model (employed by the German

Weather Service) are based on the (virtual) potential temperature (e.g., Zängl et al., 2015), which, in the case of dry conditions, reduces to the dry air potential temperature. An example of such a case is the upper-atmosphere extension of the ICON model (Borchert et al., 2019). We reformulated the paragraph accordingly and included the references. In addition to these numerical weather forecasting models, we now mention two chemical transport models in our new Section 7, which are Lagrangian models and use a formulation based on potential temperature.

- **Lines 105:** You say: "it was pointed out by Li and Chen (2019) that this approach could suffer from not accounting for the temperature dependence of the isobaric specific heat capacity $c_p$ of the respective atmospheres gas composition". I spent some time checking this out in Li and Chen (2019), and I find (page 2): "Furthermore, the expressions of potential temperature and equivalent potential temperature become complicated when the heat capacity of the atmosphere varies with temperature or when multiple condensing species exist in the atmosphere." Here as elsewhere, could you quote the pages and/or equations corresponding to your citations, to help the reader find his way around in articles or books with very many pages?
  Unfortunately, we erroneously cited the wrong article. The correct article is Li et al. (2018), where the authors characterize the moist adiabats, especially for atmospheres of other planets. Within their derivation, they explicitly take the temperature dependency of the specific heat capacity into account. In addition, they stress the importance to incorporate temperature dependent versions of the heat capacities for Jupiter: "Because the measured absolute brightness temperature is precise to about a few percent and the limb darkening is precise to 0.1 % (Janssen et al. 2017), traditional Jovian thermodynamics—assuming constant heat capacity and small mixing ratios of condensates—needs to be carefully reviewed and refined according to the requirement of the new instrument." We corrected the references and reformulated the paragraph. Moreover, we went again over the manuscript to include numbers of pages or equations for certain references.

- **Lines 117:** It is customary, at the end of the introduction, to present the outline of the article, with a summary of the content of each forthcoming sections. This should be included at the end of your Section 1.
  We followed this suggestion and included an outline of the subsequent sections.

- **Lines 134:** Your value for $R_a = R/M_{\mathrm{mol},a}$ is known with $R$ given up to $\pm 0.0001$ I believe? You could retain the value 8.31446 for example? Anyhow you have to give the resulting value $R_a = 287.115$ at least, with perhaps the associated precision $\pm 0.005$?

  We followed this suggestion and specified an interval within which the value of $R_a$ is contained, when the division is carried with the indicated precision for $M_{\mathrm{mol},a}$.

  Recently, the value of the molar gas constant $R$ was defined to have the indicated value, see `http://physics.nist.gov/constants`, therefore there is no uncertainty in the value of $R$. We hint the reader on this fact in a footnote.

- **Lines 183:** It was indeed indicating by WMO that the variability of $c_p$ ranges from 994 J/K/kg to 1011 J/K/kg. But the real recommendation is rather a value close to 1005 J/K/kg, in line with the values presently used in most General Circulation (GCM) and Numerical Weather Prediction (NWP) models:

  - - - - - - - -‖- - - - - - - - - - - - - - - - - - - - - - - - - - - - - - - - - - - - -
  $c_p$ (J/K/kg) GCM and/or NWP models
  - - - - - - - -‖- - - - - - - - - - - - - - - - - - - - - - - - - - - - - - - - - - - - -
  1005.0       Unified-Model (UKMO, UK, from Adrian Lock)
  1005.0       COSMO (DWD, Germany, from Dmitrii Mironov)
  1004.7       IFS (ECMWF, Reading, UK, from "sucst.F90")
  1004.7       ARPEGE (Meteo-France, Toulouse, France, from "sucst.F90")
  1004.7       AROME (Meteo-France, Toulouse, France, from "sucst.F90")
  1004.7       Meso-NH (L.A.+Meteo-France, Toulouse, France, from "sucst.F90")
  1004.7       LMD-Z (IPSL, Paris, France, from "suphec.F90")
  1004.6       ICON (DWD, Germany, from Dmitrii Mironov)
  1004.6       GFS (USA, from "physcons.f")
  - - - - - - - -‖- - - - - - - - - - - - - - - - - - - - - - - - - - - - - - - - - - - - -

  We thank for the compilation of the values of $c_p$ used by several General Circulation and Numerical Weather Prediction models. Indeed, the WMO recommends the value $1005 \, \mathrm{J \, kg^{-1} \, K^{-1}}$, hence we always compare our results to the resulting potential temperature $\theta_{1005}$, see also our general response #1.

- **Lines 191-192:** These old WMO values of 994 J/K/kg and 1011 J/K/kg are too extreme and unrealistic, because they are not used in any current GCM and NWP model. Or could you indicate the models where these

values might be used?

We refer the reviewer to our response to the previous comment as well as to our general response #1.

- **Page 8, Table 1:** The values of 994 J/K/kg, 1000 J/K/kg, 1003 J/K/kg and 1011 J/K/kg do not seem relevant.

  - The value 994 J/K/kg comes from an old book I couldn't find, and the accuracy of the data obtained before 1933 can be questioned. This is like the measurement of the speed of light, the accuracy of which cannot be the "meeting of all possibilities", including for example the measurements of Romer and Huygens in 1675 (220, 000 km/s), Bradley in 1729 (301, 000 km/s), Fizeau in 1849 (315, 000 km/s) or Foucault in 1862 (298, 000 km/s)? It is the same for the measurement of the numerical values of $\gamma = c_p/c_v$ for diatomic gases, where the value of 1.421 retained by Poisson in 1833 or of 1.41 by Thomson in 1862 cannot be compared with the modern value of $7/5 = 1.40$? It is the same for the measurement of absolute scale of temperature, with a constant corresponding to 267 K in Gay-Lussac (1802) and Carnot (1824), to 273.22 K in Thomson (1848), before to be presently fixed to 273.15 K (see the review in Marquet, 2019a).

  - The value of 1000 J/K/kg attributed to Valis (2009) seems to be easily questionable: see the legend in Figure 6 in section 3 bellow.

  - I don't know where the value of 1003 J/K/kg published in Tripoli and Cotton (1981) comes from. But one can also have doubts about their values of $c_p$ for ice (2100 J/K/kg instead of 2106 J/K/kg) and liquid water (4187 J/K/kg instead of 4218 J/K/kg), with important differences for both dry air, liquid water and ice from the values commonly used in GCM and NWP model.

  - Other than the mention in the WMO recommendations, I have never seen an application of the value 1011 J/K/kg. Could you indicate such an application of the value $c_p = 1011$ J/K/kg for dry air?

We agree, the value $1000\,\mathrm{J\,kg^{-1}\,K^{-1}}$ from Vallis (2006) seems to be intended as a rough order of magnitude estimate of $c_p$, therefore we removed this value from our Table. However, the same value $1000\,\mathrm{J\,kg^{-1}\,K^{-1}}$ nevertheless appears in the recent textbook by Roedel and Wagner (2011).

The value $1003\,\mathrm{J\,kg^{-1}\,K^{-1}}$ from Tripoli and Cotton (1981) appears in their appendix. Since our Table 1 is intended to provide a synopsis of values from literature, we prefer to keep this value within the table.

We cannot provide an application where the value $1011\,\mathrm{J\,kg^{-1}\,K^{-1}}$ is used. However, the document WMO (1966) explicitly states the values $1003\,\mathrm{J\,kg^{-1}\,K^{-1}}$ and $1011\,\mathrm{J\,kg^{-1}\,K^{-1}}$ as the minimum and maximum "range of actual values", hence in our opinion such a range should be part of our synopsis of literature values.

We additionally refer the reviewer to our general response #1.

- **Lines 194-199 and Figure 2:** Assuming new extreme values of 1004.5 J/K/kg and 1007.5 J/K/kg (later demonstrated), I was able to redo your figures 2 (a) and (b) with the same US standard atmosphere profile: see Figure 7 in section 3 bellow, with indeed the same difference $\Delta\theta_{c_p} = \theta_{994} - \theta_{1011}$ (in black) as in your paper. The new differences $\theta_{1004.5} - \theta_{1007.5}$ (in red) are much smaller, by an order of magnitude or so (divided by a factor of about 5 to 7). The new differences are less than 2 K at 20 km, 5 K at 35 km and 14 K (instead of 75 K) at 50 km. These new differences $\theta_{1004.5} - \theta_{1007.5}$ may modified your comments and conclusions in your section 3.

  We appreciate the effort for reproducing this figure. Since the range of extreme values in our Table 1 should only reflect the range provided in the literature, we included the curve for $\theta_{1000} - \theta_{1010}$ in the updated Figure 2b. These curves only serve to illustrate the range of values of $\theta_{c_p}$ for the values indicated in the synopsis of Table 1. Moreover, these difference plots also serve to indicate the sensitive response of $\theta_{c_p}$ to even small perturbations in the value for $c_p$. We also refer the reviewer to our general response #1.

- **Section 4, Lines 226-260 and Figure 3:** I disagree with many of the points you've drawn on your figure 3. So I redid your figure 3 by deleting the old and questionable data (see Figure 8 in section 3 bellow). I kept the values of 1004 J/K/kg, 1004.832 J/K/kg, 1005 J/K/kg and 1005.7 J/K/kg, the data of Vassermann et al (1966) and NIST-REFPROP as well as the two curves of Lemmon et al (2000) and Dixon (2007). This new figure shows that constant values of $c_p$ between 1004.5 J/K/kg and 1007.5 J/K/kg agree with the selected points for the range of temperatures observed in the atmosphere (say 200 to 320 K). The two curves of Lemmon et al (2000) and Dixon (2007) are retained here because they are valid for the approximation of ideal gases and allow to measure the differences with formulations for real gases, such as Vassermann et al (1966) and NIST-REFPROP. The impact of real gases properties on $c_p$ increases with decreasing values of $T$ bellow 260 K, and is larger than 4 J/K/kg at 200

K.

We agree that the relatively old measurement data may not be very accurate and we stress this aspect in our text. Apart from the possible limited accuracy of the old measurements, these data are not always given for pressures of "about one atmosphere", i.e. about 1000 hPa, but also include measurement data for lower pressures. In any case, this figure follows two purposes: On the one hand, it provides an impression of the available measurement data, even though these might stem from older sources. On the other hand, even these (maybe more uncertain) measurement data nicely show that $c_p$ should indeed be considered as a function of temperature which is the key aspect of that subsection. We reformulated some sentences to point out our intension more clearly.

We also refer the reviewer to our response to his general comment #5.

- **Lines 245, legend of Fig 3, lines 315-321 and Eqs.(18) and (19):** It should be mentioned that your formula (18) with the coefficients (19) of Lemmon et al (2000) disagrees with the observed values given in Table A2 of the same article Lemmon et al (2000). And indeed, while formula (18) leads to decreasing values of $c_p(T)$ for decreasing $T$, the values of $c_p(T)$ in Table A2 show a minimum around 250 K and become increasing for decreasing temperatures up to 81.72 K (see Figs.9 and 10 in section 3 bellow). It should also be mentioned that your equation (18) corresponds to equation (18) (page 345) in Lemmon et al (2000).

  We indicated the number of the equation for the parameterization $\frac{C_p^0(T)}{R}$ in the work by Lemmon et al. (2000).

  Table A2 in the Lemmon et al. (2000) paper is computed with the pseudo-pure fluid model (see reply to your major comment #5). It is therefore not directly relevant to compare with ideal-gas calculations (the same is true of the real-gas tables of $N_2$, $O_2$, and Ar discussed near the end of the review). We do show real-gas results (at 1013.25 hPa) from the more rigorous of the two models of Lemmon (via REFPROP), which should be close to those from the simpler pseudo-pure model (for example, showing a minimum near 250 K).

- **Lines 246 and 325-329:** You should mention that the equation of Dixon (2007, p.376) used to compute the dry-air value $c_p(T)$ plotted in your Fig.3 is
  $$c_p(T) = 1002.5 + 275 \cdot 10^{-6}(T - 200)^2 \, \text{J/K/kg}$$

(see Figure 11 in section 3 bellow).
We followed the suggestion and included the relevant page in Dixon's book, the stated accuracy and the formula.

- **Lines 356 / Eq.(11):** The gaz constant "$R_a$" is missing before the integral $\int_{p_0}^{p} dp'/p'$
  We added the missing gas constant.

- **Lines 356 / Eq.(21), Lines 359 / Eq.(22), Lines 374 / Eq.(23), Lines 411 / Eq.(25), Line 426 and 429, Line 691 / Eq.(C1), Line 693 / Eq.(C2):** You should used the same dummy variable "$T'$" as in your Eq.(11) line 153 ($\int_{T_0}^{T} dT'/T'$) to write all the integrals of the kind $\int_{\theta}^{T} c_p(T')dT'/T'$ . The use of the dummy variable "$z$" can lead to unfortunate confusion with the altitude variable, which is then used in the rest of your paper to describe the true vertical coordinate.
  We followed this suggestion and used the dummy variable $T'$ instead of $z$.

- **Page 11 / Fig 3:** I have plotted in Figure 12 (top, see section 3 bellow) the equivalent of your Figure 3, but with different formulations that correspond to observed ("real gases") values of $c_p(T)$, with a zoom (Figure 12 bottom) around the usual atmospheric temperatures.

  I first reported (from your Fig.3) the points of your calculations made with the (paid) application of NIST-REFPROP. These NIST-REFPROP values are comparable to those I have computed with the (free) SIA software (http://www.teos-10.org/software.htm) corresponding to the IAPWS-2010 (Feistel et al, 2010) and TEOS-10 (Feistel, 2018) formulations. There is a similar minimum $c_p(T) \approx 1005.5$ to $1005.7$ J/K/kg at around 250 K and with the same higher values of about 1007 J/K/kg at 320 K and 1006.7 J/K/kg at 200 K.

  The values published in Table A2 of Lemmon et al. (2000) are fairly comparable to those of NIST-REFPROP and IAPWS-TEOS10, with a similar minimum of $c_p(T)$ at around 250 K.

  The same applies to the values of $c_p(T)$ for N2 and O2 published in Marquet (2015), with the values for dry air completed with the values of $c_p(T)$ for Argon.

  The minimum of $c_p(T)$ for N2 is at around 290 K in both Stewart and Jacobsen (1989, Table 5.73, see Fig.13 bellow) and Span et al. (2000, page 1410, see Fig.14 bellow). The resulting figure 5 for N2 published in Marquet (2015) is recalled in Fig.15 bellow.

The minimum of $c_p(T)$ for O2 is at around 220 K in Jacobsen et al. (1997, Table 5.79, see Figs.16 and 17). The resulting figure 4 for O2 published in Marquet (2015) is recalled in Fig.18 bellow.

Values of $c_p(T)$ for Argon increases for decreasing $T$ for both Tegeler et al. (1999, Table 34, see Fig.19) and Stewart and Jacobsen (1989, Table 15, Figs.20 and 21). The unpublished Fig.22 plotted bellow shows that values for Tegeler et al. (1999) and Stewart and Jacobsen (1989) fairly coincide for $150 < \text{T} < 300$ K.

The unpublished Fig.23 bellow shows that it is equivalent to use $c_p(T)$ computed for N2, O2 and H20 vapour by using Statistical and Quantum Physics (dashed lines) or by the "calorimetric method" (third law and integration of $c_p(T')/T'$ from 0 K to T , sum of $L(T_k)/T_k$ for all changes of phases at $T_k$ , add the Pauling-Nagle residual entropy at 0 K for H2O). It thus appears that it is for these temperature-dependent values of $c_p(T)$ for gases that the agreement between the calorimetric and quantum methods can be obtained, an agreement which is not obtained with "ideal gas" formulations.

I have also plotted on Figure 12 bellow the constant values used in many GCM and NWP models (1004.6, 1004.7, 1005 J/K/kg, depicted by coloured horizontal dashed lines). It appears, considering all these values of $c_p$ constant or dependent on $T$, and in the range of atmospheric temperatures ($200 < \text{T} < 320$ K), that the imprecision on $c_p(T)$ is between 1004.5 and 1007.5 J/K/kg. These extreme values have been used earlier in this review to plot several figures, instead of the (old) WMO extreme values 994 and 1011 J/K/kg you used in your study.

Regarding the search for an accurate average value $c_p(T) \approx c_p^0$, it appears that $c_p^0 \approx 1005.8 J/K/kg$ could be more realistic (for $200 < \text{T} < 320$ K) than those presently used in GCM and NWP models (1004.6, 1004.7, 1005 J/K/kg).

However, the impact of these new formulations ($c_p(T)$ or $c_p^0$ ) should be small in our CMGs and NWP models. Moreover, taking into account the dependence of $c_p(T)$ on temperature, not only for dry air but also for water vapor, liquid water and ice, would greatly complicate the writing of the physical parameterizations of these models, and would greatly increase the cost of these physical parameterizations.

It is true that, if only the heat capacity at atmospheric temperatures (e.g. from $200\,\text{K}$ to $320\,\text{K}$) is of interest, a constant value near $1005\,\text{J}\,\text{kg}^{-1}\,\text{K}^{-1}$ might be adequate. However, for calculations in the upper atmosphere,

the potential temperatures become much larger, and the assumption of the constancy of $c_p$ becomes increasingly erroneous. This may be considered as the main reason for the increasing deviation between $\theta_{\text{ref}}$ and one of $\theta_{c_p}$ at increasing altitude.

The reviewer is correct that accounting for this effect would complicate atmospheric models. We are not advocating that all atmospheric modeling use our more rigorous reference potential temperature. The purpose of our paper is to document the error caused by the typical simplifying assumptions on calculations of the potential temperature so that scientists can make an informed decision about whether it is worthwhile to implement the more rigorous calculation in their particular context. We now emphasize this aspect in the introductory paragraph of the new Section 7.

**3 Response to Reviewer #2**

- Abstract, line 1: 'changes of state' → should this be changed to or complemented by 'motions'; changes of state in my understanding mostly refers to thermodynamic changes of state, e.g. for a mixture of air and water. Potential temperature is already extremely useful for dry air undergoing displacements in the atmosphere, or even simply experiencing pressure changes.
  We thank for this hint, but as the thermodynamic "state" of a fluid already refers to temperature and pressure, we prefer to leave the formulation as it is.

- l42-44: meaning not clear, although I believe I know what is meant; the formulation is somewhat confusing
  We rephrased this sentence.

- l59: "Occasionally" means "on occasion, now and then", according to the Merriam-Webster dictionnary; it does not seem appropriate for this sentence. Suggestion: "Examples of definitions based on the potential vorticity include..."
  We followed the suggestion and rephrased the sentence to avoid the use of "occasionally".

- l185: has $\xi$ been introduced before, or does it make sense only upon reding Weigel et al 2016? If that is the case, perhaps it is sufficient to mention 'a coefficient factor, cf Weigel et al 2016'?
  The symbol $\xi$ indeed only makes sense upon reading the reference. Therefore, we followed the suggestion and eliminated the symbol.

- l192 and 194: it seems odd that the same reference (WMO, 1966) both suggests the value of 1005 and 1011 J/(kg K)
  The WMO (1966) offers three values in total:

  – a recommended value $1005 \, \mathrm{J \, kg^{-1} \, K^{-1}}$,

  – the minimum "range of actual values" $1003 \, \mathrm{J \, kg^{-1} \, K^{-1}}$,

  – the maximum "range of actual values" $1011 \, \mathrm{J \, kg^{-1} \, K^{-1}}$

  where the indicated "range of actual values" might be interpreted as to highlight the uncertainty on the precise values at that time, although this is not explicitly stated in WMO (1966). We added a subsentence to indicate, that $1011 \, \mathrm{J \, kg^{-1} \, K^{-1}}$ represents the upper limit in the reference WMO (1966). We also refer to our general response #1.

- l224: the range of uncertainty displayed in figure 2 is an upper bound, obtained using the extreme values one may find in textbooks for $c_p$. A more plausible interval is probably 1004 - 1006, with key references like Holton (2004) and Emmanuel (1994) serving as classical references for one and the other extreme. Perhaps the authors could indicate how this more limited range modifies the $\Delta c_p$ at 50 km (from 75 K down to ..?)
  Following also a comment of Reviewer #1, we included a second curve in Figure 2b, showing the absolute difference $\theta_{1000} - \theta_{1010}$. We agree, that these values of $c_p$ are still quite extreme, but these values are suggested in the literature. In any case, we reformulated the text to emphasize, that these absolute differences only serve to indicate the possible range of deviations between the $\theta_{c_p}$ based on the range of values of $c_p$ found in the literature. Moreover, these curves illustrate the sensitive response of $\theta_{c_p}$ to even small perturbations in the value of $c_p$. Moreover, we also emphasize, that all comparisons are made in reference to $\theta_{1005}$, based on the recommended value by WMO (1966).

- l347: seductive → appealing? attractive? tempting?
  We agree that the word "seductive" is not appropriate; we substituted it by "tempting".

- l495: should the authors recall what effects are dominant in the difference between ideal and real gas, or would this be too redundant with the first sections?
  We thank for this suggestion and included a reference to Section 4 where the differences are discussed.

- l618: '...depending on the textbook consulted.' Perhaps recall the range of values, or at least refer to the table so the reader can quickly find the range of values (this table is useful and thought-provoking).
  We followed this suggestion and recalled the minimum and maximum value from recent textbooks. In addition, we included a reference to Table 3.

[revised manuscript text omitted]

---

## Referee Report (RR1)

**Review by Pascal Marquet of the revised paper "acp_2020_361"**

entitled: *Reappraising the appropriate calculation of a common meteorological quantity: Potential Temperature.*

by Manuel Baumgartner, Ralf Weigel, Ulrich Achatz, Allan H. Harvey, and Peter Spichtinger.

**1   General Comments / Recommendation**

I still disagree with the use of too extreme and unrealistic values of $C_p$. Indeed, the values $C_p = 1010$ in the new cited papers Chang et al (2010) and Tiwary Williams (2019) are only indicative (it is a mere scale value for computing the LMO for the latter). The authors deliberately blacken the picture by retaining values that should obviously be discarded outside the relevant interval between 1003.5 and 1006.5 (namely $1005 \pm 1.5$).

Moreover, I am still not convinced by the possible applications in NWP and climate models of the quantity $\theta_{ref}$ defined by the authors. I wouldn't recommend them for our GCM and NWP models in any case. And section 7.1 is a bit caricatural, although interesting in fact, since it now only shows that one should not apply the usual and inaccurate formula (2) with $\theta_{ref}$, namely $N^2 = (g/\theta_{ref})\partial\theta_{ref}/\partial z$, and thus simply apply formulations of DK82 or MG13.

However, I appreciate the consideration of the criticisms and comments made about the first version of the paper. **Accordingly, my opinion is that the revised version of the article deserves to be published.**

I would suggest one last modification, because on rereading the paper it seems to me that it is difficult on first reading (and even on subsequent readings) to know how $\theta_{ref}$ is calculated. I would suggest in between present Eqs. (23)-(24), and by possibly replacing some words in lines 386-395, to add:

$$F(\theta_{ref}) \; = \; \int_{\theta_{ref}}^{T} \frac{c_p^0(T')}{T'} \, dT' \; - \; R_a \, \ln\left(\frac{p}{p_0}\right) \; = \; 0$$

or even the more simple versions:

$$\int_{\theta_{ref}}^{T} \frac{c_p^0(T')}{T'} \, dT' \; - \; R_a \, \ln\left(\frac{p}{p_0}\right) \; = \; 0$$

or:

$$\int_{\theta_{ref}}^{T} \frac{c_p^0(T')}{T'} \, dT' \; = \; R_a \, \ln\left(\frac{p}{p_0}\right)$$

---

## Author Response (AR3)

**Response to Reviewer Comments**

Manuel Baumgartner[1,2], Ralf Weigel[2], Allan H. Harvey[3],
Felix Plöger[4], Ulrich Achatz[5], and Peter Spichtinger[2]

[1]*Zentrum für Datenverarbeitung, Johannes Gutenberg
University Mainz, Germany*
[2]*Institute for Atmospheric Physics, Johannes Gutenberg
University Mainz, Germany*
[3]*Applied Chemicals and Materials Division, National
Institute of Standards and Technology, Boulder, CO, USA*
[4]*Forschungszentrum Jülich GmbH, Institute of Energy and
Climate Research (IEK-7), Jülich, Germany*
[5]*Institut für Atmosphäre und Umwelt, Goethe-Universität
Frankfurt, Frankfurt am Main, Germany*

November 2, 2020

**Contents**

**1 General Response**

Once again we would like to thank both reviewers for the second revision of the manuscript, on the one hand for the trust and confidence expressed by the anonymous reviewer 2, and on the other hand for Pascal Marquet's repeatedly very helpful comments and suggestions, which again led to a further improved manuscript version.

This document contains our responses (black text) to the reviewer comments (cited in blue text). Moreover, the output of "latexdiff" is included to highlight the changes made to the manuscript (blue text is added, red text is removed; note that the changes made to the figures 2 and 4 are not highlighted).

**2 Response to Pascal Marquet**

I still disagree with the use of too extreme and unrealistic values of $C_p$. Indeed, the values $C_p = 1010$ in the new cited papers Chang et al (2010) and Tiwary Williams (2019) are only indicative (it is a mere scale value for computing the LMO for the latter). The authors deliberately blacken the picture by retaining values that should obviously be discarded outside the relevant interval between 1003.5 and 1006.5 (namely $1005 \pm 1.5$).

We note that in the textbook by Tiwary and Williams (2019) the value $1010 \, \mathrm{J \, kg^{-1} \, K^{-1}}$ does not seem to be used only as a scale value. Unfortunately, within this book, $c_p$ is introduced twice explicitly with $1010 \, \mathrm{J \, kg^{-1} \, K^{-1}}$:

- Section 8.1.1.1, for the calculation of the dry adiabatic lapse rate (below equation 8.8) and

- Section 8.1.2.3.2 in connection with the determination of the Monin-Obukhov length (below equation 8.18).

Further equations throughout this textbook refer to the heat capacity, e.g. for the calculation of the potential temperature (Section 8.1.1.4). However, none of the applications in this book indicates the use of this specific $c_p$ value as a scale value.

For the calculation in equation (7) provided in Chang et al. (2006), $c_p$ is also given explicitly as $1010 \, \mathrm{J \, kg^{-1} \, K^{-1}}$ without any further explanation.

In the textbook by Brusseau et al. (2019), which is also listed in Table 1 of the manuscript, $c_p$ is also explicitly introduced with $1010 \, \mathrm{J \, kg^{-1} \, K^{-1}}$ (Section 4.7, page 59) in connection with an exercise.

It is common to all three references that they could be used for the education and training of students and scientists, or as a literature source for any research,

as all other references could be used, which are listed in Table 1 of the manuscript. Our intention was and is to account for the surprisingly wide range of values found in the literature in connection with a physical quantity, which is generally assumed to be constant ([citation of our manuscript]: "In Table 1, some of the available values of constant specific heat capacity for dry air are compiled, indicating a variability of $c_p$ that ranges from $994\,\mathrm{J\,kg^{-1}\,K^{-1}}$ to $1011\,\mathrm{J\,kg^{-1}\,K^{-1}}$. However, the extremes in Table 1 are from old references of historical interest only; ...", lines 192-194). Indeed, we would like to sensitise the reader to the fact that there is a much narrower range of reasonable values for $c_p$, in line with the objection raised. Particularly the synopsis of the entire range of $c_p$ values illustrates how sensitive calculations (e.g. of $\theta$) are to the use of different, albeit referenced input values. Indeed, there are a few, more reasonable values for $c_p$, e.g. in the interval between $1003.5\,\mathrm{J\,kg^{-1}\,K^{-1}}$ and $1006.5\,\mathrm{J\,kg^{-1}\,K^{-1}}$ (namely $1005\,\mathrm{J\,kg^{-1}\,K^{-1}} \pm 1.5\,\mathrm{J\,kg^{-1}\,K^{-1}}$) ([citation of our manuscript]: "The green curve corresponds to the more realistic $c_p$ interval $1005\,\mathrm{J\,kg^{-1}\,K^{-1}} \pm 1.5\,\mathrm{J\,kg^{-1}\,K^{-1}}$ as recommended by the WMO; the difference reaches approximately 13 K at the stratopause.", lines 210-212). Hence, in the revised manuscript we added graphs in Figure 2 (Panel b) and Figure 4 (Panel b) to account for this narrower, more reasonable $c_p$ range.

Moreover, I am still not convinced by the possible applications in NWP and climate models of the quantity $\theta_{\mathrm{ref}}$ defined by the authors. I wouldn't recommend them for our GCM and NWP models in any case. And section 7.1 is a bit caricatural, although interesting in fact, since it now only shows that one should not apply the usual and inaccurate formula (2) with $\theta_{\mathrm{ref}}$, namely $N^2 = (g/\theta_{\mathrm{ref}})\partial\theta_{\mathrm{ref}}/\partial z$, and thus simply apply formulations of DK82 or MG13.

However, I appreciate the consideration of the criticisms and comments made about the first version of the paper. Accordingly, my opinion is that the revised version of the article deserves to be published.

We agree with Pascal Marquet's remark that the application of the introduced reference potential temperature must be considered individually ([citation of our manuscript]: "The purpose of this examination is to document the magnitude of errors to allow a well-founded, individual decision for each application of the potential temperature whether it is worth applying the more rigorous calculation in the particular context", lines 589-591). We are extremely grateful to Pascal Marquet for his critiques and comments, which have contributed significantly to the improvement of the present article, and we appreciate his opinion that the manuscript deserves publication after this revision.

I would suggest one last modification, because on rereading the paper it seems to me that it is difficult on first reading (and even on subsequent readings) to know how $\theta_{\mathrm{ref}}$ is calculated. I would suggest in between present Eqs. (23)-(24),

and by possibly replacing some words in lines 386-395, to add:

$$F(\theta_{ref}) = \int_{\theta_{ref}}^{T} \frac{c_p^0(T')}{T'} \, dT' - R_a \ln\left(\frac{p}{p_0}\right) = 0$$

or even the more simple versions:

$$\int_{\theta_{ref}}^{T} \frac{c_p^0(T')}{T'} \, dT' - R_a \ln\left(\frac{p}{p_0}\right) = 0$$

or:

$$\int_{\theta_{ref}}^{T} \frac{c_p^0(T')}{T'} \, dT' = R_a \ln\left(\frac{p}{p_0}\right)$$

Furthermore, we fully agree with Pascal Marquet on his proposal and have inserted the following expression (as new equation 24) after line 395:

$$0 = F(\theta_{\text{ref}}) = \int_{\theta_{\text{ref}}}^{T} \frac{c_p(T')}{T'} \, \mathrm{d}T' - R_a \ln\left(\frac{p}{p_0}\right).$$

**3 Response to Timothy Garrett**

I would first like to acknowledge the thorough examination of your work by both reviewers. Both reviewers are recommending the paper for publication. My personal assessment, and I suspect the reviewers would agree, is to congratulate you and your co-authors on such an investigation, for giving the fundamentals of atmospheric sciences a second close look.

However reviewer 1 raises a couple of minor points that should be addressed. Most concerning is perhaps the argument that the extreme values of the specific heat referred to unnecessarily "blacken the picture". As part of a revised manuscript, I would like to see this issue addressed. In general, it is difficult for any reader to appreciate how such a fundamental parameter as $C_p$ could be substantially in error, so a convincing, very sober assessment is especially important.

If the final comments of Reviewer 1 can be addressed satisfactorily, I will forward the paper for publication in ACP.

We would like to join in the thanks to both reviewers for their contributions. We would also like to acknowledge your contribution to the revision process as editor of this article by enclosing a sentence within the acknowledgements. We highly value the opinion of Pascal Marquet, but we do not believe that our approach corresponds to what we would understand as "blackening the picture". Instead, we are well aware of the validity of the involved extreme values, but would like to emphasise our intention: We wanted to account for the surprisingly wide range of values found in the literature in connection with a physical quantity that is generally assumed to be constant. We have indicated in the text of the manuscript that some of the values in Table 1 stem from old references which are of historical interest only. However, we intended to make the reader aware of the danger inherent with the variety of given $c_p$ values and of the fact that there is a much narrower range of reasonable values for $c_p$, in line with Pascal Marquet's objection. We believe that a view of the full range of $c_p$ values from the literature illustrates the sensitivity of calculations when using different input values (potentially in a slightly pedagogic way as indicated by the anonymous reviewer 2 in the review of the initially submitted paper). As the article progresses, the reader is guided to the more reasonable values for $c_p$, (namely around $1005\,\mathrm{J\,kg^{-1}\,K^{-1}}$), which is now better highlighted in the article. In the revised manuscript, we have modified two graphs (Figure 2, panel b, and Figure 4, panel b) accordingly, and we hope that we have met the requested treatment of this topic.

**References**

[revised manuscript text omitted]

$$
\begin{aligned}
\mathrm{AHR_{ref}}\left(\frac{dq}{dt}\right) &= \frac{dT}{dt} = \frac{1}{c_p^0(T)}\frac{dq}{dt}, \\
\mathrm{AHR_{1005}}\left(\frac{dq}{dt}\right) &= \frac{dT}{dt} = \frac{1}{1005\,\mathrm{J\,kg^{-1}K^{-1}}}\frac{dq}{dt}.
\end{aligned}
\tag{38}
$$

Again, the distinction was made between the temperature-dependent $c_p^0(T)$ and the constant $c_p = 1005\,\mathrm{J\,kg^{-1}K^{-1}}$ specific heat capacity. From the defining equations (38), the relative difference between these absolute heating rates, where $x$ designates an arbitrary diabatic heating rate, is

$$
\frac{\mathrm{AHR_{1005}}(x) - \mathrm{AHR_{ref}}(x)}{\mathrm{AHR_{ref}}(x)} = \frac{c_p^0(T)}{1005\,\mathrm{J\,kg^{-1}K^{-1}}} - 1.
\tag{39}
$$

Apart from the absolute heating rates for the change of absolute temperature, the change of potential temperature due to a diabatic heating rate $\frac{dq}{dt}$ is of interest.  For example, it is the change of potential temperature that modifies the altitude of modelled trajectories in Lagrangian chemical transport models based on isentropic coordinates rather than the change in absolute temperature (e.g., the SLIMCAT (Chipperfield, 2006) or CLaMS model (Pommrich et al., 2014)).

Taking the relation $T\,ds = dq$ for the specific entropy into account, Gibbs' equation (8) may be rewritten as

$$
\frac{dq}{T} = \frac{c_p(T)}{T}\,dT - R_a\frac{dp}{p}.
\tag{40}
$$

Comparing the right-hand side of this equation to the total derivative of the new reference potential temperature $\theta_{\mathrm{ref}}$ (see Appendix E for the detailed computation and Equation (E6) for the result) equation (40) amounts to

$$
\frac{dq}{T} = c_p(\theta_{\mathrm{ref}})\frac{d\theta_{\mathrm{ref}}}{\theta_{\mathrm{ref}}}.
\tag{41}
$$

Consequently, the following two diabatic heating rates

$$
\begin{aligned}
\frac{d\theta_{\mathrm{ref}}}{dt} &= \frac{\theta_{\mathrm{ref}}}{c_p^0(\theta_{\mathrm{ref}})T}\frac{dq}{dt} = \mathrm{HR_{ref}}\left(\frac{dq}{dt}\right), \\
\frac{d\theta_{1005}}{dt} &= \frac{\theta_{1005}}{(1005\,\mathrm{J\,kg^{-1}K^{-1}})\cdot T}\frac{dq}{dt} = \mathrm{HR_{1005}}\left(\frac{dq}{dt}\right)
\end{aligned}
\tag{42}
$$

for the potential temperatures $\theta_{\mathrm{ref}}$ and $\theta_{1005}$ may be defined. Denoting again by $x$ an arbitrary diabatic heating rate, the relative difference between the heating rates (42) is

$$
\frac{\mathrm{HR_{1005}}(x) - \mathrm{HR_{ref}}(x)}{\mathrm{HR_{ref}}(x)} = \frac{\theta_{1005}}{\theta_{\mathrm{ref}}}\frac{c_p^0(\theta_{\mathrm{
[revised manuscript text omitted]